# Autophagosomes fuse to phagosomes and facilitate the degradation of apoptotic cells in *Caenorhabditis elegans*

**Omar Peña-Ramos, Lucia Chiao, Xianghua Liu†, Xiaomeng Yu, Tianyou Yao, Henry He‡, Zheng Zhou***

Verna and Marrs McLean Department of Biochemistry and Molecular Biology, Baylor College of Medicine, Houston, United States

**\*For correspondence:**
zhengz@bcm.tmc.edu

**Present address:** †Caris Life Sciences, Inc, Tempe, United States; ‡Department of Neurology Residency Program, University of Texas Southwestern Medical Center, Dallas, United States

**Competing interest:** The authors declare that no competing interests exist.

**Abstract** Autophagosomes are double-membrane intracellular vesicles that degrade protein aggregates, intracellular organelles, and other cellular components. During the development of the nematode *Caenorhabditis elegans*, many somatic and germ cells undergo apoptosis. These cells are engulfed and degraded by their neighboring cells. We discovered a novel role of autophagosomes in facilitating the degradation of apoptotic cells using a real-time imaging technique. Specifically, the double-membrane autophagosomes in engulfing cells are recruited to the surfaces of phagosomes containing apoptotic cells and subsequently fuse to phagosomes, allowing the inner vesicle to enter the phagosomal lumen. Mutants defective in the production of autophagosomes display significant defects in the degradation of apoptotic cells, demonstrating the importance of autophagosomes to this process. The signaling pathway led by the phagocytic receptor CED-1, the adaptor protein CED-6, and the large GTPase dynamin (DYN-1) promotes the recruitment of autophagosomes to phagosomes. Moreover, the subsequent fusion of autophagosomes with phagosomes requires the functions of the small GTPase RAB-7 and the HOPS complex. Further observations suggest that autophagosomes provide apoptotic cell-degradation activities in addition to and in parallel of lysosomes. Our findings reveal that, unlike the single-membrane, *L*C3-*a*ssociated *p*hagocytosis (LAP) vesicles reported to facilitate phagocytosis in mammals, it is the canonical double-membrane autophagosomes that facilitate the clearance of *C. elegans* apoptotic cells. These findings add autophagosomes to the collection of intracellular organelles that contribute to phagosome maturation, identify novel crosstalk between the autophagy and phagosome maturation pathways, and discover the upstream signaling molecules that initiate this crosstalk.

## Editor's evaluation

Peña-Ramos et al., describe a novel interaction between phagosomes and autophagosomes in the degradation of apoptotic cell corpses. Using time-lapse fluorescence microscopy to measure dynamic changes in phagosomes, as well as electron microscopy, the authors follow cell corpse degradation in specific phagocytic cells of developing *C. elegans* embryos. They find that autophagosomes attach to phagosomes and promote their degradation by controlling acidification. The study uncovers a novel function of autophagosomes, and presents a new paradigm for how cell corpses are degraded.

## Introduction

During metazoan development and adulthood, a large number of cells undergo apoptosis; these dying cells are engulfed by phagocytes and degraded inside phagosomes, vacuoles composed of the

lipid bilayers originated from the plasma membrane (*Reddien and Horvitz, 2004*; *Nagata, 2018*). Swift engulfment and degradation of apoptotic cells are critical for tissue remodeling, the resolution of the wound area, and the prevention and suppression of harmful inflammatory and autoimmune responses induced by the content of the dying cells (*Nagata, 2018*). Critical to the degradation of phagosomal contents is the fusion of intracellular organelles, including lysosomes and early endosomes, to phagosomes, which results in the delivery of the content of these organelles to the phagosomal lumen (*Levin et al., 2016*). Lysosomes, which contribute many kinds of hydrolytic enzymes, including proteases, nucleases, lipases, and hydrolyzing enzymes for polysaccharides to the lumen of phagosomes, are the most pivotal organelles that support phagosomal degradation (*Levin et al., 2016*). The fusion of lysosomes to phagosomes also helps acidify the phagosomal lumen, creating a low pH condition in which the digestive enzymes are active (*Levin et al., 2016*). Besides lysosomes and endosomes, whether other kinds of intracellular organelles fuse to phagosomes and contribute to the degradation of the apoptotic cells inside remains elusive.

Mammalian microtubule-associated protein 1 light chain 3 (MAP1-LC3, or LC3) protein is a member of the ATG8 protein family (*Schaaf et al., 2016*). LC3 molecules conjugated to the lipid phosphatidylethanolamine (PE) are most often observed on the surfaces of autophagosomes, double-membrane organelles that are a key structure of autophagy (*Schaaf et al., 2016*). In fact, LC3 is a well-established marker for autophagosomes (*Schaaf et al., 2016*). In mammalian cells, lipidated LC3 molecules were also reported to label a novel kind of vesicles referred to as LC3-associated phagocytosis (LAP) vesicles, which are single-membrane vesicles (*Sanjuan et al., 2007*). LAP vesicles were reported to be the third kind of intracellular vesicles that fuse to phagosomes and facilitate the degradation of apoptotic cells in mice (*Green et al., 2016*; *Martinez et al., 2011*; *Martinez et al., 2016*).

Autophagy is an evolutionarily conserved cellular event that plays an essential role in maintaining cellular homeostasis by enveloping harmful protein aggregates and damaged cellular organelles in double-membrane autophagosomes and subsequently degrading them via fusion with lysosomes (*Morishita and Mizushima, 2019*). Autophagy also supports cell survival during nutrient starvation by capturing intracellular organelles into autophagosomes and converting them to nutrients and energy sources (*Morishita and Mizushima, 2019*). Autophagosome formation requires the organized action of a set of proteins known as **au**topha**g**ic related (ATG) proteins. It is a process of three sequential steps: initiation, nucleation, and expansion, until an autophagosome fully forms and closes (*Nakatogawa, 2020*). After formation, autophagosomes undergo a maturation process through fusion with lysosomes, which provide digestive enzymes to degrade autophagosomal contents, generating a fusion product referred to as 'autolysosomes' (*Nakatogawa, 2020*). In the nematode *C. elegans*, several autophagy genes have been reported to facilitate the clearance of apoptotic cells (*Cheng et al., 2013*; *Huang et al., 2013*; *Li et al., 2012*). However, it is unknown whether autophagosome, as a particular type of cellular organelle, is involved in the clearance of apoptotic cells or whether these *atg* genes have additional functions, such as forming LAP vesicles.

Although both are labeled with LC3, mammalian LAP vesicles and autophagosomes are different in several key aspects. First, LAP vesicles are single-membrane vesicles, whereas autophagosomes are double-membrane vesicles (*Sanjuan et al., 2007*; *Nakatogawa, 2020*). In addition, although the formation of LAP vesicles relies on many autophagy genes, *ulk1*, *atg13*, and *atg14*, three genes whose products act in the initiation complexes for autophagosomes, are dispensable for the generation of LAP vesicles (*Sanjuan et al., 2007*; *Martinez et al., 2011*; *Martinez et al., 2015*). During the initiation of autophagosomes formation, which starts with the appearance of a membrane structure known as a phagophore, ULK1, a serine-threonine kinase, forms a protein complex with ATG13 and two other proteins and phosphorylates the Class-III phosphoinositide 3-kinase (PI3-kinase) VPS34 as well as the rest of the VPS34 complex (ATG6, ATG14, and VPS15), triggering the production of PtdIns(3)P on the phagosphore (*Nakatogawa, 2020*). These distinct features of autophagosomes and LAP vesicles are critical for distinguishing whether an LC3-labeled vesicle is an autophagosome or a LAP vesicle.

During *C. elegans* embryonic development, 131 somatic cells undergo apoptosis and are swiftly engulfed and degraded by neighboring cells (*Sulston and Horvitz, 1977*; *Sulston et al., 1980*). Apoptotic cells display a 'button-like' structure under the Differential Contrast Interference (DIC) microscopy and are referred to as cell corpses (*Sulston and Horvitz, 1977*; *Sulston et al., 1980*). Mutants defective in the clearance of cell corpses exhibit an increased number of persistent cell corpses, a phenotype known as *ce*ll *d*eath abnormal (Ced) (*Ellis et al., 1991*). In the *C. elegans* hermaphrodite

gonad, 300–500 germ cells undergo apoptosis and are cleared by the neighboring gonadal sheath cells (*Gumienny et al., 1999*). Previous genetic studies revealed two parallel, partially redundant pathways that primarily drive the clearance of *C. elegans* cell corpses. These include a signaling pathway led by CED-1, a phagocytic receptor for apoptotic cells, CED-6, an adaptor protein for CED-1, and DYN-1, a large GTPase playing many roles in membrane trafficking, and the other pathway led by the small Rac1 GTPase CED-10, and CED-5 and CED-12, the bipartite nucleotide exchange factor for CED-10 (*Mangahas and Zhou, 2005*). Unlike the CED-10 pathway, which primarily regulates cell corpse engulfment, the CED-1 pathway has two separate functions, regulating both the engulfment and degradation of cell corpses (*Yu et al., 2008*). CED-1 on neighboring engulfing cells recognizes the 'eat me' signal on the surfaces of cell corpses and is enriched to the region of the plasma membrane facing a cell corpse (*Zhou et al., 2001*). This enrichment initiates the extension of pseudopods along the cell corpse and the subsequent closure of the phagocytic cup to form a nascent phagosome (*Yu et al., 2006*; *Shen et al., 2013*). Moreover, CED-1 plays a distinct role in initiating the degradation of phagosomal contents (*Yu et al., 2008*). CED-1 remains transiently enriched on the surface of nascent phagosomes, where it facilitates the CED-6 and DYN-1-mediated sequential recruitment of the Class II PI3-kinase PIKI-1, the Class III PI3-kinase VPS-34, and the small GTPases RAB-5 and RAB-7 to phagosomal surfaces (*Yu et al., 2008*; *Lu et al., 2012*). The robustly produced PtdIns(3)P and the RAB-5 and RAB-7 GTPases further recruit effectors for PtdIns(3)P and the RAB proteins, respectively, which drive the recruitment and fusion of early endosomes and lysosomes to a phagosome, leading to the degradation of the cell corpse inside (*Lu et al., 2012*; *Lu et al., 2011a*; *Lu and Zhou, 2012*). RAB-7, in particular, is essential for the fusion of lysosomes to phagosomes (*Yu et al., 2008*).

Autophagy and phagocytosis are two distinct lysosomal-mediated cellular degradation pathways designated to eliminate intracellular and extracellular components, respectively. Previously, whether canonical autophagosomes were involved in the degradation of phagosomal contents was unknown. We report here that, in *C. elegans*, during the maturation of phagosomes that contain apoptotic cells, LC3-labeled, double-membrane canonical autophagosomes are recruited to phagosomal surfaces and subsequently fuse to these phagosomes. We have further discovered that this event facilitates the degradation of apoptotic cells and is driven by the signaling pathway led by CED-1. This autophagosome-phagosome fusion represents a novel mechanism that contributes to the degradation of phagosomal contents and reveals a new function of autophagosomes.

## Results

### Vesicles labeled with GFP-tagged LC3 are recruited to the surfaces of phagosomes

The ATG8 protein family is composed of two subfamilies, the LC3 subfamily and the GABARAP subfamily, which are very close to each other in sequence; the ATG8 family is also referred to as the LC3/GABARAP family (*Schaaf et al., 2016*). *C. elegans* has two LC3/GABARAP family members, LGG-1 and LGG-2, which belong to the LC3 and GABARAP subfamilies, respectively (*Figure 1A*; *Manil-Ségalen et al., 2014*). Both LGG-1 and LGG-2 are attached to autophagosomes, except they each label autophagosomes of different maturity (*Manil-Ségalen et al., 2014*).

To determine whether autophagosomes interact with phagosomes that contain apoptotic cells in *C. elegans* embryos, we constructed the GFP-tagged LGG-1 and LGG-2 reporters that were expressed under the control of the *ced-1* promoter ($P_{ced-1}$), a well-documented engulfing cell-specific promoter (*Zhou et al., 2001*; *Lu et al., 2009*). In embryos, we observed numerous GFP::LGG-1[+] and GFP::LGG-2[+] puncta (*Figure 1D and G*). Using our previously established time-lapse recording protocol (*Lu et al., 2009*), we observed the enrichment of GFP::LGG-1[+] and GFP-LGG-2[+] puncta to the surfaces of the phagosomes, including the phagosomes containing apoptotic cells C1, C2, and C3 (*Figure 1B, D and G*). C1, C2, and C3 are localized to the ventral surface of an embryo and undergo apoptosis at approximately 330 min post-1st cleavage (*Lu et al., 2009*). Apoptotic C1, C2, and C3 are each engulfed and degraded by a particular ventral hypodermal cell (*Figure 1B*). Inside the engulfing cell that expresses GFP::LGG-1 in the cytoplasm, the phagosomes containing C1, C2, and C3 (also referred to as phagosomes C1, C2, and C3 for convenience) appear like dark discs (*Figure 1D and G*). The GFP-labeled puncta were not observed inside the phagosomal lumen (*Figure 1D and G*, top panels).

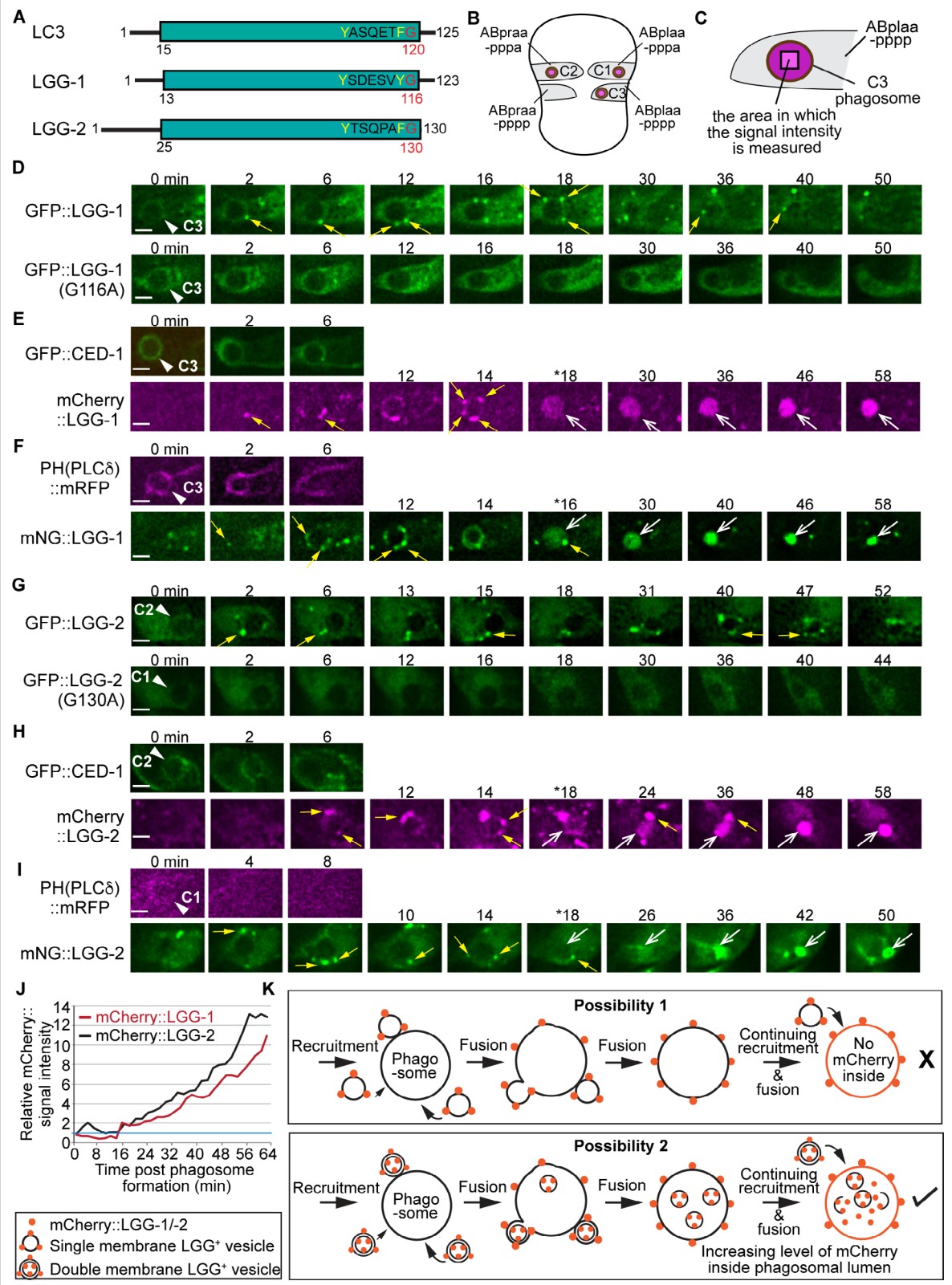

**Figure 1.** The vesicles labeled with LGG-1 or LGG-2 are recruited to the surface of phagosomes and subsequently fuse to phagosomes. (**A**) Domain structures of mammalian LC3 and *C. elegans* LGG-1 and LGG-2. The green box indicates the conserved ubiquitin-like domain. Residues in white are conserved among the three proteins. The glycine residue in red is the site where the lipid tail is attached to. (**B**) A diagram illustrating the three phagosomes that contain cell corpses C1, C2, and C3, with which we monitor the dynamic recruitment and fusion of autophagosomes, at ~330 min

*Figure 1 continued on next page*

*Figure 1 continued*

post-the 1st embryonic division. Both the positions of C1, C2, and C3 (brown dots) and the identities of their engulfing cells are shown. (**C**) A diagram illustrating that the relative mNG signal in the center of a phagosome is measured over time to create sub-figure (**J**). At time point T (time after engulfment), the Relative signal intensity $_T$ = (Unit Intensity(phagosome center)$_T$ – Unit Intensity (background)$_T$) / (Unit Intensity(phagosome center)$_{T0}$ – Unit Intensity(background)$_{T0}$). (*Figure 1—source data 1*). (**D–I**) Time-lapse images of indicated reporters starting when a nascent phagosome (white arrowheads) just formed (time point '0 min'). All reporters were expressed under the control of P$_{ced-1}$. Scale bars are 2 µm. Solid white arrowheads label nascent phagosomes. Yellow arrows mark a few LGG-labeled puncta on the surface of phagosomes. Open white arrows in (**E, F, H, I**) label the phagosomal lumen where the LGG signal is observed. '*' is the time point when the LGG signal is first seen inside the phagosomal lumen. CED-1::GFP (**E, H**) and PH(PLCγ)::mRFP (**F, I**) are co-expressed markers that label the surfaces of nascent phagosomes. (**D**) GFP::LGG-1-labeled puncta are observed on the surface of a C3 phagosome, but the GFP signal is not seen inside the phagosomal lumen. No GFP::LGG-1(G116A)-labeled puncta are seen on the surface of phagosomes. (**E–F**) The mCherry::LGG-1 (**E**) and mNG::LGG-1 (**F**) puncta are observed on the surface of a C3 phagosome and subsequently accumulate inside the phagosome lumen. (**G**) GFP::LGG-2-labeled puncta are observed to attach on the surface of a C2 phagosome, but the GFP signal does not enter the phagosomal lumen, whereas no GFP::LGG-1(G130A)-labeled puncta are seen on the surface of phagosomes. (**H–I**) The mCherry::LGG-2 (**H**) and mNG::LGG-2 (**I**) puncta are observed on the surface of a C2 (**H**) or C1 (**I**) phagosome, respectively, and subsequently accumulate inside the phagosome lumen. (**J**) The relative mCherry::LGG-1 or –2 signal intensity in the center of a phagosome (Y-axis) over time (in the 2 min interval) (X-axis). '0 min' indicates the moment when a nascent phagosome just formed. One blue horizontal line indicates value '1', where no signal enrichment above background level is observed. (**K**) A diagram illustrating that those double membrane-vesicles labeled with mCherry::LGG on their outer and inner membranes are recruited to phagosomal surfaces and fused to the phagosomal membrane. After the fusion between the outer membrane of these vesicles and the phagosomal membrane, the mCherry::LGG-tagged inner membrane is released into the phagosomal lumen. The continuing incorporation of these double-membrane vesicles to phagosomes increases the mCherry signal level in the phagosomal lumen over time. If the LGG-1 or LGG-2-labeled vesicles are of a single membrane, no fluorescence signal is expected to enter the phagosomal lumen.

The online version of this article includes the following video, source data, and figure supplement(s) for figure 1:

**Source data 1.** Relative mCherry::LGG-1 and mCherry::LGG-2 signal intesity over time in 1J.

**Figure supplement 1.** The fusion of lysosomal particles to phagosomes results in the incorporation of lysosomal membrane protein CTNS-1 to the phagosomal membrane but not the lumen.

**Figure supplement 2.** Besides the phagosomes containing C1, C2, and C3, the attachment and fusion of LGG$^+$-vesicles to other phagosomes are also observed during embryonic development.

**Figure 1—video 1.** mCherry::LGG-1-labeled vesicles are recruited to the surface of a phagosome and subsequently fuse to the phagosome. https://elifesciences.org/articles/72466/figures#fig1video1

**Figure 1—video 2.** mCherry::LGG-2-labeled vesicles are recruited to the surface of a phagosome and subsequently fuse to the phagosome. https://elifesciences.org/articles/72466/figures#fig1video2

**Figure 1—video 3.** mNG::LGG-1-labeled vesicles are recruited to the surface of a phagosome and subsequently fuse to the phagosome. https://elifesciences.org/articles/72466/figures#fig1video3

*C. elegans* LGG-1 and LGG-2 were both reported to specifically attach to autophagosomes through their lipid tails (*Manil-Ségalen et al., 2014*; *Wu et al., 2015*). To verify that the observed GFP puncta are LGG-labeled lipid vesicles and not artifacts of protein aggregation, we tested two mutant constructs, GFP::LGG-1(G116A) and GFP::LGG-2(G130A), which bear mutations in the lipidation sites of the LGG proteins and are deficient for membrane targeting (*Manil-Ségalen et al., 2014*). We found that both GFP::LGG-1(G116A) and GFP::LGG-2(G130A) display a diffuse cytosolic localization pattern (*Figure 1D*, **G**, bottom panels), in stark contrast to the punctate pattern presented by GFP::LGG-1 and GFP::LGG-2. This result indicates that GFP::LGG-1 and::LGG-2 are membrane attached and thus label lipid vesicles.

## The LGG-tagged puncta that fuse to phagosomes are double-membrane vesicles

The fluorophore within GFP is sensitive to acidic pH (pKa = 6.0) (*Tsien, 1998*), thus its signal diminishes when GFP is inside the acidic lumen of lysosomes and phagosomes. To further monitor the fate of the LGG$^+$ puncta after they are recruited to the surfaces of phagosomes, we replaced GFP with mCherry (pKa <4.5) or mNeonGreen (mNG) (pKa = 5.1), both of which are more resistant than GFP to the acidic pH environment (*Shaner et al., 2004*; *Shinoda et al., 2018*), allowing fluorescence signal inside the phagosomal lumen to be detected. Co-expressed with the mCherry::LGG or mNG::LGG reporter are the CED-1::GFP or PH(hPLCγ)::mRFP reporters, which were used in the time-lapse experiments as markers for the extending pseudopods, allowing us to determine the moment when the pseudopods sealed up and a nascent phagosome was born (*Figure 1E, F, H*, **I**, top panels) (*Zhou et al., 2001*;

*Shen et al., 2013*). In time-lapse image series of the clearance process of C1, C2, and C3, we found that like the GFP::LGG reporters, the mCherry::LGG and mNG::LGG reporters were enriched on the surfaces of phagosomes; moreover, unlike the GFP::LGG reporters, the mCherry and mNG-tagged reporters subsequently entered phagosomal lumen (*Figure 1E, F, H*, I). The fluorescence signal intensity increases over time with the continuous recruitment of the LGG$^+$ puncta on phagosomal surfaces (*Figure 1E, F, H*, I, *Figure 1—videos 1–3*). We measured the intensity of the mCherry::LGG-1 and mCherry::LGG-2 signal in the center of the C3 phagosomes over time from the start of the phagosome (Materials and methods) (*Figure 1C*) and observed over 10-fold increases of the signal intensity within 60 min (*Figure 1J*). The appearance of the membrane-attached mCherry::LGG and mNG::LGG signal in the phagosomal lumen indicates that the LGG$^+$ vesicles fuse with the phagosomal membrane (*Figure 1K*). They further suggest that these vesicles are composed of double membranes labeled with the LGG reporter molecules on both the outer and inner membranes (*Figure 1K*). If these LGG$^+$ vesicles were single-membrane, no membrane-attached LGG reporter would end up inside the phagosomal lumen because, as a result of fusion, the reporter molecules will be retained on the phagosomal membrane (*Figure 1K*). Indeed, when CTNS-1, a lysosomal transmembrane protein, is tagged with mRFP, which is acid-resistant (pKa = 4.5 [*Shaner et al., 2004*]), on its C-terminus, the lysosome-phagosome fusion event resulted in the incorporation of the mRFP signal into the phagosomal membrane; as a result, the CTNS-1::mRFP signal is only observed on the phagosomal surface, not in the phagosomal lumen (*Figure 1—figure supplement 1*; *Yu et al., 2008*).

## The LGG$^+$ vesicles that are incorporated into phagosomes are canonical autophagosomes

The observation that the LGG$^+$ vesicles incorporated into phagosomes are likely double-membrane vesicles reminded us of canonical autophagosomes rather than LAP vesicles. In support of this model, previously, using electron microscopy and immune-gold staining, Manil-Ségalen et al have shown that in *C. elegans* embryos, LGG-1 or LGG-2-labeled vesicles are double-membrane autophagosomes (*Manil-Ségalen et al., 2014*). To further determine whether these vesicles represent canonical autophagosomes, we examined whether loss-of-function mutations of *atg-7*, *atg-13*, and *epg-8*, which are defective for the biogenesis of autophagosomes (*Tian et al., 2010*; *Yang and Zhang, 2011*), impaired the production of these vesicles. *C. elegans atg-13* and *epg-8* encode homologs of mammalian ATG13 and ATG14, respectively, which are essential for the biogenesis of autophagosomes but dispensable for that of LAP vesicles (Introduction) (*Zhang and Baehrecke, 2015*). *atg-7* encodes a homolog of mammalian ATG7, a protein essential for conjugating a phospholipid tail onto the LC3 family proteins and thus for the biogenesis of both autophagosomes and LAP vesicles (*Nakatogawa, 2020*; *Zhang and Baehrecke, 2015*). We first scored whether the mCherry::LGG-1/–2 reporters were observed in the center of 15 phagosomes during the phagosome maturation process (*Figures 1C and 2*), which is indicative of the fusion of double-membrane mCherry::LGG$^+$ vesicles to phagosomes. In wild-type embryos, the steady entry of mCherry into phagosomes over time (*Figure 1E and H*) results in the increase of the average mCherry::LGG-1 and -LGG-2 intensities to 9.2- and 8.4-fold of that at 0 min time point, respectively, at 60 min after the formation of a phagosome (*Figure 2H and J*). However, in *atg-7(bp411)* (*Gomes et al., 2016*) mutant embryos, hardly any mCherry signal was observed inside phagosomes (*Figure 2A, D and G–J*). At 60 min after the phagosome formation, the average mCherry signal intensities were merely 1.2 and 1.4-fold at 0 min time point (*Figure 2H and J*). These observations indicate a lack of LGG$^+$ vesicles that fuse to phagosomes. In *atg-13(bp414)* and *epg-8(bp251)* mutant embryos, similar observations were made except that the defects were slightly weaker (*Figure 2B, C, E, F and G–J*).

We next examined whether the mCherry::LGG-1/-2$^+$ vesicles are produced in cells of the *atg-7*, *atg-9*, *atg-13*, and *epg-8* mutant embryos. In wild-type embryos at mid-embryonic developmental stages (~330, ~ 350, ~ 400 min post-1st embryonic cell division), numerous mCherry$^+$ puncta were observed (*Figure 2—figure supplements 1A and 2A*). In the *atg-7* mutant embryos, rarely any such puncta existed (*Figure 2—figure supplements 1B and 2B*), consistent with a previous report (*Tian et al., 2010*). *atg-9* encodes ATG9, the only transmembrane protein in the core autophagy machinery (*Tian et al., 2010*; *Zhang and Baehrecke, 2015*). ATG9 plays an essential role in the expansion of phagophore and the biogenesis of autophagosomes (*Nakatogawa, 2020*). In *atg-9(bp564)* mutant embryos that expressed mNG::LGG-1 or::LGG-2, much fewer mNG$^+$ puncta that might represent

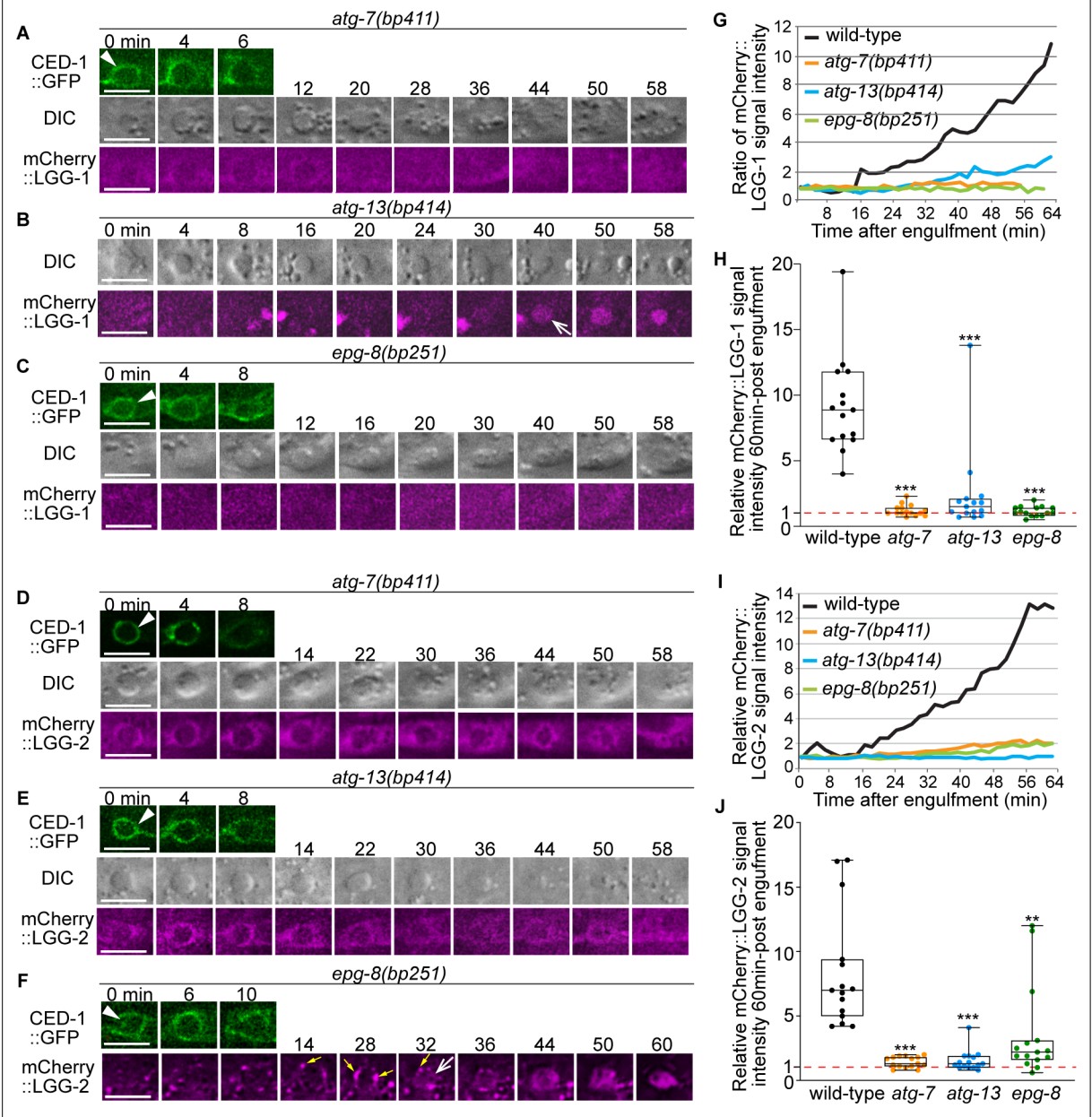

**Figure 2.** In autophagosome-formation mutants, the enrichment of the LGG⁺ vesicles on the phagosomal surface and the entry of the LGG signal into the phagosomal lumen are severely defective. (**A–F**) Time-lapse image series monitoring the enrichment of the puncta labeled with mCherry-tagged LGG-1 (**A–C**) or LGG-2 (**D–F**) on phagosomes (white arrowheads) and the subsequent entry of mCherry signal into the phagosomal lumen in *atg-7*, *atg-13*, and *epg-8* mutant embryos. '0 min' is when a phagosome is just sealed (determined by CED-1::GFP). Open white arrow denotes the time point that the mCherry signal starts to appear inside the phagosomal lumen. Scale bars are 2 µm. Yellow arrows in (**F**) mark mCherry::LGG-2 puncta on the surface of a phagosome. (**G and I**) The relative mCherry::LGG-1 or –2 signal intensity in the center of a phagosome (Y-axis) over time (in the 2 min interval) (X-axis). "0 min" indicates the moment when a phagosome is just sealed. (**G**) The data for the wild-type, *atg-7(bp411)*, *atg-13(bp414)*, and *epg-8(bp251)* mutant embryos are from ***Figure 1E*** and 2(A-C), respectively. (**I**) The data for the wild-type, *atg-7(bp411)*, *atg-13(bp414)*, and *epg-8(bp251)* mutant embryos are from ***Figure 1I*** and 2(D-F), respectively. (***Figure 2—source data 1***). (**H and J**) Box-and-Whiskers plots of the relative mCherry signal intensity measured in the center of phagosomes 60 min-post the formation of nascent C3 phagosomes from 15 each of wild-type, *atg-7(bp411)*, *atg-13(bp414)*, and *epg-8(bp251)* mutant embryos. Red dashed lines indicate the position of value 1, which represents no signal enrichment relative to the background signal. (***Figure 2—source data 2***). ***, p < 0.001, **, < 0.001 p < 0.01, Student *t*-test of each mutant comparing to the wild-type value.

The online version of this article includes the following source data and figure supplement(s) for figure 2:

**Source data 1.** Relative mCherry::LGG-1 and mCherry::LGG-2 singnal intesity over time in ***Figure 2G and I***.

**Source data 2.** Relative mCherry::LGG-1 and mCherry::LGG-2 signal intensity at 60min-post engulfment in ***Figure 2H and J***.

*Figure 2 continued on next page*

Figure 2 continued

**Figure supplement 1.** The *atg-7*, *atg-13*, and *epg-8* mutants are severely defective in the production of LGG-1-labeled autophagosomes.

**Figure supplement 2.** The *atg-7*, *atg-13*, and *epg-8* mutants are severely defective in the production of LGG-2-labeled autophagosomes.

autophagosomes were observed, and the mNG reporters label large aggregates (*Figure 3—figure supplement 1*), consistent with a previous report (*Lin et al., 2013*). In *atg-13* and *epg-8* mutant embryos, the numbers of mCherry-LGG+ puncta were also significantly reduced (*Figure 2—figure supplement 1C-D* and *Figure 2—figure supplement 2C-D*), suggesting that the *atg-13(bp414)* and *epn-8(bp251)* mutations severely impaired the biogenesis of autophagosomes, a phenotype that is consistent with previous reports (*Tian et al., 2010*; *Yang and Zhang, 2011*). The drastic reduction in the number of LGG-1/–2-labeled puncta in *atg-7*, *atg-9*, *atg-13*, and *epg-8* loss-of-function mutants strongly indicates that these puncta belong to canonical autophagosomes.

To further confirm that autophagosomes fuse to phagosomes, we examined the subcellular localization of a mCherry::ATG-9 reporter expressed in engulfing cells. Besides the LGG proteins, ATG-9 is also an established autophagosome marker as it is the only transmembrane protein in autophagosomes (*Lu et al., 2011b*). In embryonic hypodermal cells that co-expressed mCherry::ATG-9 and either mNG::LGG-1 or::LGG-2, puncta labeled with mCherry are recruited to the surfaces of phagosomes (*Figure 3*). Furthermore, the mCherry signal gradually accumulates in the phagosomal lumen like the LGG reporters (*Figure 3*). During the phagosome maturation process, the co-localization between mCherry::ATG-9 and each of the two LGG reporters on the puncta on phagosomal surfaces and inside phagosomal lumen was nearly perfect (*Figure 3*). Together, the above observations verified that the ATG-9 and LGG double-positive autophagosomes are recruited to phagosomal surfaces and subsequently fuse to phagosomes containing apoptotic cells.

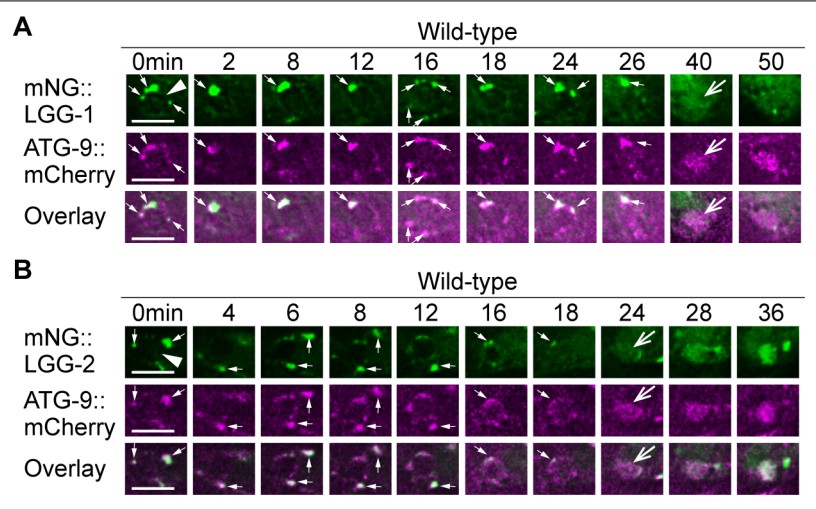

**Figure 3.** LGG-1+ and LGG-2+ puncta colocalize with ATG-9, a component of autophagosomes. The mNG- and mCherry-tagged reporters are expressed in wild-type embryos under the control of P*ced-1*. White arrowheads mark nascent phagosomes. Small white arrows mark the regions where LGG+ and ATG-9+ puncta colocalize. Open white arrows indicate when the fluorescent signal is first detected inside the phagosomal lumen. Scale bars are 5 μm. (**A**) Time-lapse microscopy showing the localization of mNG::LGG-1 and mCherry::ATG-9. Images are from ABplaapppp, which engulfs C3. (**B**) Time-lapse microscopy showing the localization of mNG::LGG-2 and mCherry::ATG-9. Images are from ABplaapppa, which engulfs C1.

The online version of this article includes the following figure supplement(s) for figure 3:

**Figure supplement 1.** *atg-9* is another gene essential for the production of autophagosomes and the incorporation of autophagosomes into phagosomes.

## The autophagosome-phagosome interaction is a general phenomenon observed in embryos and the adult gonad

Besides the C1, C2, and C3 phagosomes, in embryos co-expressing mCherry::LGG-1 or –2 and CED-1::GFP reporters, we also observed the recruitment of the mCherry signal to other phagosomes and the subsequent entry of the mCherry signal to the phagosomal lumen (*Figure 1—figure supplement 2A-E*). Time-lapse image series (*Figure 1—figure supplement 2A-E*) demonstrate the recruitment and fusion of LGG$^+$ puncta to apoptotic cells C4 and C5 in the tail (*Shen et al., 2013*). In mid-stage (1.5-fold) embryos, multiple phagosomes are observed to either have LGG$^+$ puncta attached to the surface (*Figure 1—figure supplement 2 **F(b, d)***, yellow arrows) or LGG$^+$ signal inside the lumen (*Figure 1—figure supplement 2 **F(b,d)***, white open arrows). None of these phagosomes are C1, C2, C3, C4, or C5, judging by their positions in the embryo. These observations indicate that the fusion between autophagosomes and phagosomes is a general phenomenon during embryogenesis.

In each *C. elegans* adult hermaphrodite gonad, germ cells that die of apoptosis undergo cellularization and are engulfed by neighboring gonadal sheath cells (*Gumienny et al., 1999*). We used transmission electron microscopy (TEM) (Materials and methods) to examine whether there were any double-membrane vesicles attaching to phagosomes containing germ cells. In wild-type adults, due to the swift engulfment and degradation activities, on average only 2–4 germ cell corpses are observed in each gonadal arm in an adult hermaphrodite 48 hr post mid-L4 larval stage (*Yu et al., 2008*), making it very difficult to find any phagosomes in the 50 nm thin sections in TEM. We thus chose to analyze phagosomal surfaces in the *rab-7* mutants, in which many germ cell corpses persist in the gonad due to the blockage of phagosome degradation (*Yu et al., 2008*). In TEM images, germ cell corpses are identified by their higher density than live germ cells and complete cellularization (*Yu et al., 2008*). The *rab-7(ok511)* mutants are maternal-effect embryonic lethal. *rab-7(ok511)* adult hermaphrodites are viable and produce dead embryos (*Yu et al., 2008*). In *rab-7(ok511)* mutant adult gonads, many undegraded phagosomes containing germ cell corpses are observed inside the sheath cell in thin (50 nm) TEM sections (*Figure 4B and C*), as reported previously (*Yu et al., 2008*). Furthermore, in the *rab-7* mutant gonad, mCherry::LGG-1 expressed in sheath cells (under P$_{ced-1}$) is observed to accumulate on the surfaces of phagosomes (*Figure 4A*), like in *rab-7* mutant embryos (see Results section 'The small GTPase RAB-7 and the HOPS complex are essential for the fusion between autophagosomes and phagosomes' below). This result is consistent with the observation of LGG$^+$ puncta on phagosomal surfaces in embryos. In *rab-7* mutant gonad, we did identify multiple double-membrane vesicles in close contact with the phagosomal surfaces in TEM thin sections (*Figure 4D–H*). Besides being composed of double membranes, these vesicles also resemble autophagosomes identified in *C. elegans* cells (*Kovacs et al., 2013*; *Zhang et al., 2015*). These vesicles vary in diameter from 200 nm to 800 nm, consistent with that of autophagosomes reported in the literature (*Kovacs et al., 2013*; *Manil-Ségalen et al., 2014*; *Zhang et al., 2015*). The above observations support our conclusion that the LGG$^+$ puncta that are recruited to phagosomal surfaces are indeed canonical autophagosomes. They further suggest that the autophagosome-phagosome interaction is general rather than a cell-specific phenomenon.

## Autophagosomes facilitate the degradation of apoptotic cells inside phagosomes

To examine whether the incorporation of autophagosomes into phagosomes affects the clearance of the engulfed apoptotic cells, we first quantified whether, in mutants of genes essential for the biogenesis of autophagosomes, apoptotic cells were un-degraded and thus persisted in embryos. In addition to the *atg-7*, *atg-9*, *atg-13*, and *epg-8* mutants characterized above, we also characterized loss-of-function mutants of *lgg-1* and *lgg-2*, and of *atg-3*, whose gene product is essential for the conjugation of a phosphatidylethanolamine (PE) tail to the LC3 family proteins (*Tian et al., 2010*; *Wu et al., 2016*), of *atg-2* and *atg-18*, whose gene products function together with ATG-9 in the expansion of phagophore (*Lu et al., 2011b*), and of *unc-51*, which encodes a *C. elegans* homolog of ULK1, an autophagic protein kinase (*Lu et al., 2011b*). In twofold stage wild-type embryos, which are ~460 min post the first embryonic cell division, an average of 11.3 cell corpses were scored (*Figure 5A*). The mutant twofold stage embryos examined bore 46.9–91.2% more cell corpses (*Figure 5*), indicating that the clearance of cell corpses is defective. Together, the findings reported in *Figures 2–5* indicate that autophagosomes made a substantial contribution to the clearance of cell corpses.

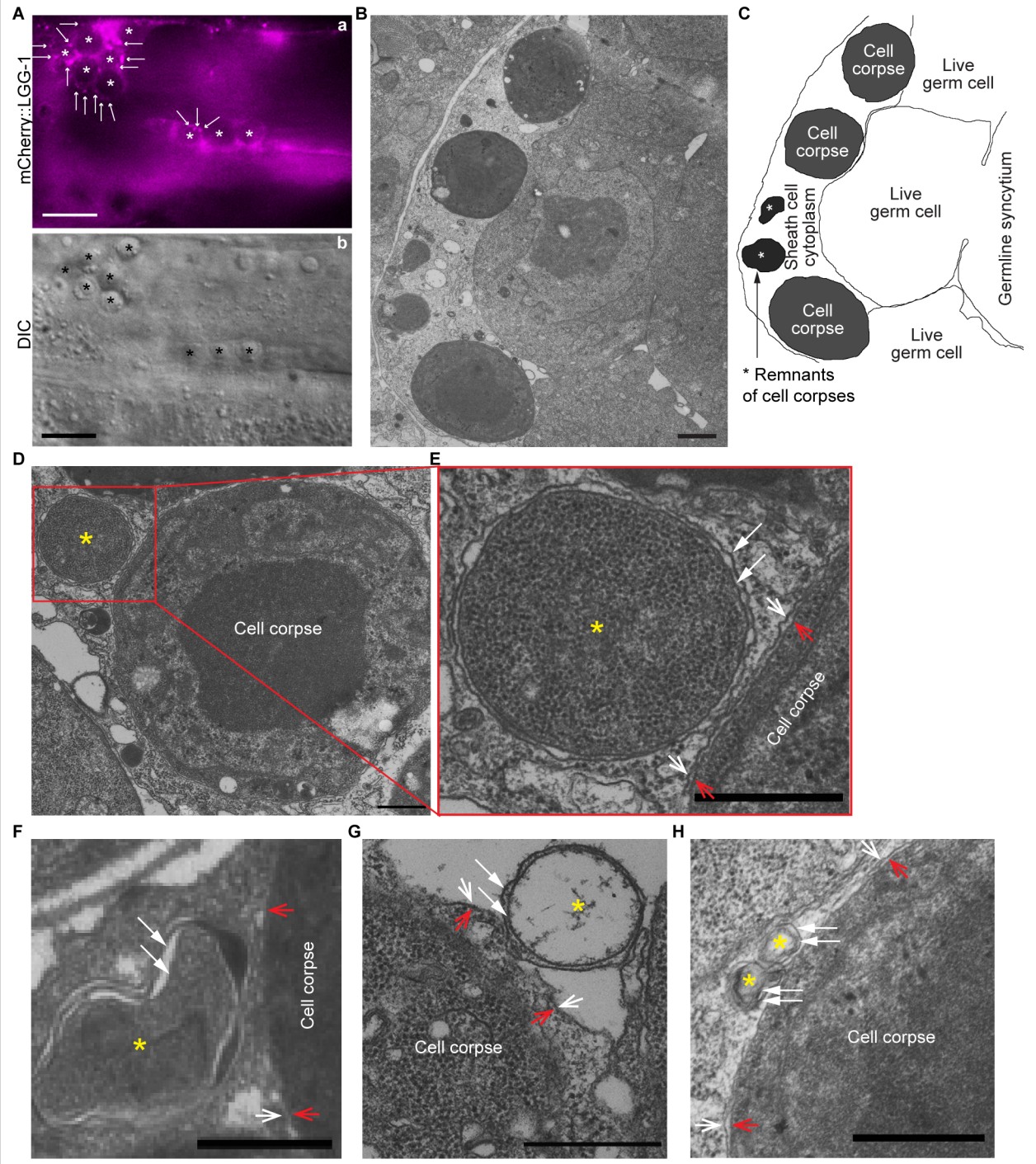

**Figure 4.** Double-membrane vesicles were observed to attach on the surfaces of phagosomes bearing germ cell corpses All samples are from the distal gonadal arms of *rab-7(ok511)* adult hermaphrodites. (**A**) In the gonadal arm of a *rab-7(ok511)* hermaphrodite expressing P*ced-1* *mCherry::lgg-1* in gonadal sheath cells, mCherry+ puncta (arrows) are found on the surfaces of phagosomes carrying germ cell corpses (white (a) and black (b) asterisks). Scale bars are 10 μm. (**B**) A thin cross-section (50 nm in thickness) TEM image of half of a distal gonad. The scale bar is 1 μm. (**C**) Traces of membranes corresponding to (**B**). All three germ cell corpses are inside the gonadal sheath cell. Two asterisks mark the remnants of two engulfed germ cell corpses. (**D-H**) Scale bars are 500 μm. Examples of five double-membrane vesicles (yellow asterisks) were observed on phagosomal surfaces. White arrows mark each layer of the double-layer membranes of the vesicles of interest. Open arrows mark the phagosomal (white) and germ cell corpse (red) membranes. (E) is an enlarged image of the region framed by the red box in (**D**). (**G**) The luminal content of the double-membrane vesicle (*) is missing due to the damage in sample preparation. Due to the same damage, the phagosomal membrane in (G) is unclear.

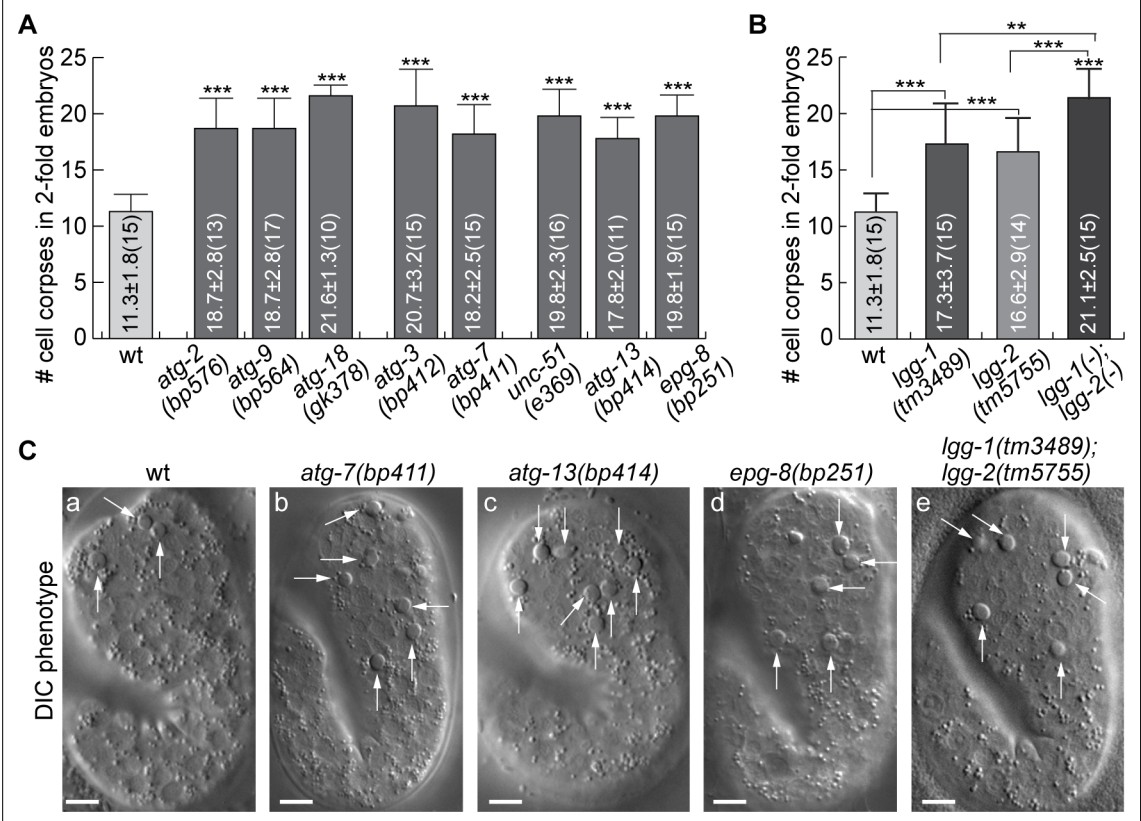

**Figure 5.** Mutations in autophagy genes impair the clearance of apoptotic cells. (**A–B**) Bar graph displaying the average numbers of somatic cell corpses in twofold stage wild-type and various mutant embryos. Bars and error bars represent mean and standard deviation (sd), respectively, the actual values of which are displayed inside the bars. Numbers in parentheses indicate the number of embryos scored. ***, $p < 0.001$, Student $t$-test of each mutant comparing to the wild-type value. (**Figure 5—source data 1**). (**C**) DIC images of cell corpses in twofold stage embryos of various genotypes. White arrows indicate button-like structures characteristic of cell corpses. Scale bars are 5 µm.

The online version of this article includes the following source data and figure supplement(s) for figure 5:

**Source data 1.** Cell corpse count and statistical analysis of atg mutants at 2-fold stage.

**Figure supplement 1.** The expression of *lgg-1* and *lgg-2* cDNA in engulfing cells suppresses the Ced phenotype of *lgg-1* and *lgg-2* null mutants, respectively.

**Figure supplement 1—source data 1.** Cell corpse count and statistical analysis of atg mutants at 1.5-fold stage.

To determine whether the lack of autophagosomes impairs the engulfment or degradation of cell corpses, we monitored the formation and degradation of phagosomes containing apoptotic cells C1, C2, and C3 (**Figure 1B**) in wild-type, *atg-7*, and *lgg* mutant embryos in real-time using an established protocol (Materials and methods) (**Lu et al., 2011a**; **Lu et al., 2009**). The CED-1::GFP expressed in engulfing cells labels the extending pseudopods and enables us to monitor the process of phagosome formation, starting from the budding and ending at the sealing of the pseudopods (**Zhou et al., 2001**; **Yu et al., 2006**). 2xFYVE::mRFP (also expressed under $P_{ced-1}$), a reporter for phagosomal surface PtdIns(3)P, enables us to monitor the shrinking of a phagosome, an indication of phagosome degradation (**Yu et al., 2008**; **Lu et al., 2012**; **Lu et al., 2011a**). In *atg-7*, *lgg-1*, and *lgg-2* mutant strains that co-expressed CED-1::GFP and 2xFYVE::mRFP, we found that engulfment was completed in 4–8 min, just like in wild-type embryos (**Figure 6A–D and F**), indicating that defects in autophagosomes biogenesis do not affect the engulfment of cell corpses. However, the lifespans of phagosomes (Materials and methods) were much longer in *atg-7*, *lgg-1*, and *lgg-2* mutants than in wild-type embryos (**Figure 6A–D and G**). All wild-type phagosomes have a lifespan between 40–60 min (**Figure 6G**). In *atg-7*, *lgg-1*, and *lgg-2* mutants, the lifespan varied in a much more extensive range. Remarkably, 43.8%, 82.4%, and 53.3% of phagosomes in *atg-7*, *lgg-1*, and *lgg-2* mutant embryos, respectively,

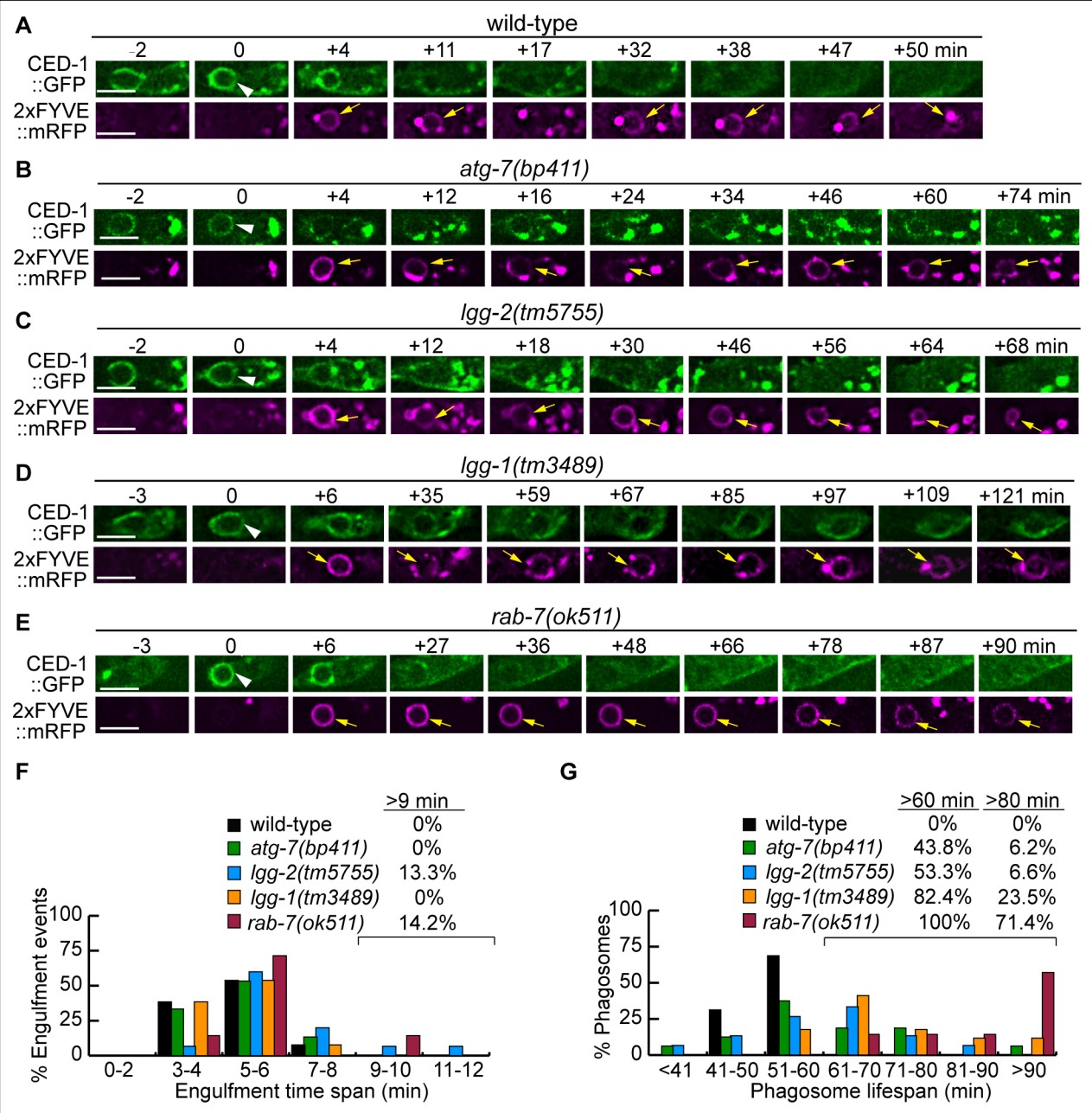

**Figure 6.** Mutations in *rab-7*, *atg-7*, *lgg-1*, and *lgg-2* impair the degradation of cell corpses to different degrees. (**A–E**) Time-lapse recording conducted in wild-type and different mutant embryos monitoring the dynamics of the pseudopod marker CED-1::GFP and the phagosome marker 2xFYVE::mRFP (both expressed in engulfing cells) during the engulfment and degradation processes of cell corpse C3 by ABplaapppp. '0 min' is the first time point when a nascent phagosome (white arrowheads) is formed, as indicated by the closure of a green GFP::CED-1 ring. 2xFYVE::mRFP labels the surface of a phagosome (yellow arrows) until it is degraded. Scale bars are 2 µm. (**F**) Histogram depicting the distribution of the time it takes to engulf 15 C3 cell corpses in wild-type, *atg-7*, *lgg-1*, and *lgg-2* and 7 cell corpses in *rab-7*(m⁻z⁻) homozygous embryos. The engulfment time is defined as the period between the first time when pseudopods (labeled with CED-1::GFP) are observed and when a full circle is observed forms around C3. (*Figure 6—source data 1*). (**G**) Histogram depicting the lifespan distribution of 15 C3 phagosomes in wild-type, *atg-7*, *lgg-1*, *lgg-2*, and seven phagosomes in *rab-7*(m⁻z⁻) homozygous embryos. Phagosome lifespan is measured as the time interval between the '0 min' time point when a nascent phagosome is just sealed and when the phagosome shrinks to one-half of its original diameter. (*Figure 6—source data 2*).

The online version of this article includes the following source data for figure 6:

**Source data 1.** Time of engulfment of C3 phagosomes.

**Source data 2.** Time of degradation of C3 cell corpses.

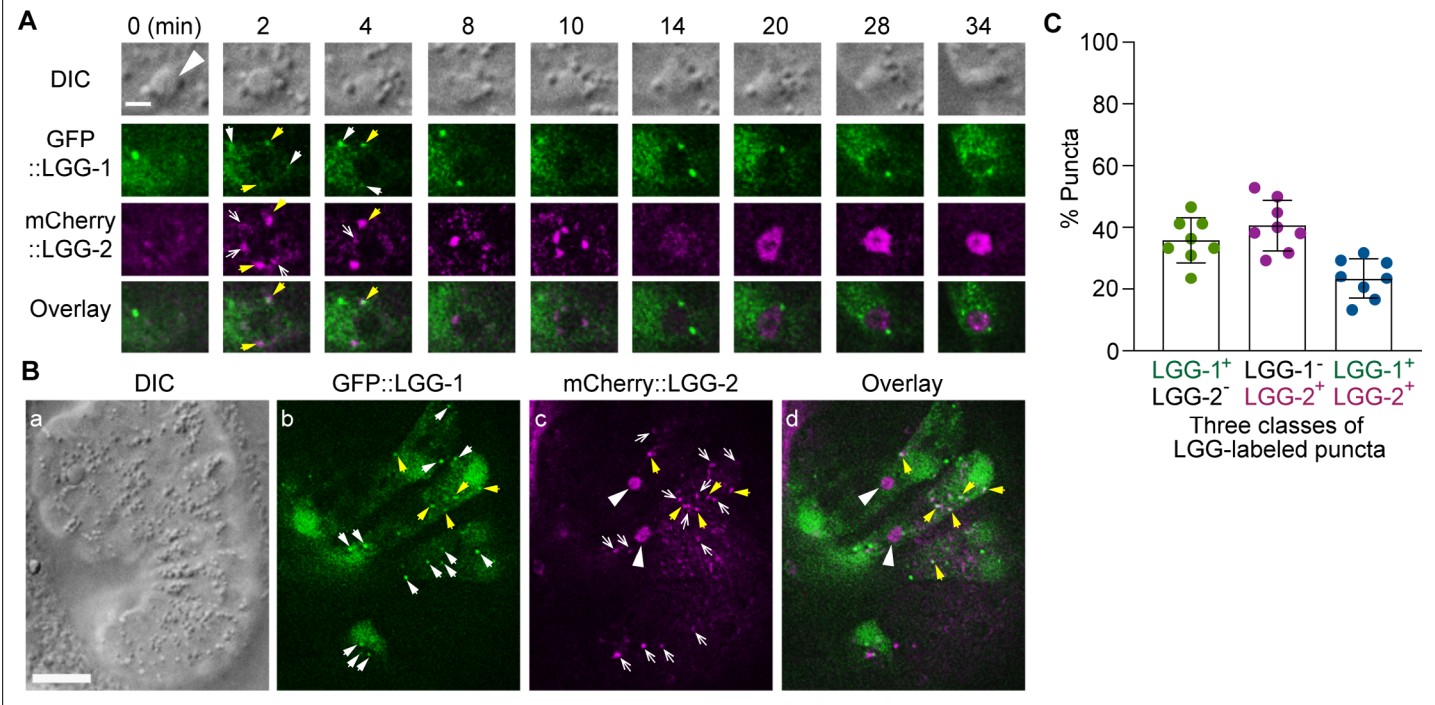

**Figure 7.** The puncta labeled with LGG-1 and/or LGG-2 define three distinct populations of vesicles The GFP- and mCherry- tagged reporters are expressed in wild-type embryos under the control of P$_{ced-1}$. (**A**) Time-lapse image series of a C2 phagosome (big white arrowhead) monitoring the localization of LGG-1$^+$, LGG-2$^+$, and LGG-1$^+$ - LGG-2$^+$ double-positive puncta on the surface of the phagosome. '0 min' is the moment when a nascent phagosome just seals. The scale bar is 2 µm. White arrows mark GFP$^+$ mCherry$^-$ puncta, white open arrows mark GFP$^-$ mCherry$^+$ puncta, and yellow arrows mark GFP$^+$ mCherry$^+$ puncta. (**B**) DIC and fluorescence images of an embryo exhibiting LGG$^+$ puncta outside phagosomes in multiple cells. The scale bar is 10 µm. The white arrows, white open arrows, and yellow arrows mark GFP$^+$ mCherry$^-$, GFP$^-$ mCherry$^+$, and GFP$^+$ mCherry$^+$ puncta on the surface of the phagosome, respectively. The big white arrowheads in (c) and (d) indicate the mCherry signal internalization to the phagosomal lumen. (**C**) Bar graph indicating the distribution of single and double-labeled puncta in the LGG-labeled population in the engulfing cells for C1, C2, and C3, scored immediately prior to the point when the LGG signal was observed inside the phagosomal lumen. Eight engulfing cells were scored. Bars represent the mean, the error bars indicate standard deviation, and each dot represents one sample. (*Figure 7—source data 1*).

The online version of this article includes the following source data for figure 7:

**Source data 1.** Percentage distribution of single and double-labeled puncta in the LGG-labeled population in the engulfing cells for C1, C2, and C3.

have lifespans longer than 60 min (*Figure 6G*). These observations indicate that autophagosomes made a specific contribution to the efficient degradation of phagosomal contents.

## LGG-1 and LGG-2 act in engulfing cells, and together they define three subpopulations of autophagosomes that are incorporated into phagosomes

LGG-1 and LGG-2 have distinct structural features and were observed to attach to different sub-populations of autophagosomes, which represent autophagosomes at different maturation stages (*Manil-Ségalen et al., 2014*; *Wu et al., 2015*). The sub-populations of autophagosomes labeled by LGG-1 or LGG-2 are incorporated into phagosomes (*Figure 1*). *lgg-1* and *lgg-2* single mutants are both inefficient in clearing cell corpses (*Figure 5B*). Furthermore, in *lgg-1; lgg-2* double mutant embryos at the twofold stage, the Ced phenotype is further enhanced significantly (*Figure 5B*) -- the number of cell corpses is 22.0% and 27.1% larger than in the *lgg-1* and *lgg-2* single mutants, respectively. This enhanced phenotype indicates an additive effect and suggests that the autophagosomes labeled with LGG-1 or LGG-2 play parallel and partially redundant roles in promoting phagosome degradation. In embryos co-expressing GFP::LGG-1 and mCherry::LGG-2, we observed that both in the cytoplasm of the engulfing cells and on the surfaces of phagosomes, puncta that were either labeled with GFP or mCherry alone or with both GFP and mCherry (*Figure 7A and B*). In hypodermal cells, the average distribution of LGG-1$^+$ LGG-2$^-$, LGG-1$^-$ LGG-2$^+$, and LGG-1$^+$ LGG-2$^+$ puncta is 35.8%,

40.7%, and 23.5%, respectively (*Figure 7C*). These observations indicate that in addition to the LGG-1[+]-only and LGG-2[+]-only sub-populations, a third, LGG-1[+] LGG-2[+] double-positive sub-population of autophagosomes exists. This sub-population likely corresponds to an intermediate stage in the maturation path of autophagosomes.

To determine whether LGG-1 and LGG-2 act in engulfing cells to facilitate phagosome degradation, we examined whether the specific expression of each gene in engulfing cells, under the control of P*~ced-1~*, would rescue the Ced phenotype of the corresponding mutant embryos. We tested the *lgg* cDNAs that are tagged with either *gfp* or *mCherry* for the rescuing activity by counting the number of cell corpses in 1.5-fold stage transgenic embryos. In the *lgg-1* and *lgg-2* null mutants, both the *gfp*- and *mCherry*-tagged corresponding *lgg* cDNA efficiently rescued the Ced phenotype (*Figure 5—figure supplement 1*). The *gfp::lgg-1* and *mCherry::lgg-1* transgenes lowered the number of cell corpses from on-average 152% of wild-type level observed in the *lgg-1(tm3489)* mutants to 118% and 123% of the wild-type level, respectively. Similarly, the *gfp::lgg-2* and *mCherry::lgg-2* transgenes lowered the number of cell corpses from on-average 143% of wild-type level observed in the *lgg-2(tm5755)* mutants to 109% and 112% of the wild-type level, respectively (*Figure 5—figure supplement 1*). These results indicate that *lgg-1* and *lgg-2* primarily act in engulfing cells to facilitate the clearance of apoptotic cells.

## The small GTPase RAB-7 is enriched on the surfaces of autophagosomes

The small GTPase Rab7 is well known to specifically label late endosomes and lysosomes (*Stenmark, 2009*). In addition, Rab7 proteins in mammalian and *C. elegans* are also recruited from the cytoplasm to the phagosomal membrane shortly after the formation of a phagosome and mediate the fusion between the maturing phagosome and lysosomes and do that through their effector, the HOPs complex (*Levin et al., 2016*; *Lu and Zhou, 2012*). Furthermore, in yeast, *Drosophila*, and mammalian cells, Rab7 is directly recruited to the surfaces of autophagosomes and plays an important role in the fusion between autophagosomes and lysosomes (*Szatmári and Sass, 2014*; *Gao et al., 2018*; *Hegedus et al., 2016*; *Vaites et al., 2018*). *C. elegans* RAB-7 also plays an essential role in the fusion between autophagosomes and lysosomes (*Manil-Ségalen et al., 2014*). To examine whether RAB-7 is localized to LGG-1[+] and LGG-2[+] autophagosomes that fuse to *C. elegans* phagosomes, we generated two transgenic *C. elegans* strains that co-expressed the mCherry::LGG-1/ GFP::RAB-7 or mCherry::LGG-2/ GFP::RAB-7 pairs of reporters (Materials and methods). We observed that GFP::RAB-7 was localized to some but not all of the LGG-1[+] or LGG-2[+] puncta (*Figure 8B–D*). The LGG[+] RAB-7[+] double-positive autophagosomes were observed both freely distributed in the cytoplasm of ventral hypodermal cells (*Figure 8B* (a-c) and C) and on phagosomal surfaces (*Figure 8B* (d-f), D). In addition, *Figure 8B* (d-f) depicts that GFP::RAB-7 is both evenly distributed to the surface of a phagosome (d) as previously reported (*Yu et al., 2008*), and highly enriched on LGG-1[+] autophagosomes that are recruited to the phagosomal surface (d-f, white arrows). *Figure 8D* shows a time-lapse series of a dynamic fusion event of an LGG-2[+]/RAB-7[+] punctum (marked by the bottom arrow in the '0 min' time point) to the phagosome: this punctum is first seen attached to the phagosome membrane and subsequently becoming part of the phagosomal surface at the ' + 6 min' time point. Quantitative analysis of the green and red puncta distribution reveals that on average, 66.2% and 63.5% of LGG-1[+] and LGG-2[+] puncta observed in the cytoplasm of the engulfing cells are RAB-7[+], respectively (*Figure 8E*). The LGG[+] but RAB-7[-] puncta (*Figure 8B and C*, white arrowheads) are likely immature autophagosomes not acquired RAB-7 yet. We also observed puncta that were LGG[-] but RAB-7[+] (*Figure 8B and C*, yellow arrows). These puncta represent intracellular organelles such as late endosomes or lysosomes. Different from the observation made in the cytoplasm, 100% of the autophagosomes that were observed on phagosomal surfaces were RAB-7[+] (*Figure 8B, D and F*). This observation suggests that RAB-7 might play an essential role in the interaction between autophagosomes and phagosomes.

## RAB-7 and the HOPS complex are essential for the fusion between autophagosomes and phagosomes

We examined whether the recruitment and fusion of mCherry::LGG-1[+]- or LGG-2[+]- autophagosomes to the C1, C2, and C3 phagosomes were normal in *rab-7(ok511)* null mutant embryos (*Yu et al., 2008*). The recruitment event can be evaluated by the level of enrichment of mCherry puncta on the surfaces

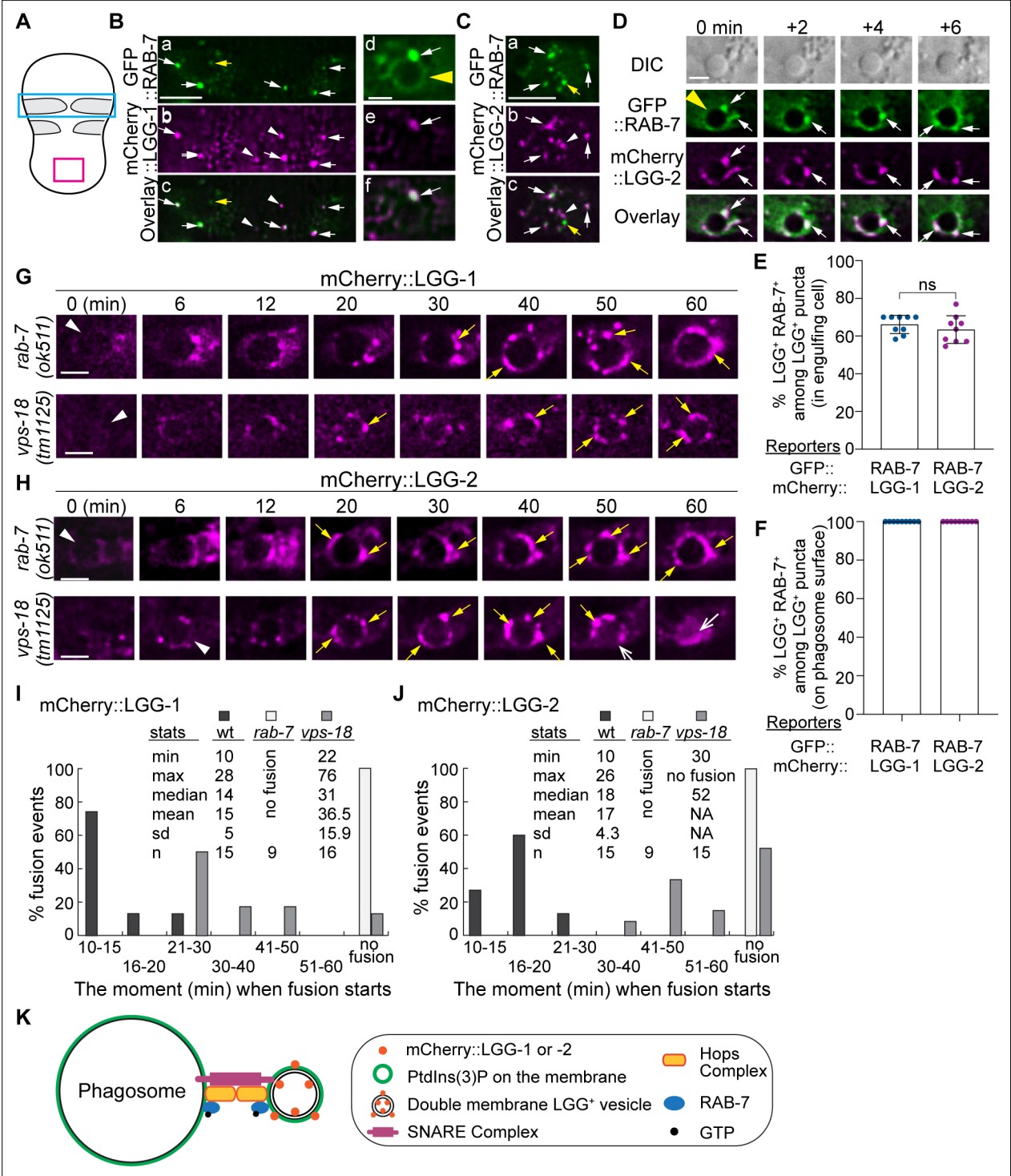

**Figure 8.** RAB-7 is enriched on a portion of autophagosomes, and RAB-7 and VPS-18 are essential for the fusion between autophagosomes and phagosomes. (**A**) Diagram of the ventral surface of an embryo at ~330 min post 1st embryonic division. (**B–D**) Images of part of the ventral surface of an embryo co-expressing P_ced-1_gfp::rab-7 and P_ced-1_mCherry::lgg-1 (**B**) or P_ced-1_mCherry::lgg-2 (**C–D**). B(a-c) depicts the region framed by the blue box in (**A**). B(d-f) depicts a C1 phagosome (a yellow arrowhead). C(a-c) depicts the region framed by the red box in (**A**). (**D**) A time-lapse image series of a C1 phagosome (a yellow arrowhead) indicates the dynamic recruitment and fusion of GFP and mCherry double-positive puncta to the phagosomal membrane. '0 min' is when the first puncta are observed on the phagosomal surface. White arrows mark several puncta that are both GFP+ and mCherry+. Yellow arrows mark puncta that are GFP+ but mCherry-. White arrowheads mark puncta that are GFP- but mCherry+. Scale bars for B(a-c) and (**C**) are 5 μm, and for B(d-f) and (**D**) are 2 μm. (**E–F**) Bar graphs depicting the percentage of LGG-1+ or LGG-2+ puncta that are also RAB-7+ in the

*Figure 8 continued on next page*

*Figure 8 continued*

cytoplasm of the engulfing cells for C1, C2, and C3 (**E**) or on the surfaces of the phagosomes 2 min before the autophagosome-phagosome fusion occurs (**F**). Nine engulfing cells and the phagosomes inside were scored for each of the LGG-1$^+$ and LGG-2$^+$ categories. Bars and error bars represent the mean and standard deviation values. Each dot represents one sample. ns, not significant. (*Figure 8—source data 1*). (**G–H**) Time-lapse images monitoring the recruitment and fusion of puncta labeled with mCherry::LGG-1 (**E**) or::LGG-2 (**F**) to the C1, C2, and C3 phagosomes in *rab-7(ok511)* and *vps-18(tm1125)* mutant embryos. '0 min' represents the moment when a phagosome just seals (white arrowheads). Yellow arrows mark the mCherry$^+$ puncta on the phagosomal surfaces. Open white arrow marks the mCherry signal inside the phagosomal lumen. Scale bars are 2 μm. (**I–J**) Histograms depicting the distribution of the time it takes for LGG-1$^+$ or LGG-2$^+$ puncta to fuse to phagosomes, measured from the '0 min' point to the time when mCherry was detected in the center of a phagosome. C1, C2, and C3 phagosomes were recorded. *n*, the number of phagosomes scored. 'No fusion': no mCherry signal entry was observed even after 72–114 min post-nascent phagosome formation. NA: not applicable. (*Figure 8—source data 2*). (**K**) A Diagram depicts the mechanism driving autophagosome-phagosome fusion. RAB-7 is enriched on the surfaces of both phagosomes and autophagosomes. RAB-7 and VPS-18, a subunit of the HOPs complex, are proven essential for autophagosome-phagosome fusion. Other factors are proposed to play roles in this event based on the general knowledge of intracellular membrane fusion.

The online version of this article includes the following video and source data for figure 8:

**Source data 1.** Percentage of distribution of LGG-1+ or LGG-2+ puncta that are also RAB-7+.

**Source data 2.** The time it takes for LGG-1 or LGG-2 autophagosomes fuse to with phagosomes.

**Figure 8—video 1.** In *rab-7 (ok511),* mCherry::LGG-1-labeled vesicles fail to fuse with phagosomes.

https://elifesciences.org/articles/72466/figures#fig8video1

**Figure 8—video 2.** In *rab-7(ok511)* mutants, mCherry::LGG-2-labeled vesicles fail to fuse with phagosomes.

https://elifesciences.org/articles/72466/figures#fig8video2

of phagosomes, whereas the subsequent fusion event can be measured by the level of accumulation of the mCherry signal inside the phagosomal lumen. In *rab-7(ok511)*(m⁻z⁻) mutant embryos produced by *rab-7(ok511)*(m⁺z⁻) mothers, robust enrichment of both the mCherry::LGG-1$^+$ and::LGG-2$^+$ puncta to phagosomal surfaces are prominent (*Figure 8G–H*), indicating that the recruitment of autophagosomes was normal. However, no mCherry signal was observed entering the phagosomal lumen (*Figure 8G–H*, *Figure 8—videos 1; 2*). We quantified the time between the moments a phagosome was just born and that when an obvious mCherry signal was observed inside the phagosomal lumen. In the wild-type embryos, this time is <30 min in 100% of the samples analyzed, and the median value is 14 and 18 min for LGG-1$^+$ and LGG-2$^+$ autophagosomes, respectively (*Figure 1(E and H)* and **8(I-J)**). In stark contrast, in *rab-7* mutants, the mCherry signal was not detected in the lumen inside any of the phagosomes for LGG-1 or LGG-2 reporters over a time span of 0–60 min after phagosome formation (*Figure 8I–J*). For over half of the samples, the observation period was extended beyond 90 min after phagosome formation, and still, no mCherry signal was observed in the phagosomal lumen within this period. These results strongly indicate that the function of RAB-7 is essential for autophagosome-phagosome fusion but not required for the recruitment of autophagosomes to phagosomal surfaces.

The HOPS complex acts as an effector for Rab7 (*Balderhaar and Ungermann, 2013*). In *C. elegans*, a null mutation in *vps-18*, which encodes a subunit of the HOPS complex, impairs phagosome maturation (*Xiao et al., 2009*). The lack of autophagosome-phagosome fusion observed in *rab-7* mutants led us to subsequently examine the *vps-18* null mutant embryos. We found that, like in *rab-7* mutants, the recruitment of autophagosomes to phagosomes appeared normal in *vps-18* mutants (*Figure 8G–H*). Also, like in *rab-7* mutants, in *vps-18* mutants, the fusion of LGG-1$^+$ and LGG-2$^+$ autophagosomes to phagosomes was severely defective (*Figure 8G–J*). However, whereas in *rab-7* mutants, the accumulation of the mCherry signal in the phagosomal lumen was blocked completely in all samples, in some *vps-18* mutant embryos, the entry of mCherry signal still occurred, *albeit* severely delayed (*Figure 8I–J*). In contrast, in other samples, the entry was blocked (*Figure 8I–J*), indicating a fusion defect that is less severe than that caused by the *rab-7* null mutation and suggesting the existence of the residual HOPS function in *vps-18* mutants. Together, our observations indicate that RAB-7 and the HOPS complex play a critical and specific role in driving the fusion between autophagosomes and phagosomes.

To evaluate the defect in phagosome degradation caused by the *rab-7(ok511)* null mutation, which abolishes the fusion of both lysosomes and autophagosomes to phagosomes, we measured the phagosome lifespan in *rab-7(ok511)*(m⁻z⁻) embryos using the phagosome lifespan analysis used in *Figure 6A–D*. Compared to that observed in the *atg-7*, *lgg-1*, and *lgg-2* mutant embryos, the phagosome lifespan is significantly longer in *rab-7* mutants. In *rab-7* mutants, 100% of phagosomes

last longer than 60 min, among which 71.4% last longer than 80 min and 57% last longer than 90 min (*Figure 6E and G*). The difference in severity in the phagosome degradation defects displayed by *rab-7* and autophagy mutants suggest that autophagosomes and lysosomes both contribute to phagosome degradation, and that autophagosomes provide a phagosome degradation activity in an additive and possibly independent manner to lysosomes.

## The CED-1 pathway drives the recruitment of autophagosomes to phagosomes

The signaling pathway led by the phagocytic receptor CED-1 plays essential roles in initiating the maturation of phagosomes containing apoptotic cells (*Yu et al., 2008*). The CED-1 pathway is known to drive the incorporation of early endosomes and lysosomes to phagosomes (*Yu et al., 2008*; *Yu et al., 2006*). Here, we further examined whether the loss-of-function mutations in members of the CED-1 pathway affect the incorporation of autophagosomes to phagosomes. CED-1 promotes the recognition, engulfment, and degradation of apoptotic cells (*Yu et al., 2008*). In *ced-1(e1735)* null mutant embryos, although the recognition and engulfment are delayed or blocked due to the loss of CED-1's engulfment activity, the majority of the cell corpses are eventually engulfed inside phagosomes (*Yu et al., 2008*). This is why in *ced-1* mutants, there are phagosomes that contain C1, C2, or C3 available for analysis of phagosome degradation. The partially penetrant recognition and engulfment defects observed in the *ced-1* mutants are due to the compensation of the activities by the *ced-5/–10/–12* and the *rab-35* pathways (*Reddien and Horvitz, 2004*; *Yu et al., 2006*; *Haley et al., 2018*). In *ced-1(e1735)* mutants expressing the mNG::LGG-1 or –2 reporters, we observed severe defects in the incorporation of autophagosomes into phagosomes. First of all, only a very dim mNG signal was observed inside the phagosomal lumen 50 min post phagosome formation, a time point well past the observed initiation time for autophagosomes/phagosome fusion in the wild-type condition (*Figure 9A, C, F and G*, *Figure 9—video 1*). Whereas in wild-type embryos, the median relative LGG-1 and –2 signal intensities are 6.3 and 5.3 at 50 min-post phagosome formation, respectively, in *ced-1* mutant embryos, the median values are merely 2.1 and 1.6, respectively (*Figure 9K–L*). Secondly, unlike in *rab-7* mutants, where autophagosomes were observed accumulating on phagosomal surfaces (*Figure 8G–J*), in *ced-1* mutants, very few LGG-1- or LGG-2-labeled puncta were observed on phagosomal surfaces (*Figure 9A and C*).

Further quantitative measurement of the samples presented in (*Figure 9A and C*) and 14 additional samples for each of the reporters confirmed that the mNG signal was not enriched on the surfaces of phagosomes, in contrast to wild-type embryos (*Figure 9H–L*). These results indicate a severe defect in the recruitment of autophagosomes to phagosomes. Unlike in the *atg* mutants that we have examined (*Figure 2—figure supplements 1–2*), in *ced-1* mutant embryos, normal numbers of LGG-1$^+$ or LGG-2$^+$ puncta were observed (*Figure 9—figure supplement 1*), indicating that the biogenesis of autophagosomes is normal. Thus the recruitment defect observed in *ced-1* mutants is not a consequence of the lack of autophagosomes; rather, it is likely a result of a defect in signaling between phagosomes and autophagosomes.

We further examined whether CED-6 and DYN-1, two other members of the CED-1 pathway, were also needed for the incorporation of autophagosomes into phagosomes. In the *ced-6(n2095)* and *dyn-1(n4039)* loss-of-function mutant embryos, the median relative LGG-1 and LGG-2 signal intensities in the center of phagosomes are much lower than that in wild-type samples, respectively, at 50 min-post phagosome formation mutants *Figure 9B,D,F,G, K–L* and *Figure 10A, D E,F,I,J*, although the defects are not as severe as in *ced-1* mutants *Figure 9A, C, F, G, K and L*. Further observation discovered that the LGG-1 or LGG-2-labeled puncta were rarely observed on the surfaces of phagosomes in these mutants *Figure 9B,D1,J* and *Figure 10A, F*, demonstrating severe defects in the recruitment of autophagosomes to phagosomal surfaces.

In *ced-1*, *ced-6*, and *dyn-1* mutants, the fusion between autophagosomes and phagosomes might also be defective. However, the severe recruitment defects resulted in the lack of LGG-labeled puncta on phagosomal surfaces, making it difficult to evaluate whether there are additional fusion defects and how severe the fusion defects are.

We also examined whether the pathway composed of the small GTPase CED-10 and its bipartite Guanine Nucleotide Exchange Factor (GEF) CED-5 and CED-12, which acts parallel to the CED-1 pathway in the engulfment of cell corpses, plays any role in promoting the incorporation

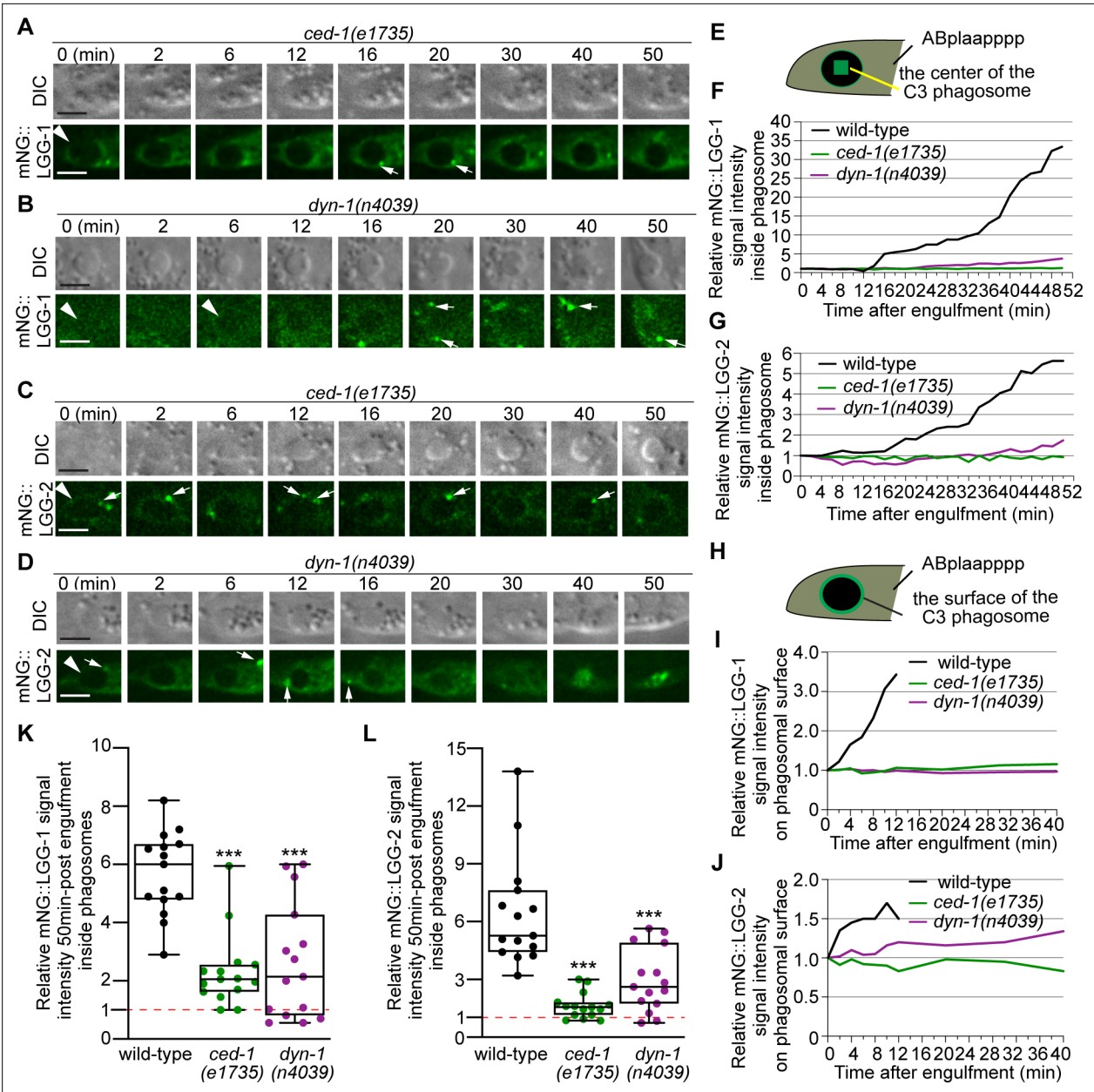

**Figure 9.** CED-1 and DYN-1 are essential for the incorporation of autophagosomes into phagosomes. (**A–D**) Time-lapse image series monitoring the presence or absence of puncta (white arrows) labeled with mNG::LGG-1 (**A–B**) or –2 (**C–D**) on C3 phagosomes (white arrowheads) and the subsequent entry of the mNG signal into the lumen in *ced-1* and *dyn-1* mutant embryos. DIC images mark the position of the cell corpse. '0 min' is the moment when phagosomes are just sealed. Scale bars are 2 μm. (**E**) A diagram illustrating that the relative mNG signal in the center of a phagosome is measured over time to create sub-figures (**F**) and (**G**). At time point t (time after '0 min'), the Relative Signal Intensity $_T$ = (Unit Intensity(phagosome center)$_T$ – Unit Intensity(background)$_T$) / (Unit Intensity(phagosome center)$_{T0}$- Unit Intensity (background)$_{T0}$). (**F–G**) The relative mNG::LGG-1 (**F**) or –2 (**G**) signal intensity in the center of a phagosome (Y-axis) over time in the 2 min interval (X-axis). '0 min' is the moment when pseudopods are sealed and a nascent phagosome forms. (**F**) The data for the wild-type, *ced-1(e1735),* and *dyn-1(n4039)* mutant embryos are from *Figure 1F* and 8(A-B), respectively. (**G**) The data for the wild-type, *ced-1(e1735),* and *dyn-1(n4039)* mutant embryos are from *Figure 1I* and 8(C-D), respectively. (*Figure 9—source data 1*) (**H**) A diagram illustrating that the relative mNG signal on the surface of a phagosome is measured over time to create sub-figures (**I**) and (**J**). At time point T (time after '0' min), the Relative signal intensity $_T$ = (Unit Intensity(phagosome surface (the green ring))$_T$ –Unit Intensity(background)$_T$) / (Unit Intensity(phagosome surface)$_{T0}$ - Unit Intensity (background)$_{T0}$). (**I–J**) The relative mNG::LGG-1 or –2 signal intensity on the surface of a phagosome (Y-axis) over time in the 2 min interval (X-axis). '0 min' indicates the moment when pseudopods are sealed and nascent phagosome forms. (**I**) The data for the wild-type, *ced-1(e1735),* and *dyn-1(n4039)* mutant embryos are from *Figure 1F* and 8(A-B), respectively. (**J**) The data for the wild-type, *ced-1(e1735),* and *dyn-1(n4039)* mutant embryos are from *Figure 1I* and 8(C-D), respectively. (*Figure 9—source data 2*). (**K–L**) Box-and-Whiskers plots of the relative mNG signal intensity measured in the center of phagosomes 50 min-post the formation of nascent C3 phagosomes from 15 each of wild-type, *ced-*

*Figure 9 continued on next page*

*Figure 9 continued*

1(e1735), and *dyn-1(n4039)* mutant embryos. Red dashed lines indicate the position of value 1, which represents no signal enrichment relative to the background signal. ***, p < 0.001, Student *t*-test of each mutant compared to the wild-type value. (*Figure 9—source data 3*).

The online version of this article includes the following video, source data, and figure supplement(s) for figure 9:

**Source data 1.** Singal intesity over time of mNG::LGG-1 and mNG::LGG-1 in *Figure 9F, G*.

**Source data 2.** Recruitment of mNG::LGG-1 and mNG::LGG-2 to the surface of phagosomes in *Figure 9I, J*.

**Source data 3.** Relative mNG::LGG-1 and mNG::LGG-2 signal intensity at 50min-post engulfment.

**Figure supplement 1.** The generation of autophagosomes is normal in *ced-1* mutants.

**Figure 9—video 1.** In *ced-1(e1735)* mutants, mNG::LGG-1-labeled vesicles fail to be recruited to the phagosomal surface.

https://elifesciences.org/articles/72466/figures#fig9video1

of autophagosomes into phagosomes. In the *ced-5(n1812)* null mutant and *ced-10(n1993)* loss-of-function mutant embryos, due to the presence of the parallel engulfment pathways (the CED-1 and RAB-35 pathways), some C1, C2, or C3 phagosomes eventually form, although they usually suffer a severe delay, allowing the analysis of phagosome degradation (*Yu et al., 2008*; *Yu et al., 2006*; *Haley et al., 2018*). In these two mutants, the accumulation of the LGG-1 and LGG-2 signals on the surfaces of the phagosomes and the subsequent accumulation of signals inside phagosomal lumen were normal both in the time course and in the levels of signal enrichment (*Figure 10B, C–E, G and H–J*), indicating that both the recruitment and fusion of autophagosomes to phagosomes are normal. We thus conclude that, unlike the CED-1/–6 /DYN-1 pathway, the CED-5/–10 pathway is not involved in regulating the incorporation of autophagosomes to phagosomes (*Figure 10K*).

## The incorporation of lysosomes into phagosomes is not significantly affected by the lack of autophagosome biogenesis

The incorporation of lysosomes into phagosomes is an essential force that drives the degradation of phagosomal contents (*Levin et al., 2016*). As a portion of the autophagosomes would fuse with lysosomes and become autolysosomes that retain lysosomal features (*Morishita and Mizushima, 2019*), the autolysosomes might contribute to phagosome degradation by depositing lysosomal luminal proteins to the phagosomal lumen. To test whether this is the case, we examined whether mutations that specifically impair autophagosome biogenesis would reduce the amount or speed of the incorporation of organelles with lysosomal features into phagosomes. *C. elegans* NUC-1 is an endonuclease belonging to the DNase II family and resides in the lysosomal lumen (*Wu et al., 2000*; *Guo et al., 2010*). Using a NUC-1::mCherry reporter expressed in engulfing cells as a lysosomal luminal marker, we quantified the level and rate of lysosome-phagosome fusion over time. In wild-type embryos, in phagosomes containing cell corpses C1, C2, and C3, we observed first the attachment of NUC-1::mCherry puncta on the surfaces of phagosomes and subsequently the accumulation of the mCherry signal inside the phagosomal lumen (*Figure 11A*). This dynamic process represents the recruitment and the subsequent fusion of lysosomal particles to phagosomes, which result in the delivery of NUC-1::mCherry into the phagosomal lumen. We next monitored NUC-1::mCherry inside the phagosomal lumen in *lgg-1(tm3489)*, *lgg-2(tm5755)*, and *atg-7(bp411)* embryos, which are defective in the biogenesis of autophagosomes. In these mutants, numerous mCherry::NUC-1 puncta are enriched on the surfaces of phagosomes C1, C2, and C3 and are subsequently fused to phagosomes (*Figure 11B–D*). As a result, the accumulation of the mCherry signal is observed in the phagosomal lumen in all samples like in wild-type embryos (*Figure 11B–E*). Quantitative analysis of 15 phagosomes for each genotype found that 60 min after phagosome formation, the average folds of increase of the luminal mCherry signal over the '0 min' time point in *lgg-1*, *lgg-2*, and *atg-7* mutants are not significantly different from that observed from wild-type samples (*Figure 11F*). In addition, the median value of the first time point when NUC-1::mCherry signal was detected inside the phagosomal lumen was not significantly different in all four genotypes (*Figure 11G*). These data strongly suggest that the incorporation of lysosomes into phagosomes in *lgg-1*, *lgg-2*, and *atg-7* mutants is as efficient as in wild-type embryos, at least within the detection range of our assay. Therefore, defects in the biogenesis of autophagosomes, which potentially would indirectly result in the lack of autolysosomes, do not appear to significantly affect the incorporation of lysosomes into phagosomes.

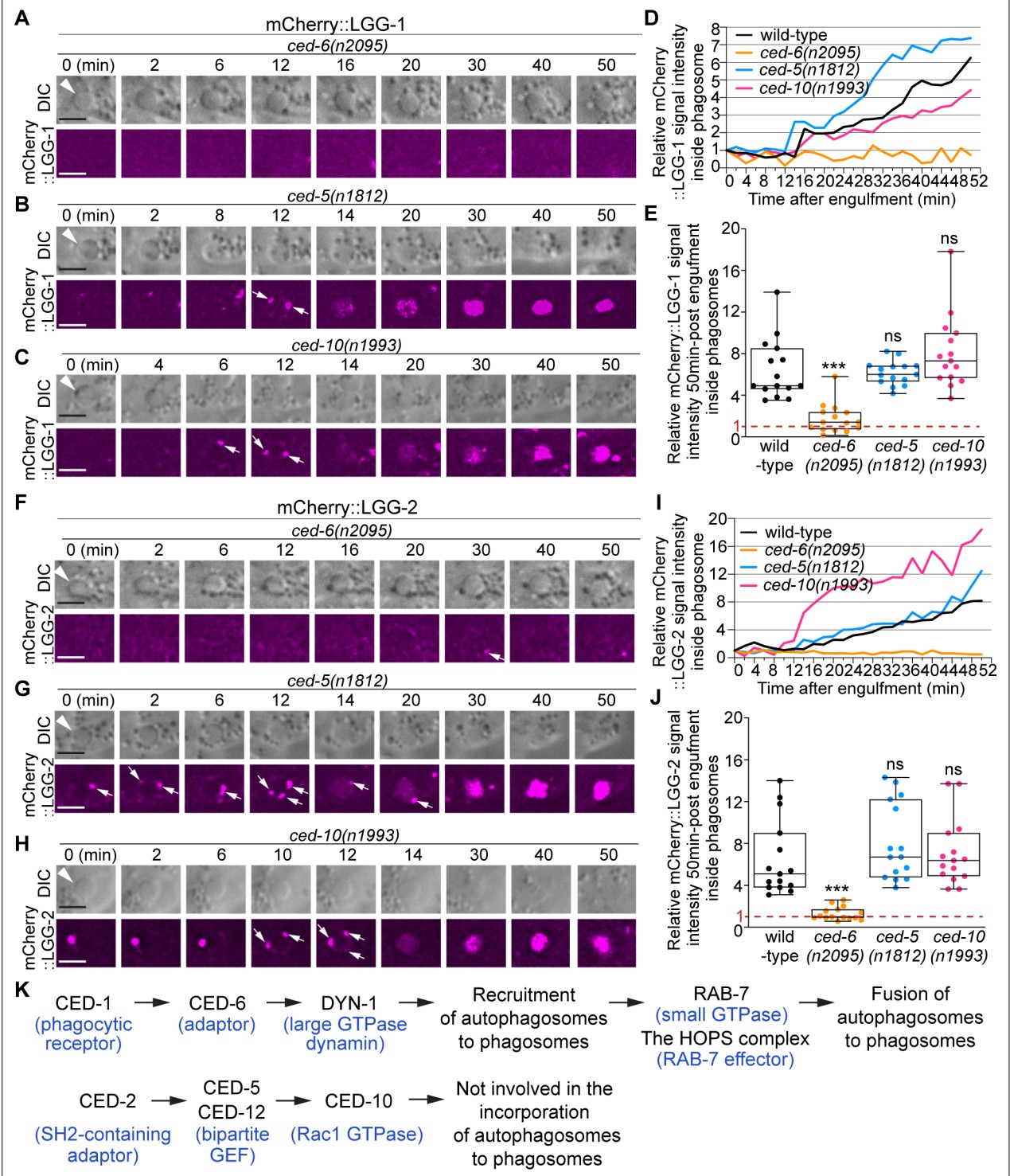

**Figure 10.** *ced-6*, but not *ced-5* or *ced-10*, is required for the incorporation of autophagosomes into phagosomes. (**A–C and F–H**) Time-lapse image series monitoring the presence or absence of puncta (white arrows) labeled with mCherry::LGG-1 (**A–C**) or –2 (**F–H**) on C3 phagosomes (white arrowheads) and the subsequent entry of the mCherry signal into the phagosomal lumen in *ced-6*, *ced-5*, and *ced-10* mutant embryos. DIC images mark the position of the cell corpse. '0 min' is the moment when a nascent phagosome just seals. Scale bars are 2 μm. (**D and I**) The relative mCherry::LGG-1 (**D**) or –2 (**I**) signal intensity in the center of a phagosome (Y-axis) over time (in the 2 min interval) (X-axis). "0 min" indicates the moment when a nascent phagosome just seals. (**D**) The data for the wild-type, *ced-6(n2095)*, *ced-5(n1812)*, and *ced-10(n1993)* mutant embryos are from *Figure 1E* and 9(A-C), respectively. (**I**) The data for the wild-type, *ced-6(n2095)*, *ced-5(n1812)*, and *ced-10(n1993)* mutant embryos are from *Figure 1H* and 9(F-H), respectively. (*Figure 10—source data 1*). (**E and J**) Box-and-Whiskers plots of the relative mCherry signal intensity measured in the center of phagosomes 50

*Figure 10 continued on next page*

Figure 10 continued

min-post the formation of nascent C3 phagosomes from 15 each of wild-type, *ced-6(n2095)*, *ced-5(n1812)*, and *ced-10(n1993)* mutant embryos. The red dashed lines indicate where value one is, representing no signal enrichment relative to the background signal. "***", p < 0.001; ns, not significant, Student *t*-test against the wild-type samples. (*Figure 10—source data 2*). (K) A diagram illustrating that between the two parallel pathways that regulate the clearance of apoptotic cells, only the CED-1 pathway, but not the other pathway, plays an essential role in promoting the incorporation of autophagosomes into phagosomes. Blue letters in parentheses are the names of the mammalian homolog of the corresponding *C. elegans* proteins.

The online version of this article includes the following source data for figure 10:

**Source data 1.** Relative signal intesity over time mCherry::LGG-1 and mCherry::LGG-2 in *Figure 10 D and I*.

**Source data 2.** Relative mCherry::LGG-1 and mCherry::LGG-2 signal intensity 50min-post engulfment in *Figure 10 E and F*.

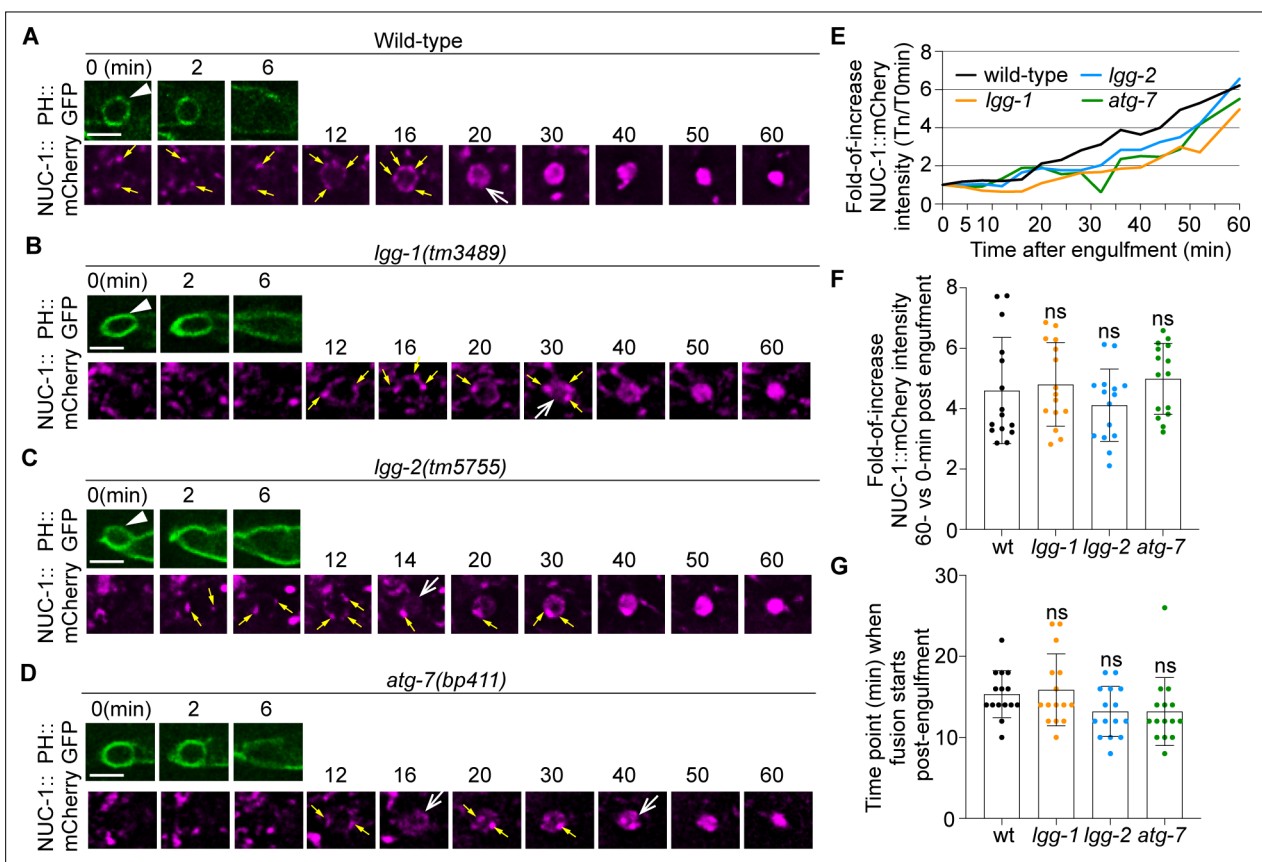

**Figure 11.** Defects in the formation of autophagosomes do not significantly affect the timing or efficiency of lysosomes incorporation into phagosomes. The time-lapse recording was conducted on phagosomes containing C1, C2, and C3 in wild-type and named mutant embryos carrying $P_{ced-1}nuc-1::mCherry$, the lysosomal lumen reporter, and $P_{ced-1}PH(PLC\gamma)::gfp$, the marker for extending pseudopods and nascent phagosomes. (A-D) Fluorescence time-lapse images of a C3 phagosome in each strain with the indicated genotype. '0 min' is the moment when a nascent phagosome (white arrowhead) just seals. Yellow arrows mark the lysosomal particles that are located on phagosomal surfaces. White open arrows mark the phagosomes with mCherry signals in the lumen. Scale bars are 2.5 µm. (E) The relative NUC-1::mCherry signal intensity in the center of a phagosome (Y-axis) over time (in the 2 min interval) (X-axis). '0 min' indicates the moment when a nascent phagosome is just sealed. Data are from *Figure 10 (A–D)*. (*Figure 11—source data 1*) (F) Bar graphs of the average fold-of-increase of the mCherry signal intensity at the center of phagosomal lumen 60 min-post the formation of nascent C3 phagosomes. Bars represent the mean, the error bars indicate standard deviation, and each dot represents a sample. 15 phagosomes of the indicated genotype were scored. Student *t*-test of each mutant compared to the wild-type value. ns, not significant. (*Figure 11—source data 2*) (G) Bar graphs of the average time when the NUC-1::mCherry signal is first detected inside the lumen of 15 C3 phagosomes in the indicated genotypes. Bars represent the mean, the error bars indicate standard deviation, and each dot represents a sample. Student *t*-test of each mutant compared to the wild-type value. ns, not significant. (*Figure 11—source data 2*).

The online version of this article includes the following source data for figure 11:

**Source data 1.** NUC-1::mCherry signal intensity over time in *Figure 11E*.

**Source data 2.** Time of fusion of NUC-1::mCherry and relative signal intesity at 60 min-post in *Figure 11 F and G*.

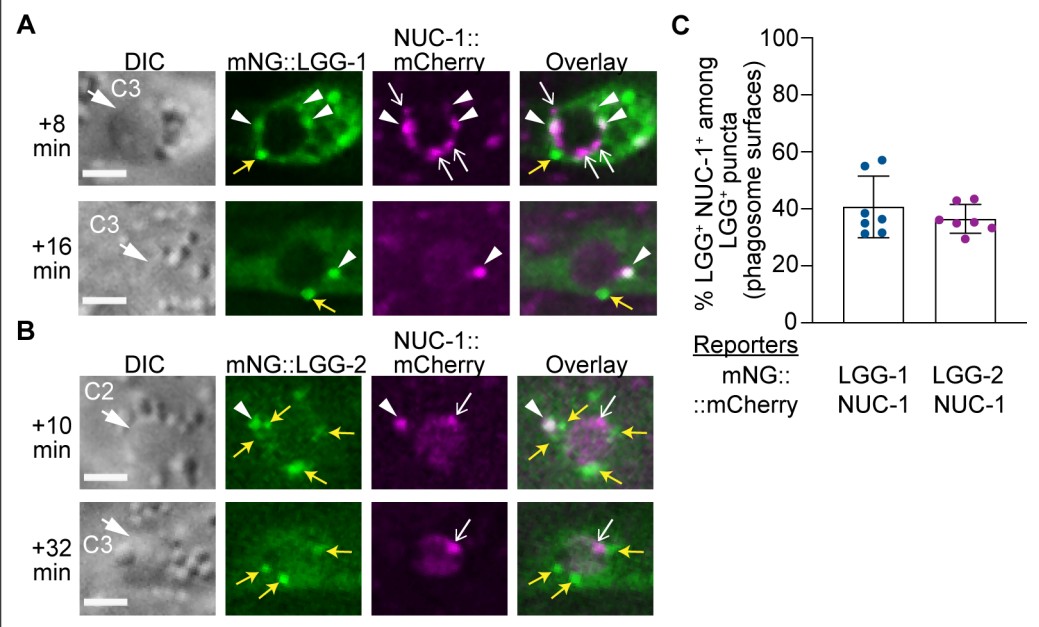

**Figure 12.** Visualizing the enrichment of lysosomes, autophagosomes, and autolysosomes on phagosomal surfaces. (**A–B**) DIC and fluorescence images of C2 and C3 phagosomes in wild-type embryos co-expressing P$_{ced-1}$ *nuc-1::mCherry* with P$_{ced-1}$ *mNG::lgg-1* or P$_{ced-1}$ *mNG::lgg*-2. Images were captured at the indicated time point after phagosome formation ('0 min'). DIC images mark the position of the cell corpse (white arrows). Yellow arrows, open white arrows, and white arrowheads label the GFP$^+$ mCherry$^-$ puncta (autophagosomes), GFP$^-$ mCherry$^+$ puncta (lysosomes), and GFP$^+$ mCherry$^+$ double-positive puncta (autolysosomes), respectively, on phagosomal surfaces. Scale bars are 2 μm. (**C**) Bar graph depicting the percentage of autolysosomes (GFP$^+$ mCherry$^+$ puncta) among LGG-1$^+$ or –2$^+$ puncta on the surface of phagosomes. Each sample (dot) represents the distribution of the aforementioned puncta collected from the C1, C2, and C3 phagosomes of one embryo. Seven embryos were scored. Bars and error bars represent mean and standard deviation values. (*Figure 12—source data 1*).

The online version of this article includes the following source data for figure 12:

**Source data 1.** Percentage distribution percentage of autolysosomes (GFP+ mCherry+ puncta) among LGG-1+ or -2+ puncta on the surface of C1, C2, and C3 phagosomes.

In the wild-type embryos that co-express the mNG::LGG-1 or –2 reporters with the NUC-1::mCherry reporter in engulfing cells, we observed that, in addition to the lysosomal particles that were mNG$^-$ but mCherry$^+$, there were two kinds of mNG$^+$ puncta that were recruited to phagosomal surfaces and subsequently fused to phagosomes. Those puncta that are not labeled with mCherry represent auto-phagosomes that are not fused to lysosomes; on the other hand, the mNG$^+$ mCherry$^+$ puncta represent autolysosomes (*Figure 12A–B*). Quantification of these two kinds of puncta on the surfaces of the C1, C2, and C3 phagosomes reveals that 40.6% and 36.5% of LGG-1$^+$ and –2$^+$ puncta are autoly-sosomes, respectively (*Figure 12C*). Whereas the LGG$^+$ NUC-1$^+$ autolysosomes are capable of contrib-uting both autophagosomal and lysosomal materials to phagosomes, the LGG$^+$ NUC-1$^-$ population of autophagosomes are likely to deliver certain autophagosome-specific material to phagosomes.

## Impairing autophagosome biogenesis results in moderate defects in phagosome acidification and the digestion of DNA from apoptotic cells

Acidification of the phagosomal lumen is a critical event for the degradation of phagosomal content. To examine whether defects in autophagosome biogenesis would affect the acidification of phago-somes, we developed an acidification reporter for the phagosomal lumen. This reporter, P$_{his-72}$ *his-72::gfp::mCherry*, expresses a HIS-72 (histone H3.3)::GFP::mCherry fusion protein in all cells (*Ooi et al., 2006*), including cells that undergo programmed cell death. We quantified the GFP/mCherry signal ratio in the phagosomal lumen over time and normalized it over the GFP/mCherry signal ratio at the '0 min' time point when a nascent phagosome just formed. We refer to this normalized value as an 'acidification index'. In a typical phagosome in a wild-type embryo, the acidification index reduces

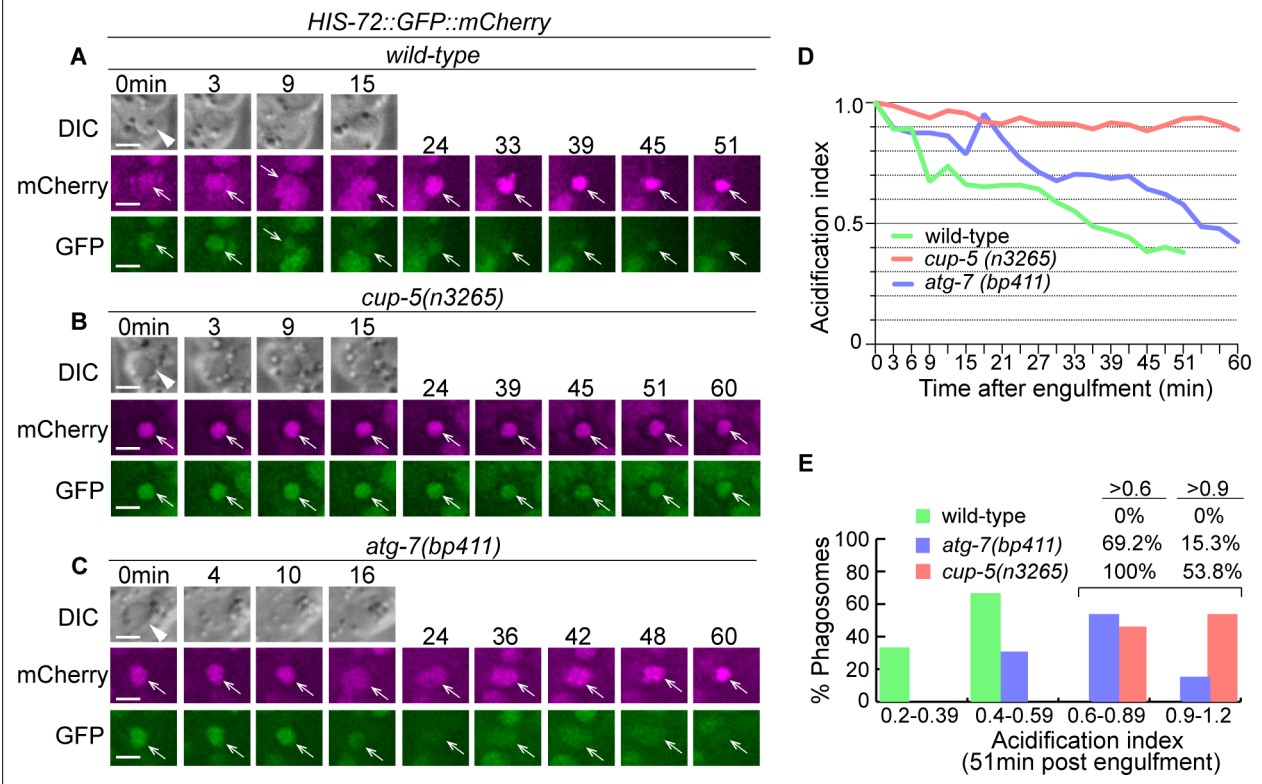

**Figure 13.** Inactivation of autophagy results in a modest phagosomal acidification defect. (**A–C**) Time-lapse imaging series of phagosomes (white arrowheads in DIC images) of wild-type, *cup-5*, and *atg-7* mutant embryos expressing P*his-72 his-72::gfp::mCherry*. Open white arrows depict the nuclei of engulfed cell corpses, labeled with both the GFP and mCherry markers. Reduction of the GFP signal intensity over time is indicative of phagosome acidification. '0 min' is when a phagosome is just sealed. Scale bars are 2 μm. (**D**) The acidification index curves of three phagosomes (Y-axis) over time (in the 3 min interval) (X-axis) in embryos with the labeled genotypes. '0 min' indicates the moment when a phagosome is just sealed. The data of the wild-type, *cup-5 (n3265)*, and *atg-7 (bp411)* are from A-C, respectively. (*Figure 13—source data 1*). (**E**) Histogram depicting the distribution of the acidification index measured at 51-min-post the formation of nascent phagosomes. In wild-type, *cup-5 (n3265)*, and *atg-7 (bp411)* mutant embryos, 6, 12, and 12 phagosomes were scored. (*Figure 13—source data 2*).

The online version of this article includes the following source data for figure 13:

**Source data 1.** The acidification index curves of phagosomes over time in *Figure 13D*.

**Source data 2.** Distribution of the acidification index measured at 51-min-post the formation of phagosomes in *Figure 13E*.

continuously from 1.0 (t = 0 min post engulfment) to 0.4 (t = 51 min) (*Figure 13A and D*). In addition, in 100% of the phagosomes (n = 6), the acidification index values at the 51 min time point are <0.6, among which in 35% of the samples, the index values are <0.4 (*Figure 13E*). On the contrary, in *cup-5(n3264)* mutant embryos, the acidification of the phagosomal lumen is severely defective (*Figure 13B, D and E*). The acidification index values of all the phagosomes (n = 13) at t = 51 min are >0.6, and in 53.8% of the phagosomes, the index values are >0.9, indicating a minimal reduction of the GFP signal. *cup-5* encodes a lysosomal TRP channel homologous to human Mucolipin IV (*Treusch et al., 2004*; *Campbell and Fares, 2010*). CUP-5 and its mammalian homologs play conserved and essential roles in supporting lysosomal biogenesis and functions (*Treusch et al., 2004*; *Campbell and Fares, 2010*). The severe acidification defect observed here emphasizes the essential role of lysosomes for phago-some degradation. Compared to the *cup-5* mutant embryos, in *atg-7(bp411)* null mutant embryos, we observed a moderate phagosome acidification defect (*Figure 13C, D and E*). The average acidifica-tion index values for wild-type, *atg-7(bp411)*, and *cup-5(n3264)* mutants are 0.443, 0.679, and 0.912, respectively, at t = 51 min. In *atg-7(bp411)* mutants, the acidification index values in 69.2% of the phagosomes at t = 51 min are >0.6, yet the values of only 15.3% phagosomes are >0.9. This moderate defect is distinct from that observed from the *cup-5* mutants.

The overall phagosome degradation efficiency in wild-type and autophagy mutant embryos has been measured by the reduction of phagosomal size over time (*Figure 6*). To further evaluate the

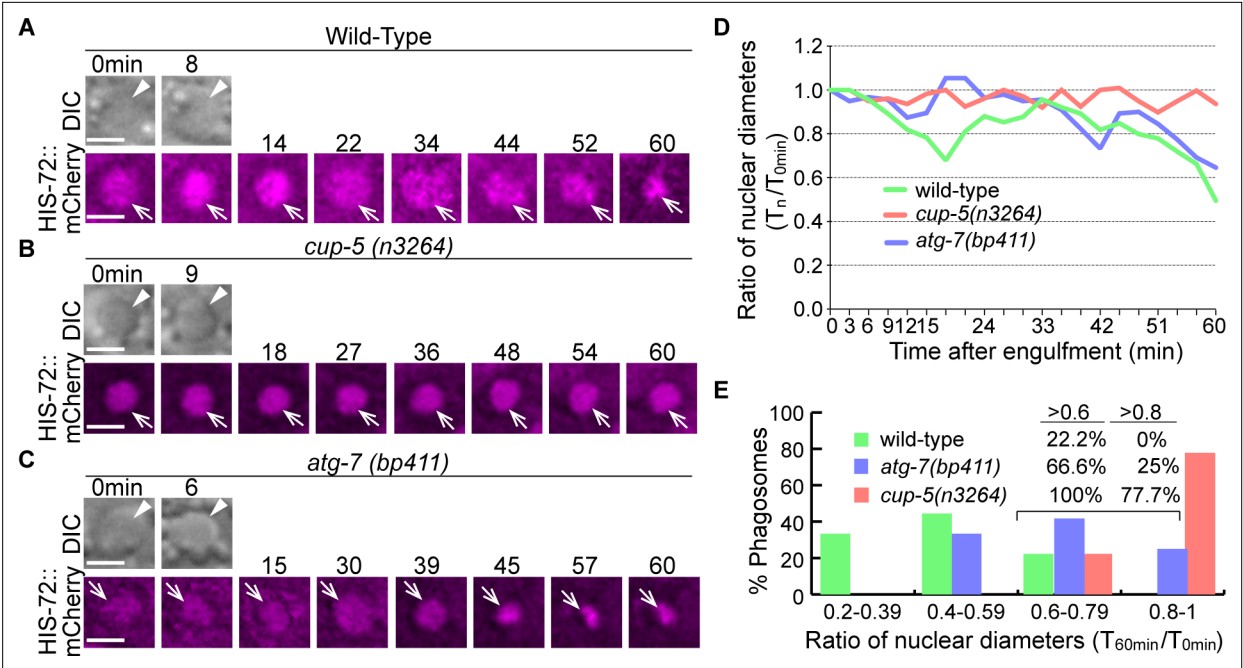

**Figure 14.** Inactivation of autophagy results in a modest delay of the degradation of apoptotic cell DNA. (**A–C**) Time-lapse imaging series monitoring the shrinkage of the apoptotic cell nucleus inside a C1, C2, or C3 phagosome (white arrowhead in DIC images) in three different strains expressing P$_{his-72}$ his-72::mCherry. Apoptotic cell nuclei are labeled with HIS-72::mCherry (open white arrows). '0 min' is when a phagosome is just sealed. Scale bars are 2 μm. (**D**) The ratio of the nuclear diameter of a phagosome (Y-axis) at labeled time points compared to that of the '0 min' diameter (in the 3 min interval) (X-axis). '0 min' indicates the moment when a phagosome is just sealed. The data of the wild-type, cup-5 (n3265), and atg-7 (bp411) are from *Figure 1(A–C)*, respectively. (*Figure 14—source data 1*). (**E**) Histogram depicting the distribution of the ratio of nuclear diameters measured at 60min-post phagosome formation. In wild-type, cup-5 (n3265), and atg-7 (bp411) mutant strains, 9, 9, and 12 engulfed apoptotic cells were scored. (*Figure 14—source data 2*).

The online version of this article includes the following source data for figure 14:

**Source data 1.** The ratio of the nuclear diameter curves of phagosomes over time in *Figure 14D*.

**Source data 2.** Distribution of the ratio of nuclear diameters measured at 60min-post phagosome formation in *Figure 14E*.

hydrolytic activities inside the phagosomal lumen in the autophagy mutant embryos, we measured the rate of the degradation of chromatin DNA in engulfed apoptotic cells. We found that, unlike a previous report that concluded that the NUC-1 endonuclease acts in cells undergoing programmed cell death to generate chromatin DNA fragments (*Wu et al., 2000*), NUC-1 acts in engulfing cells, in the phagosomal lumen to degrade chromatin DNA of engulfed apoptotic cells (Pickett J., Auld N., Lucas, L., Pena-Ramos O., and Zhou, Z., unpublished results). During the above study, we established HIS-72::mCherry as a reporter for chromatin DNA. Histone b H3.3, together with other histones and the DNA wrapping around them, form the nucleosome, the fundamental subunit of the chromatin. In Z-section images, inside the phagosomal lumen, the HIS-72::mCherry signal appears as a condensed red disc, outlining the nucleus of the engulfed apoptotic cell. We observed that in the wild-type embryos, the degradation of the apoptotic cell-chromatin DNA inside the phagosomal lumen allowed the reduction of the size of the mCherry$^+$ disc over time (*Figure 14A and D*). Within 60 min after the formation of a phagosome, the average diameter of the mCherry$^+$ disc reduces to 49% of the 0 min value (*Figure 14E*). On the contrary, in the nuc-1 mutant embryos, the diameter of the mCherry$^+$ disc remains the same over the entire period of time-lapse observation (> 80 min) (Pickett J., Auld N., Lucas, L., Pena-Ramos O., and Zhou, Z., unpublished observation), indicating the lack of degradation of chromatin DNA when NUC-1 activity is absent.

We observed that, as predicted for a mutant severely defective in lysosomal biogenesis and functions, in cup-5(n3264) mutant embryos, within 60 min post phagosome formation, the average diameter of the mCherry$^+$ disc remains at 89% of the 0 min-value (*Figure 14B and E*). In comparison, in atg-7(bp411) mutant embryos, that value is reduced to 66% of the 0 min-value; in addition, the

diameters of the mCherry[+] discs in 25% of the phagosomes are >80% of the 0 min-value. These results indicate that in *atg-7* mutants, there is a defect in the digestion of chromatin DNA, yet this defect is relatively moderate.

## Discussion

Autophagosomes play essential roles in cellular homeostasis by eliminating harmful protein aggregates and damaged organelles, and in stress response by converting intracellular organelles to nutrients during starvation (*Morishita and Mizushima, 2019*). We have identified a novel function of autophagosomes in facilitating the degradation of apoptotic cells through fusing to phagosomes. In *C. elegans* cells that engulf apoptotic cells, autophagosomes are recruited to the surfaces of phagosomes and subsequently fuse to the phagosomal membrane. Through this previously unknown interaction, autophagosomes facilitate the degradation of apoptotic cells inside phagosomes, presumably through providing certain materials to phagosomes. We have further identified a signaling pathway that promotes the recruitment and subsequent fusion of autophagosomes to phagosomes. Together, these findings reveal a novel mechanism through which the pathways that control autophagy and phagocytosis converge, underlining the importance of this mechanism in the degradation of apoptotic cells. We propose that this mechanism might be conserved in the metazoans and has a significant impact in the regulation of the immunological consequences of dying cells inside human bodies.

### canonical autophagosomes play an essential role in the degradation of apoptotic cells

Previously, *C. elegans* autophagy genes have been implicated in facilitating the clearance of apoptotic cells. Autophagy occurring in apoptotic cells was proposed to facilitate the exposure of phosphatidylserine, which is the 'eat me' signal that attracts engulfing cells to the surfaces of apoptotic cells in mice embryonic bodies and *C. elegans* (*Qu et al., 2007*; *Jenzer et al., 2019*). Multiple autophagy genes were reported to participate in the clearance of apoptotic cells in *C. elegans* engulfing cells (*Cheng et al., 2013*; *Huang et al., 2013*; *Li et al., 2012*; *Jenzer et al., 2019*). They were reported to work together with Class II PtdIns3 kinase PIKI-1 to facilitate the production of PtdIns(3)P on phagosomal surfaces (*Cheng et al., 2013*), to promote the recruitment of RAB-5 and RAB-7 to the phagosomal surfaces (*Li et al., 2012*), and to promote phagosome degradation (*Jenzer et al., 2019*). All the above findings suggest that the autophagy genes examined might have separate functions in the degradation of phagosomal contents unrelated to their canonical roles in autophagosome biogenesis. Our work reported here, on the other hand, presents a different discovery regarding how the autophagy machinery regulates phagosome degradation.

We found that the canonical double-membrane autophagosomes actively participate in the degradation of apoptotic cells inside phagosomes. In the time-lapse recording series, we first observed that vesicles labeled with LGG-1/or –2 reporters are recruited to phagosomal surfaces; subsequently, the LGG-tagged reporters enter the phagosomal lumen. We propose that the entry of the LGG reporters is a result of the fusion of the outer membrane of a double-membrane vesicle with the membrane of a phagosome based on two lines of evidence. First, the fusion between these two membranes would allow the release of the inner vesicle into the phagosomal lumen. Moreover, the RAB-7 GTPase and its effector—the HOPS complex, are well-known membrane tethering factors that promote the fusion between two intracellular membranes (*Stenmark, 2009*; *Balderhaar and Ungermann, 2013*). Their essential roles in supporting the entry of the LGG reporters into the phagosomal lumen that we have discovered strongly indicate that this event is a result of the fusion between the outer membrane of autophagosomes and phagosomes. Formally, other possible explanations of the entry of the LGG reporters exist, such as that phagosomes engulf the LGG-labeled vesicles. However, currently, there are no reported findings that phagosomes would engulf intracellular vesicles.

Using the immuno-electron microscopy technique, Manil-Segalen et al previously found that the LGG-1 and –2 are specifically enriched on double-membrane autophagosomes in *C. elegans* embryos (*Manil-Ségalen et al., 2014*). Five critical sets of evidence we report here further indicate that the LGG-1- and –2-labeled vesicles that are incorporated into phagosomes are double-membrane autophagosomes, not the single-membrane LAP vesicles. First, only when the vesicles are composed of double membranes and when the LGG-1/ or –2 reporter molecules label both the inside and outside

membranes, the entry of the reporters into phagosomal lumen is possible (*Figure 1K*). Conversely, if the LGG-1 or –2 labeled vesicles are of a single membrane, after vesicle-phagosome fusion, the reporter signal would remain on the phagosomal membrane, just like the membrane-bound lysosomal marker CTNS-1::mRFP (*Figure 1—figure supplement 1*). Second, these vesicles are also labeled with GFP-tagged ATG-9, an integral membrane protein inserted into the autophagosomal membranes. Third, we found that besides in embryos, in the gonad of adult hermaphrodites, many LGG-1-labeled vesicles are also attached to the surfaces of phagosomes that contain germ cell corpses in the *rab-7* mutant background in which membrane fusion is presumably blocked. Also, in the *rab-7* mutant adult hermaphrodites, thin-section transmission electron micrographs reveal multiple double-membrane vesicles that are in close contact with the phagosomal membranes. These vesicles resemble autophagosomes in morphology (*Kovacs et al., 2013*; *Zhang et al., 2015*). Fourth, the production of these vesicles relies on genes pivotal for autophagosome biogenesis, such as *atg-3*, *atg-7*, and *atg-9*. Last but not least, genes encoding the *C. elegans* homologs of mammalian ULK1, ATG13, and ATG14, which are dispensable for the generation of LAP vesicles in mammalian cells (*Green et al., 2016*), are required for the formation of the LGG-labeled vesicles discussed here and for the efficient clearance of apoptotic cells, further verifying that these vesicles are distinct from the single-membrane LAP vesicles. Together, these five sets of findings demonstrate that double-membrane autophagosomes, not single-membrane LAP vesicles, are being incorporated into phagosomes.

In *C. elegans*, LC3-dependent phagocytosis was reported to function in the clearance of the midbody, a structure that is essential for the completion of cytokinesis, and the second polar body generated during female meiosis (*Fazeli et al., 2018*; *Fazeli et al., 2016*). The clearance of midbodies and second polar body is considered independent of autophagosomes based on the observation that neither *unc-51* nor *epg-8* is required for the above processes (*Fazeli et al., 2018*; *Fazeli et al., 2016*). However, whether the LGG markers attached to phagosomes carrying midbodies or polar bodies represent double-membrane or single-membrane vesicles have not been investigated (*Fazeli et al., 2018*; *Fazeli et al., 2016*). Whether LAP vesicles exist in *C. elegans* cells still needs to be demonstrated. On the other hand, we found that in *atg-13* mutant embryos, in which the biogenesis of the LAP vesicles is not supposed to be affected, the LGG-labeled puncta largely disappeared, indicating that these puncta are primarily autophagosomes instead of LAP vesicles (*Figure 2—figure supplement 1 –2*). This observation does not support the existence of a substantial population of the LAP vesicles in *C. elegans* embryos, at least not in the cells in which the LGG reporters are expressed.

The observation of the LGG-1 or –2 signals inside the phagosomal lumen is made using the mCherry reporter. The mCherry protein is resistant to acidic pH (*Shaner et al., 2004*). Phagosome luminal pH value reduces from 5.5 to 6.0 to 4.5–5.5 after incorporating lysosomes (*Vieira et al., 2002*), leading to the inactivation of the fluorophore in the commonly used GFP reporter (*Tsien, 1998*). In fact, in *C. elegans* engulfing cells, GFP-tagged LGG-1 or –2, unlike the mCherry-tagged reporters, are only observed on the surfaces but not in the lumen of phagosomes, presumably due to the acidic environment of the phagosomal lumen (*Figure 1*). It is thus conceivable that in other experimental systems, a mCherry-tagged LC3 marker might disclose a previously overlooked fusion between autophagosomes and phagosomes.

Based on the evolutionary conservation of almost all molecular mechanisms, in particular the strong conservation of the known mechanisms that control phagocytosis and autophagy in metazoans (*Lu and Zhou, 2012*; *Zhang and Baehrecke, 2015*), we propose that in mammalian cells, besides LAP vesicles, canonical autophagosomes might also contribute to the degradation of apoptotic cells in phagosomes. The demonstration of an autophagosome-phagosome fusion event in *C. elegans*, a well-established model organism for studying cell-death-related events, opens a path to investigate whether the event and mechanism of autophagosome-phagosome fusion are evolutionarily conserved. In addition, the autophagosome-phagosome fusion might contribute to the degradation of other kinds of phagosomal cargos besides apoptotic cells.

## LGG-1 and LGG-2 define three subpopulations of autophagosomes that are incorporated into phagosomes

LGG-1 and –2 are close homologs that each labels autophagosomes. We have observed three separate classes of autophagosomes that are incorporated into phagosomes: LGG-1$^+$ LGG-2$^-$, LGG-1$^-$ LGG-2$^+$, or LGG-1$^+$ LGG-2$^+$. These subpopulations might represent autophagosomes at different stages of

maturation (*Manil-Ségalen et al., 2014*). Remarkably, we have found that the *lgg-1; lgg-2* double mutants display a further enhanced Ced phenotype than that displayed by each of the *lgg-1* or *lgg-2* single mutants. Together, these results indicate that autophagosomes at different stages of maturity all contribute to the degradation of apoptotic cells. These results are also consistent with our finding that the autophagosome-phagosome fusion is primarily independent of lysosome-phagosome fusion (*Figure 11*) (Also see the next section).

## What do autophagosomes contribute to the degradation of apoptotic cells

We have found that the blockage of the biogenesis of autophagosomes or the recruitment or fusion of autophagosomes to phagosomes results in a significant delay in the degradation of phagosomal content and, consequently, the persistent appearance of engulfed cell corpses. Among other possible mechanisms, the incorporation of autophagosomes into phagosomes presumably delivers certain substances to the lumen and/or membrane of phagosomes, substances that are important for the degradation of phagosomal contents. Here, we discuss several possible candidates and mechanisms supporting the roles of autophagosomes in phagosome degradation.

We have observed that the LGG$^+$ puncta inside engulfing cells are a collection of two subpopulations: autophagosomes that are already fused to lysosomes and become autolysosomes (LGG$^+$ NUC-1$^+$) and autophagosomes that are not fused to lysosomes (LGG$^+$ NUC-1$^-$) (*Figure 12*). The autolysosome populations, which occupied 40.7% and 36.5% of the LGG-1$^+$ and LGG-2$^+$ puncta observed on phagosomal surfaces, respectively, might contribute both autosomal and lysosomal materials to the phagosomes. In *atg-7* mutants, in which the biogenesis of autophagosomes is virtually blocked, the acidification of the phagosomal lumen and the degradation of the nuclear DNA of engulfed apoptotic cells are modestly defective (*Figures 13 and 14*). These defects might be partially attribute to the lack of autolysosomes that are incorporated into phagosomes. However, in cells that maintain a stable lysosome population, the lack of autophagosomes to fuse with lysosomes is not expected to influence the overall activities of lysosomes. In *atg-7*, *lgg-1*, and *lgg-2* mutant embryos, we found that the lack of autophagosomes does not significantly affect the efficiency of lysosome-phagosome fusion (*Figure 11*). This result could be due to that in an engulfing cell, the population of autolysosomes, comparing to that of lysosomes, is a rather minor population. It supports the notion that the autophagy pathway primarily regulates activities that are additive to the lysosomal activities for promoting phagosome degradation, although the assay might not be sensitive enough to detect a modest decrease. Moreover, in *rab-7(m⁻z⁻)* mutant embryos in which the fusion of both the lysosomes and the LGG$^+$ particles to phagosomes are blocked, the phagosome degradation defect is much more severe than in autophagosome biogenesis mutants. These results suggest that the collected population of LGG$^+$ vesicles contribute to phagosome degradation in a manner in parallel to lysosomes. In addition to the autolysosomal subpopulation, the LGG-1$^+$ NUC-1$^-$ (59.3% of the LGG$^+$ puncta) or LGG-2$^+$ NUC-1$^-$ (63.5% of the LGG2$^+$ puncta) subpopulations that fuse to phagosomes are likely to contribute certain unique, lysosome-independent activity(ies) to the phagosomes.

What could be the unique activities or mechanisms provided by the autophagosomes? Through fusion to phagosomes and releasing the inner vesicles to the phagosomal lumen, autophagosomes might deliver certain protein or lipid molecules to facilitate the degradation of phagosomal cargos. For example, the outer membrane of the autophagosomes might provide important signaling molecules to the phagosomal membrane. One such candidate is PtdIns(3)P, a membrane signaling molecule essential for many membrane trafficking events. On the phagosomal surfaces, PtdIns(3)P recruits PtdIns(3)P-binding proteins, including the sorting nexins SNX-1, SNX-6, and LST-4/SNX-9 and the HOPs complex, which subsequently drive multiple membrane remodeling events that promote phagosome maturation (*Lu et al., 2011a*; *Lu and Zhou, 2012*). In *C. elegans*, PtdIns(3)P is presented on phagosomal membrane over time in a two-peak pattern (*Lu et al., 2012*). Autophagosomes are coated with PtdIns(3)P (*Nakatogawa, 2020*). Judging by the timing of autophagosome-phagosome fusion, autophagosomal membranes are likely to contribute PtdIns(3)P for the rise of the second peak of PtdIns(3)P.

Alternatively, the autophagosome-phagosome fusion incorporates membrane materials to the phagosomal membrane and thus should increase the size of phagosomal membrane. Increasing the overall amount of phagosomal membrane might aid the degradation of the apoptotic cell by

facilitating the recruitment of lysosomes to the phagosomal surfaces. We previously reported that phagosomes extend transient lipid tubules to capture lysosomal particles in the cytoplasm and bring these particles back to the phagosomes for fusion (*Yu et al., 2008*; *Lu et al., 2011a*). The increased phagosomal membrane material might support the extension of these lipid tubules. A substantial body of future investigation is required to discover the mechanisms employed by autophagosomes that facilitate phagosome degradation.

## The CED-1 signaling pathway drives the incorporation of autophagosomes to phagosomes

Autophagosomes are incorporated into phagosomes in two sequential steps: (1) they are recruited to the surfaces of phagosomes, detected by the enrichment of punctated LGG-1 or –2 fluorescence reporters on phagosomal surfaces, and (2) they subsequently fuse to phagosomes, detected by the enrichment of the LGG-1 or –2 reporter signals inside the phagosomal lumen. RAB-7 and its effector, the HOPS complex, are known to act as tethering factors that facilitate the fusion of various intracellular organelles, including autophagosomes, to lysosomes (*Manil-Ségalen et al., 2014*; *Spang, 2016*). Closely related to this study, RAB-7 and VPS-18 are pivotal for the fusion between lysosomes and phagosomes and for the degradation of apoptotic cells inside phagosomes in *C. elegans* (*Yu et al., 2008*; *Xiao et al., 2009*; *Kinchen et al., 2008*). Here, we observed that 100% of the LGG$^+$ puncta attached to phagosomal surfaces were also labeled with RAB-7. We further discovered that RAB-7 and VPS-18 played essential and specific roles in the fusion but not for the recruitment of autophagosomes to phagosomes. Our finding adds a new pair of organelles that depend on the RAB-7/HOPS complex for fusion to each other. As RAB-7 is enriched on the surfaces of both phagosomes and autophagosomes (*Yu et al., 2008* and *Figure 8A-D*), and as the HOPS complexes in mammalian cells and *Drosophila* are known to interact with the SNARE complex, the membrane fusion machinery (*Jiang et al., 2014*; *Takáts et al., 2014*), we propose that the RAB-7/HOPS complex acts on the surfaces of phagosomes and autophagosomes to facilitate autophagosome-phagosome fusion via promoting the interaction between the SNARE complexes on phagosomes and autophagosomes (*Figure 8K*). Currently, the specific SNARE proteins that catalyze the autophagosome-phagosome fusion have not been identified.

The CED-1 signaling pathway, which initiates the maturation of phagosomes that bear apoptotic cells, is essential for the enrichment of GTP-bound RAB-7 to the surfaces of phagosomes (*Yu et al., 2008*). Here we have found that CED-1, CED-6, and DYN-1 drive the incorporation of autophagosomes to phagosomes, in addition to driving the incorporation of early endosomes and lysosomes previously discovered (*Yu et al., 2008*; *Yu et al., 2006*). In *ced-1* mutants, for example, the recruitment of autophagosomes to the surfaces of phagosomes is almost completely blocked. Due to the severe recruitment defect, whether the *ced-1* null mutation further impairs the fusion between autophagosomes and phagosomes cannot be readily visualized. However, since a *ced-1* null mutation impairs the recruitment of RAB-7 to and the production of PtdIns(3)P on phagosomal surfaces (*Yu et al., 2008*; *Lu et al., 2012*), and since the recruitment of the HOPS complex to the surfaces of intracellular organelles requires both RAB7 and PtdIns(3)P (*Balderhaar and Ungermann, 2013*; *Stroupe et al., 2006*; *Jeschke and Haas, 2018*), we predict that the CED-1 pathway would also control the RAB-7/HOPS complex-mediated autophagosome-phagosome fusion (*Figure 10*). Identifying the CED-1 signaling pathway as the driving force for the incorporation of autophagosomes to phagosomes helps to reveal the molecular mechanisms behind the crosstalk between autophagy and phagocytosis.

## Materials and methods

### Key resources table

| Reagent type (species) or resource | Designation | Source or reference | Identifiers | Additional information |
|---|---|---|---|---|
| Strain, strain background (*E. coli*) | OP50 | CGC | OP50 | |
| Strain, strain background (*C. elegans*) | N2 | CGC | Wild-type Bristol N2 | |

*Continued on next page*

*Continued*

| Reagent type (species) or resource | Designation | Source or reference | Identifiers | Additional information |
|---|---|---|---|---|
| Strain, strain background (*C. elegans*) | VC308 | CGC | *rab-7(ok511) /mIn1 II* | |
| Strain, strain background (*C. elegans*) | ZH0989 | *Yu et al., 2008* | *unc-76(e911) enIs36 [punc-76(+), Pced-1 ced-1::gfp, Pced-1 2xFYVE::mRFP] V* | *Figure 6* Available from the Zhou Lab |
| Strain, strain background (*C. elegans*) | ZH2059 | This study | *unc-76(e911)V; enEx979 [punc-76(+), Phis-72 HIS-72::GFP::mCherry]* | *Figure 13* Available from the Zhou Lab |
| Strain, strain background (*C. elegans*) | ZH2105 | This study | *cup-5(n3264) III; unc-76(e911) V; enEx979* | *Figure 13* Available from the Zhou Lab |
| Strain, strain background (*C. elegans*) | ZH2573 | This study | *lgg-2(tm5755) IV; unc-76(e911) V; enEx1223 [Pced-1mCherry::lgg-2, punc-76(+)]* | *Figure 5—figure supplement 1* Available from the Zhou Lab |
| Strain, strain background (*C. elegans*) | ZH2632 | This study | *lgg-2(tm5755) IV; unc-76(e911) V; enEx1267 [Punc-76(+), Pced-1::gfp::lgg2]* | *Figure 5—figure supplement 1* Available from the Zhou Lab |
| Strain, strain background (*C. elegans*) | ZH2715 | This study | *rab-7(ok511) II / mIn1 II; unc-76(e911) V; enEx1320 [punc-76(+), Pced-1mCherry::lgg-2]* | *Figure 8* Available from the Zhou Lab |
| Strain, strain background (*C. elegans*) | ZH2782 | This study | *rab-7 (ok511) II / mIn1 II; unc-76(e911) V; enEx1376 [punc-76(+), Pced-1 mCherry::lgg-1]* | *Figure 8* Available from the Zhou Lab |
| Strain, strain background (*C. elegans*) | ZH2831 | This study | *lgg-1(tm3489) II / mIn1 I; lgg-2(tm5755) IV* | *Figure 5* Available from the Zhou Lab |
| Strain, strain background (*C. elegans*) | ZH2835 | This study | *lgg-1(tm3489) II / mIn1 I; unc-76(e911) enIs36 V* | *Figure 6* Available from the Zhou Lab |
| Strain, strain background (*C. elegans*) | ZH2838 | This study | *lgg-1(tm3489) II / mIn1 II; unc-76(e911) V; enEx1428 [punc-76(+), Pced-1 gfp::lgg-1]* | *Figure 5—figure supplement 1* Available from the Zhou Lab |
| Strain, strain background (*C. elegans*) | ZH2841 | This study | *lgg-1(tm3489) II / mIn1 II; unc-76(e911) V; enEx1431 [punc-76(+), Pced-1 mCherry::lgg-1]* | *Figure 5—figure supplement 1* Available from the Zhou Lab |
| Strain, strain background (*C. elegans*) | ZH2875 | This study | *lgg-2(tm5755) IV; unc-76(e911) enIs36 V* | *Figure 6* Available from the Zhou Lab |
| Strain, strain background (*C. elegans*) | ZH2889 | This study | *unc-76(e911) V; enEx1459 [punc-76, Pced-1 PH::mrfp,Pced-1 mNG::lgg-2]* | *Figure 1* Available from the Zhou Lab |
| Strain, strain background (*C. elegans*) | ZH2903 | This study | *atg-9(bp564) him-5(e1490)V; lin-15AB(n765ts) X; enEx1468 [Plin-15(+), Pced-1 PH::mrfp, Pced-1 mNG::lgg-1]* | *Figure 3—figure supplement 1*, Available from the Zhou Lab |
| Strain, strain background (*C. elegans*) | ZH2907 | This study | *vps-18(tm1125) II; enIs80 [punc-76(+), Pced-1 ced-1::gfp, Pced-1mCherry::lgg-1] IV; unc-76(e911) V* | *Figure 8* Available from the Zhou Lab |
| Strain, strain background (*C. elegans*) | ZH2916 | This study | *ced-1(e1735) I; unc-76(e911) V; enEx1470 [Pced-1 mNG::lgg-1, punc-76(+), Pced-1 PH::mrfp]* | *Figure 9* Available from the Zhou Lab |
| Strain, strain background (*C. elegans*) | ZH2919 | This study | *enIs82 [unc-76(+), Pced-1 ced-1::gfp, Pced-1mCherry::lgg-1] II; unc-76(e911) V* | *Figure 1* Available from the Zhou Lab |
| Strain, strain background (*C. elegans*) | ZH2921 | This study | *unc-76(e911) enIs85 [punc-76(+), Pced-1 PH::mrfp, Pced-1mNG::lgg-2] V.* | *Figure 1* Available from the Zhou Lab |
| Strain, strain background (*C. elegans*) | ZH2922 | This study | *atg-9(bp564) him-5(e1490) V; lin-15AB(n765ts) X; enEx1472 [Plin-15(+), Pced-1 PH::mrfp, Pced-1 mNG::lgg-2]* | *Figure 3—figure supplement 1*, Available from the Zhou Lab |
| Strain, strain background (*C. elegans*) | ZH2929 | This study | *vps-18(tm1125) II; unc-76(e911) V; enIs83 [punc-76(+), Pced-1 ced-1::gfp, Pced-1mCherry::lgg-2] X* | *Figure 8* Available from the Zhou Lab |
| Strain, strain background (*C. elegans*) | ZH2934 | This study | *ced-1(e1735) I; unc-76(e911) enIs85 V* | *Figure 9* Available from the Zhou Lab |
| Strain, strain background (*C. elegans*) | ZH2950 | This study | *enIs82 II; atg-7(bp411) IV; unc-76(e911) V* | *Figure 2* Available from the Zhou Lab |
| Strain, strain background (*C. elegans*) | ZH2951 | This study | *atg-7(bp411) IV; enIs83 X* | *Figure 2* Available from the Zhou Lab |
| Strain, strain background (*C. elegans*) | ZH2952 | This study | *enIs82 II; atg-13(bp414) III* | *Figure 2* Available from the Zhou Lab |

*Continued on next page*

*Continued*

| Reagent type (species) or resource | Designation | Source or reference | Identifiers | Additional information |
|---|---|---|---|---|
| Strain, strain background (*C. elegans*) | ZH2953 | This study | *atg-13(bp414) III; enIs83 X* | *Figure 2* Available from the Zhou Lab |
| Strain, strain background (*C. elegans*) | ZH2954 | This study | *epg-8(bp251) I; enIs82 II; unc-76(e911) V* | *Figure 2* Available from the Zhou Lab |
| Strain, strain background (*C. elegans*) | ZH2955 | This study | *epg-8(bp251) I; unc-76(e911) him-5(1490) V; enIs83 X* | *Figure 2* Available from the Zhou Lab |
| Strain, strain background (*C. elegans*) | ZH2980 | This study | *atg-7(bp411) IV; unc-76(e911) enIs36 V* | *Figure 6* Available from the Zhou Lab |
| Strain, strain background (*C. elegans*) | ZH2992 | This study | *enIs87 [punc-76, Pced-1 PH::mrfp, Pced-1 mNG::lgg-1] I; unc-76(e911) V* | *Figure 1* Available from the Zhou Lab |
| Strain, strain background (*C. elegans*) | ZH2994 | This study | *enIs82 II; ced-6(n2095) III; unc-76(e911) V* | *Figure 10* Available from the Zhou Lab |
| Strain, strain background (*C. elegans*) | ZH2995 | This study | *ced-6(n2095) III; unc-76(e911) V; enIs83 X* | *Figure 10* Available from the Zhou Lab |
| Strain, strain background (*C. elegans*) | ZH3009 | This study | *ced-5(n1812) IV; unc-76(e911) V; enIs83 X* | *Figure 10* Available from the Zhou Lab |
| Strain, strain background (*C. elegans*) | ZH3010 | This study | *enIs82 II; ced-5(n1812) IV; unc-76(e911) V* | *Figure 10* Available from the Zhou Lab |
| Strain, strain background (*C. elegans*) | ZH3011 | This study | *enIs82 II; ced-10(n1993) IV; unc-76(e911) V* | *Figure 10* Available from the Zhou Lab |
| Strain, strain background (*C. elegans*) | ZH3012 | This study | *ced-10(n1993) IV; unc-76(e911) V; enIs83 X* | *Figure 10* Available from the Zhou Lab |
| Strain, strain background (*C. elegans*) | ZH3014 | This study | *enIs87 I; unc-76(e911) V; dyn-1(n4039) X; enEx21[Pdyn-1 dyn-1]* | *Figure 9* Available from the Zhou Lab |
| Strain, strain background (*C. elegans*) | ZH3015 | This study | *unc-76(e911) enIs85 V; dyn-1(n4039) X; enEx21* | *Figure 9* Available from the Zhou Lab |
| Strain, strain background (*C. elegans*) | ZH3485 | This study | *unc-76(e911)V; enEx1791 [punc-76(+), Pced-1 mNG::LGG-1, Pced nuc-1::mCherry]* | *Figure 12* Available from the Zhou Lab |
| Strain, strain background (*C. elegans*) | ZH3489 | This study | *rab-7 (ok511) II / mIn1 II; unc-76(e911) V; enEx1705 [punc-76(+), Pced-1 CED-1::GFP, Pced-1 2xFYVE::mRFP]* | *Figure 6* Available from the Zhou Lab |
| Strain, strain background (*C. elegans*) | ZH3492 | This study | *atg-7(bp411) IV; unc-76(e911) V, enEx979* | *Figure 13* Available from the Zhou Lab |

## Mutations, strains, and transgenic arrays

*C. elegans* strains were grown at 20 °C as previously described (*Wood, 1988*) unless indicated otherwise. The N2 Bristol strain was used as the wild-type control strain. Mutations are described in *Riddle et al., 1997* and by the Wormbase (http://www.wormbase.org) unless noted otherwise (Key Resources Table): LG1: *ced-1(e1735)*, *epg-8(bp251)*; LGII: *lgg-1(tm3489)*, *rab-7(ok511)*, *vps-18(tm1126)*; LGIII: *atg-13(bp414)*, *ced-6(n2095)*, *cup-5(n3264)*; LGIV: *atg-3(bp412)*, *atg-7(bp411)*, *ced-5(n1812)*, *ced-10(n1993)*, *lgg-2(tm5755* and *tm6474)*; LGV: *atg-9(bp564)*, *atg-18(gk378)*, *unc-76(e911)*, *unc-51(e369)*; LGX: *atg-2(bp576)*, *dyn-1(n4039)*. *dyn-1(n4039)* homozygous mutants, which are zygotic embryonic lethal, were maintained by an extrachromosomal array carrying a wild-type *dyn-1* gene and a co-expressed P$_{egl-13}$*gfp* marker (Key Resources Table) (*Yu et al., 2006*). *dyn-1(n4039)* homozygous embryos losing the rescuing transgene were identified as the embryos not carrying P$_{egl-13}$*gfp*. The *rab-7(ok511) and lgg-1(tm3489)* homozygous strains are both maternal-effect embryonic lethal, and the mIn1 balancer maintained each allele with an integrated pharyngeal GFP marker (Key Resources Table) (*Edgley et al., 2006*). To obtain *rab-7(ok511)* m⁻z⁻ homozygous embryos, GFP⁻ *rab-7(ok511)* homozygous hermaphrodites were isolated among the progeny of the strain VC308, and their progeny were collected as embryos. The same protocol was used to collect *lgg-1(tm3489)* m⁻z⁻ embryos from the strain GK738 (Key Resources Table). Double mutants between *lgg-1(tm3489)/mIn1* and *lgg-2(tm6474)* were generated by standard genetic crosses.

Extrachromosomal arrays were generated by the microinjection of plasmids with the co-injection marker p76-18B [*punc-76(+)*] into *unc-76(e911)* mutants (*Bloom and Horvitz, 1997*; *Jin et al., 1999*).

Non-Unc mutants were identified as transgenic animals. Integrated transgenic arrays were generated by gamma irradiation (*Jin et al., 1999*). Integrated arrays generated in this study are as follows (Key Resources Table): LGI: *enIs87[P$_{ced-1}$PH(hPLCγ)::mrfp* and *P$_{ced-1}$mNG::lgg-1]*; LGII: *enIs82[P$_{ced-1}$ ced-1::gfp* and *P$_{ced-1}$ mCherry::lgg-1]*; LGV: *enIs85[P$_{ced-1}$ PH(hPLCγ)::mrfp* and *P$_{ced-1}$ mNG::lgg-2]*; LGX: *enIs83[P$_{ced-1}$ ced-1::gfp* and *P$_{ced-1}$ mCherry::lgg-2]*.

Plasmid construction *lgg-1* and *lgg-2* cDNAs were PCR amplified from a mixed-stage *C. elegans* cDNA library (*Haley et al., 2018*). To generate the P$_{ced-1}$ *gfp::lgg*-1 and P$_{ced-1}$ *gfp::lgg-2* plasmids, *lgg-1* and *lgg-2* cDNAs were cloned into the XmaI and KpnI sites of plasmid pZZ956 (P$_{ced-1}$ *5'gfp*) (*Haley et al., 2018*). P$_{ced-1}$ *mCherry::lgg-1 or –2* were constructed by replacing the cDNA of *gfp* with that of mCherry (*Shaner et al., 2004*). P$_{ced-1}$ *mNG (mNeonGreen)::lgg-1 or –2* were constructed by replacing the cDNA of *gfp* with that of mNeonGreen (*Shaner et al., 2013*). To generate P$_{ced-1}$*gfp::lgg-1(G116A)* and P$_{ced-1}$*gfp::lgg-2(G130A),* the QuickChange Site-directed Mutagenesis Kit (Stratagene, La Jolla, CA) was used to introduce the above mutations into the constructs. To construct P$_{ced-1}$ *atg-9::mCherry, the atg-9a* open reading frame was PCR-amplified from a *C. elegans* mixed-stage cDNA library and inserted between P$_{ced-1}$ and *gfp* in pZZ829 (P$_{ced-1}$ *3'gfp*) (*Haley et al., 2018*). The *gfp* cDNA was then replaced by the *mCherry* cDNA (*Shaner et al., 2004*). All plasmids contain an *unc-54* 3' UTR.

## Transmission electron microscopy

In a mixed population of the strain VC308, isolate mid-L4 stage hermaphrodite larvae that were rab-7(ok511) and place them in a new plate. Forty-eight hours later, these worms, which are adults, were processed for transmission electron microscopic studies as previously described (*Yu et al., 2008*; *Yu et al., 2006*). Cross-sections of the gonadal region were generated. In 50 nm thin sections, germ cell corpses and the neighboring gonadal sheath cells were identified as previously described. A phagosome containing a germ-cell corpses was identified in serial 50-nm-thin sections that cover the entire length of each germ-cell corpse were analyzed by determining whether a germ-cell corpse is entirely inside the vacuole inside the gonadal sheath cells. Phagosomal surfaces were carefully monitored in a thin section electron micrograph for the presence of double-membrane vesicles.

## Quantification of the number of cell corpses using Nomarski DIC microscopy

Cell corpses display a highly refractive button-like morphology under Differential Interference Contrast (DIC) microscopy. An Axionplan two compound microscope (Carl Zeiss, Thornwood, NY) equipped with Nomarski DIC optics, an AxioCam digital camera, and AxioVision imaging software was used for DIC microscopy. Using a previously established protocol (*Lu et al., 2009*), we quantified the number of cell corpses in the head region at the 1.5-fold and 2-fold stage embryos, which are ~420 and ~460 min post-first cleavage.

## Fluorescence microscopy and time-lapse imaging

A DeltaVision Elite Deconvolution Imaging System (GE Healthcare, Inc) equipped with a DIC imaging apparatus and a Photometrics Coolsnap two digital camera was used to capture fluorescence, and DIC images Applied Precision SoftWoRx 5.5 software was utilized for deconvolving and analyzing the images (*Lu et al., 2009*). To observe the amount of autophagosomes in ventral hypodermal cells that express P$_{ced-1}$, fourteen serial Z-sections in 0.5 μm interval between adjacent optical sections, starting at the ventral surface of embryos at mid-embryonic stages, were collected. The 2D projection image of each Z-stack was generated and compared among different genetic backgrounds. To track fluorescence markers on pseudopods, on the surfaces of phagosomes, or inside phagosomal lumen during the clearance process of cell corpses C1, C2, and C3, embryos were monitored on their ventral surface starting at ~310 min post-first cleavage using an established time-lapse imaging protocol (*Lu et al., 2009*). Twelve to 16 serial Z-sections (at 0.5 μm intervals) were captured every 2 min, with recordings typically lasting between 60 and 180 min. Embryos that exhibited normal elongation and movement were considered developing properly. The moment engulfment starts is defined as when the extension of pseudopods around C1, C2, or C3 is first observed. The moment a nascent phagosome form is defined as when the pseudopods around a cell corpse join and make a full closure. The life span of a phagosome is defined as the time interval between the moments when the nascent phagosome forms and when the phagosome shrinks to one-half of its original diameter.

The time spans of the engulfment and degradation processes of cell corpses C1, C2, or C3 were measured as previously established (*Haley et al., 2018*). Briefly, a pseudopod marker, either CED-1::GFP or PH(PLCγ)::GFP, was monitored over time. The moment engulfment starts is defined as when the budding pseudopods around C1, C2, or C3 is first observed. The moment a nascent phagosome form is defined as the moment when the pseudopods around a cell corpse join and make a full closure. The period between the budding and the sealing of the pseudopods is the time span of engulfment. To measure phagosome duration, a co-expressed phagosomal surface marker mCherry::2xFYVE was used to track the diameter of the phagosome over time. The life span of a phagosome is defined as the time interval between the time points when a nascent phagosome was born and when the phagosome shrank to one-half of its original diameter.

To monitor the degradation process of the chromatin DNA of apoptotic cells inside phagosomes, we measured the size of the nucleus of engulfed C1, C2, or C3 over time by following the ubiquitously expressed $P_{his-72}$ *his-72::mCherry* reporter. Briefly, the '0 min' timepoint of phagosome formation was determined by the appearance of the 'bottom-like' structure under DIC optics. The nuclear diameter of each phagosome was measured as the diameter of the mCherry$^+$ disc inside the phagosome. The measurement is performed in regular time intervals until 60 min after the phagosome formation.

## Quantitative measurement of signal intensity

### Measuring the signal intensity inside the phagosomal lumen

In embryos expressing mCherry- or mNG-tagged LGG-1 or LGG-2, to measure the fluorescence signal intensity inside the phagosomal lumen over time, we identified the boundary of a phagosome and the '0 min' time point when a nascent phagosome was formed by observing the co-expressed marker for a nascent phagosome such as CED-1::GFP or PH(PLCγ)::GFP. At each time point, the total LGG-1 or –2 image intensity of a fixed area (4 × 4 pixels) at the center of a phagosome ($Int_{phagosome}$) was recorded (*Figure 1C*), so was the intensity of an area of the same size (4 × 4 pixels) outside the embryo as the background image intensity ($Int_{background}$). The Relative image intensity (RInt) at a particular time point (Tn) comparing to the start point (T0) is calculated as $RInt_{Tn} = (Int_{phagosome}-Int_{background})_{Tn} / (Int_{phagosome}-Int_{background})_{T0}$. The $RInt_{Tn}$ value of 1.0 indicates no entry of LGG-1- or LGG-2-labeled autophagosomes into the phagosomal lumen.

## Measuring the signal intensity on the surface of a phagosome

To measure the efficiency of recruitment of autophagosomes to phagosomes, we quantified the intensity of mCherry- or mNG-labeled LGG-1 or LGG-2 on the surfaces of phagosomes. First, we identified the boundary of a phagosome and the '0 min' time point when a nascent phagosome just formed by observing co-expressed CED-1::GFP or PH(PLCγ)::GFP. At a particular time point, Tn or T0, the surface of a phagosome was outlined by two closed polygons (*Figure 9H*). The total signal intensities, as well as the areas of the polygons, were recorded. The unit signal intensity of the 'donut-shape' area between the two polygons was calculated as follows:

Unit Intensity ($UI_{phagosome}$) = ($Intensity_{external\ polygon} - Intensity_{internal\ polygon}$)/($Area_{external\ polygon} - Area_{internal\ polygon}$). The Unit Background Intensity ($UI_{background}$) was measured from a polygon outside the embryo was calculated as follow: $UI_{background} = Intensity_{background}/Area_{background}$.

At the time point Tn, the relative signal intensity ($RInt_{Tn}$) = ($UI_{phagosome} - UI_{background}$)$_{Tn}$ / ($UI_{phagosome} - UI_{background}$)$_{T0}$. The $RInt_{Tn}$ value 1.0 indicates no enrichment of LGG-1- or LGG-2-labeled autophagosomes on phagosomal surfaces comparing to the '0 min' time point.

## Measuring the acidification index of a phagosome

In embryos expressing $P_{his-72}$ *his-72::gfp::mCherry*, we quantified the fluorescence intensity of the mCherry and GFP signals at the center of phagosomes in regular time intervals. The '0 min' timepoint of phagosome formation was determined by the appearance of the 'bottom-like' structure under DIC optics. At each time point, the total mCherry and GFP signal intensity of a fixed area (3 × 3 pixels) at the center of a phagosome ($Int_{mCherry}$, $Int_{GFP}$) was recorded. The acidification index ($A_{idx}$) at a particular time point (Tn) is defined as: $A_{idx} = (Int_{GFP}/Int_{mCherry})_{Tn} / (Int_{GFP}/Int_{mCherry})_{T0}$. The $A_{idx}$ value 1.0 indicates no phagosome acidification compared to the '0 min' value.

## Quantifying the percentage of autolysosomes among the LGG-1/or-2-labeled vesicles

In wild-type embryos co-expressing either mNG::LGG-1 or mNG:: LGG-2 together with NUC-1::mCherry (mCh), we counted the numbers of $mNG^+$ $mCherry^-$ puncta and the $mNG^+$ $mCherry^+$ puncta observed on the surfaces of phagosomes. For each set of reporters, C1, C2, and C3 phagosomes were quantified in each embryo, and the numbers of puncta were added. Seven embryos were scored (n = 7), each representing one sample. The percentage of autolysosomes ($\%_{AL}$) for each embryo was calculated as follow: $\%_{AL}$ = 100*((# of $mNG^+$ $mCh^+$ puncta)/ [(# of $mNG^+$ $mCh^+$) + (# of $mNG^+$ $mCh^-$)]). The average value and standard deviation were calculated and reported.

## Quantifying the percentage of LGG-labeled puncta that are also labeled with RAB-7

In wild-type embryos co-expressing $P_{ced-1}$ *mCherry-lgg-1* or *-lgg-2* together with $P_{ced-1}$ *gfp::rab-7*, we counted the numbers of $mCh^+$ $GFP^-$ and $mCh^+$ $GFP^+$ puncta in the engulfing cells for C1, C2, and C3 in two different locations: (1) on the surfaces of the phagosomes, and (2) in the cytoplasm. For each set of co-expressed reporters, 9 engulfing cells were scored. $N_{mCh+}$ (the total number of $mCh^+$ puncta) = # of $mCh^+$ $GFP^-$ puncta + # of $mCherry^+$ $GFP^+$ puncta. % $mCh^+$ $GFP^+$ = 100 * (# $mCh+$ $GFP$ + puncta)/ $N_{mCh+}$. The average values and standard deviation values were calculated and reported.

## Acknowledgements

We thank Ying Wang and Lidan Gao for technical support and Ryan Haley for helpful comments. We thank H Robert Horvitz for advice and the use of the electron microscope. We thank Erika Hartwieg for the technical support for the TEM study. We thank the *C. elegans* Genetics Center (CGC), funded by the NIH Office of Research Infrastructure Programs (P40 OD010440), for providing some strains. We also thank the National BioResource Project of Japan and Dr. Shohei Mitani for providing some strains. NIH R01GM067848 supports this work.

## Additional information

### Funding

| Funder | Grant reference number | Author |
| --- | --- | --- |
| NIH | R01GM067848 | Zheng Zhou |

The funders had no role in study design, data collection and interpretation, or the decision to submit the work for publication.

### Author contributions

Omar Peña-Ramos, Conceptualization, Data curation, Formal analysis, Investigation, Methodology, Project administration, Resources, Supervision, Validation, Visualization, Writing – original draft, Writing – review and editing; Lucia Chiao, Formal analysis, Investigation, Methodology, Resources, Writing – review and editing; Xianghua Liu, Data curation, Investigation, Methodology, Resources; Xiaomeng Yu, Data curation, Investigation, Methodology, Resources, Visualization; Tianyou Yao, Henry He, Investigation, Methodology, Resources; Zheng Zhou, Conceptualization, Data curation, Formal analysis, Funding acquisition, Investigation, Methodology, Project administration, Resources, Supervision, Validation, Writing – original draft, Writing – review and editing

### Author ORCIDs

Omar Peña-Ramos http://orcid.org/0000-0003-1231-2602
Zheng Zhou http://orcid.org/0000-0003-2585-0418

### Decision letter and Author response

Decision letter https://doi.org/10.7554/eLife.72466.sa1
Author response https://doi.org/10.7554/eLife.72466.sa2

## Additional files

### Supplementary files
• Transparent reporting form

### Data availability
We provide the numerical data for the figures: 1, 2, 5, 6, 7, 8, 9, 10, 11, 12, 13, 14 , and Figure 5—figure supplement 1 in associated Source Data Files.

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
