## [Editor Report]

Peña-Ramos et al., describe a novel interaction between phagosomes and autophagosomes in the degradation of apoptotic cell corpses. Using time-lapse fluorescence microscopy to measure dynamic changes in phagosomes, as well as electron microscopy, the authors follow cell corpse degradation in specific phagocytic cells of developing *C. elegans* embryos. They find that autophagosomes attach to phagosomes and promote their degradation by controlling acidification. The study uncovers a novel function of autophagosomes, and presents a new paradigm for how cell corpses are degraded.

---

## [Decision Letter]

**Decision letter after peer review:**

Thank you for submitting your article "Autophagosomes fuse to phagosomes and facilitate the degradation of apoptotic cells in *Caenorhabditis elegans*" for consideration by *eLife*. Your article has been reviewed by 3 peer reviewers, and the evaluation has been overseen by Shai Shaham, as the Reviewing Editor, and Piali Sengupta as the Senior Editor. The following individuals involved in review of your submission have agreed to reveal their identity: Barbara Conradt (Reviewer #2); Marc Freeman (Reviewer #3).

The authors developed a time-lapse recording protocol that allows them to observe formation and degradation of phagosomes in specific phagocytic cells of developing *C. elegans* embryos. Using this protocol in combination with specific reporters, they find that LC3-positive vesicles fuse with phagosomes and that these vesicles are double-membraned. They provide evidence that these LC3-positive vesicles are autophagosomes and demonstrate that preventing fusion results in a general engulfment defect due to a defect in the degradation of phagosomal content. RAB-7 and the HOPS complex are necessary and the CED-1, CED-6, DYN-1 pathway are also necessary for recruiting autophagosomes to phagosomes. Preventing autophagosome-phagosome fusion does not affect lysosome-phagosome fusion, ruling out indirect effects. This is a very rigorous and convincing study that will have significant impact on understanding cell corpse engulfment and degradation. It also uncovers a novel function of autophagosomes. There are no major experimental weaknesses, however, some of the interpretations are not fully supported and additional experiments could significantly solidify the underlying hypothesis.

Essential revisions:

As you will see below the reviewers have made a number of important and useful comments on the manuscript, and you should address these with either new experiments or text revisions. Three issues about which new experiments may be revealing are:

1) Organelle identity: is it possible to perform EM or use previous EM imaging to demonstrate the existence of organelles that have fused with autophagosomes?

2) Mechanism: Can the authors use pH-sensitive reporters to understand the role of autophagosomes in phagosome maturation? Are there differences in pH of phagosomes in autophagosome mutants? Is it possible to assay hydrolytic enzyme activity in phagosomes? If so, are there differences in enzyme activity between WT and mutants lacking autophagosomes?

3) Genetic pathway: What are the effects of double mutants blocking both lysosome-phagosome fusion and autophagosome-phagosome fusion?

In the event that there are major technical limitations to performing these studies, please explain these.

*Reviewer #1 (Recommendations for the authors):*

1. Rather than using red and green for fluorescence images, I suggest using magenta and green, especially in those figures were the two colors our overlaid.

2. The authors use a visual assay and genetics to provide evidence that the LC3-positive vesicles that fuse with phagosomes are double-membrane vesicles and autophagosomes. Is there any evidence for double-membrane vesicles from EM images that the authors may have or are aware of in the literature?

3. Figure 6. Could the authors provide a quantification of LGG-1, LGG-2 and double positive vesicles? And could RAB-7 specifically be on LGG-1- and LGG-2 positive vesicles (Figure 7)?

4. ced-1 is required first for phagosome formation and subsequently for phagosome degradation. How can the authors look at autophagosome-phagosome fusion in ced-1 null mutants? Is the phagosome formation defect only partially penetrant?

5. Do the authors know whether blocking lysosome-phagosome fusion AND autophagosome-phagosome fusion results in a more severe engulfment defect compared to blocking only one of these fusion events? Basically, is there an additive or synergistic effect?

*Reviewer #2 (Recommendations for the authors):*

My main concern is really intepretation and the model. Do autophagosomes fuse with phagosomes and contribute to corpse digestion as suggested?

EM showing double-membrane nature of these autophagosomes seems like it is missing and might be needed. The arguments are all based on marker topology, which is reasonable, but a phagosome is a dynamic thing. For instance, as phagosomes digest contents like corpses, they get smaller. Does that membrane get internalized into the phagosome? If so, how does that affect the interpretation of the markers going inside the phagosome at later time points?

What if this is a mechanism that allows phagosomes to grow their lipid membranes, rather than fusion to digest what's in the autophagosome? Atg2/9 drive the transfer of lipids from the ER to autophagosomes, that's how autophagosomes grow. Maybe autophagosomes in this situation are intermediates that serve as important lipid sources for phagosomes? Is there an autophagosome target (inside) or component not in phagosomes that one could track to show actual deposition and subsequent degradation? How else can these possibilities be discriminated?

The argument that they don't fuse is reasonable, but what if autophagosomes themselves are engulfed completely into the phagosome? They show a lysosomal marker doesn't do that as an argument, which is fine, but that really only says lysosomes are acting as they normally do in this context and that their assays would detect that. What if autophagosomes are different? In the next section on ATG7 they assert this (internal fluorescence) as a marker for insertion of the internal part of a double-membrane vesicle into a phagosome, but it may or may not be.

What if LAPs in worms don't need ulk1/atg13/atg14, and they are single-membrane LAPs that are made differently? The data suggest these vesicles are double-membrane, but there are alternate interpretations (e.g. single vesicles that go in, and the stable reporters are showing that). How else are these different, or the same as LAPs in mammals?

The idea here is that the autophagosomes are needed for digestion of corpses, but what if corpses and stuff in autophagosomes are co-degraded in phagosomes? In this model, the autophagosomes would be additional material digested in what started as a phagosome. If that is blocked with mutations affecting autophagy, it could be that the delays in digestion of corpses observed are due to a checkpoint that is relevant to autophagosome fusion/digestion, rather than there being a direct role for autophagosomes in cell corpse degradation.

I think the authors have done a great job convincing me that autophagosomes are important, and that all the genes they describe are needed, but I am not fully convinced of the model they have landed on. It seems like there are many other possibilities. I think those need to be discussed.

*Reviewer #3 (Recommendations for the authors):*

The authors should try to assess the pH and hydrolytic activity inside the phagosomes that matured in the presence or absence of autophagosomes to more convincingly demonstrate that lysosomal fusion occurred normally and that, importantly, a new, heretofore unappreciated factor is required for proper degradation of the contents.

They should also establish the nature of the fusing structures: are they truly double-membraned autophagosomes or have they undergone fusion with lysosomes (i.e. become autolysosomes)?

---

## [Author Response]

Essential revisions:As you will see below the reviewers have made a number of important and useful comments on the manuscript, and you should address these with either new experiments or text revisions. Three issues about which new experiments may be revealing are:1) Organelle identity: is it possible to perform EM or use previous EM imaging to demonstrate the existence of organelles that have fused with autophagosomes?

Previously, using immuno-electron microscopy, Manil-Segalen et al., [1] have shown that in *C. elegans* embryos, the GFP::LGG-1 and GFP::LGG-2 reporters each labeled those puncta are indeed double-membrane vesicles. This result established the legitimacy of using the LGG-1 and LGG-2 as reporters for canonical autophagosomes.

During the revision period, to address the above request, we characterized the phagosomes that contain germ cell corpses and the vesicles that are in close proximity to the phagosomal surfaces in the adult hermaphrodite gonads using transmission electron microscopy (TEM). Currently we do not have the expertise or facility to conduct transmission EM procedure on embryos, the sectioning of which is more technically challenging than adult worms. On the other hand, we had previous experience performing TEM analysis in the adult gonads, which host many apoptotic germ cells. Furthermore, accumulative studies in the literature have indicated many common features and common mechanisms that are employed during the clearance of the apoptotic cells in embryos and in the adult hermaphrodite gonad. We thus decided to do the TEM analysis in the adult gonads.

In wild-type adult hermaphrodites, very few germ cell corpses remain in the gonad due to the efficient engulfment and phagosome degradation processes. In contrast, in rab-7 loss-of-function mutants, a large number of germ cell corpses remain in the gonad due to the lack of phagosome degradation [2]. In this study, we found that in rab-7(-) embryos, LGG-labeled vesicles are normally recruited to phagosomes yet fail to fuse with phagosomes, resulting in the accumulation of LGG+ vesicles on phagosomal surfaces. We observed that in rab-7(-) adult gonad, many LGG-1 puncta were attached to phagosomal membranes, suggesting that the fusion between autophagosomes and phagosomes might be blocked, like in rab-7(-) embryos (new Figure 4A). In this case, in rab-7(-) adult gonads, the chance of catching LGG-1-labeled puncta on phagosomal surfaces in 50-nm thin sections should be much increased. We thus decided to perform TEM analysis in the adult gonads of rab-7(-) hermaphrodites.

We analyzed a collection of thin section (50 nm sections) transmission electron micrographs of the gonads in rab-7(-) mutant adult hermaphrodites. We observed multiple examples of double-membrane vesicles in close contact with phagosomal surfaces. These vesicles resemble the morphology of the autophagosomes published in the literature (double membrane, with irregular space between the two lipid bilayers, the diameter is between 100 and 1000 nm, the content containing ribosomes). Our observation of double-membrane vesicles on phagosomal surfaces is consistent with this conclusion that autophagosomes are recruited to phagosomal surfaces. We reported this result in the new Figure 4, and in Results, in a subsection entitled, “The autophagosomes-phagosome interaction is a general phenomenon observed in embryos and the adult gonad”.

2) Mechanism: Can the authors use pH-sensitive reporters to understand the role of autophagosomes in phagosome maturation? Are there differences in pH of phagosomes in autophagosome mutants? Is it possible to assay hydrolytic enzyme activity in phagosomes?

To address the above request, we developed an acidification reporter for the phagosomal lumen and used it to measure the degree of acidification of the phagosomal lumen in wild-type and *atg-7* mutant embryos. As a negative control, we also did the measurement in *cup-5* mutants in which lysosomal biogenesis and functions are severely defective. This reporter, P*_his-72_ his-72::gfp::mCherry*, expresses a HIS-72 (histone H3.3)::GFP::mCherry fusion protein in all cells, including cells that undergo apoptosis. Inside the phagosomal lumen, in the nucleus of a cell corpse, the GFP signal gradually diminishes due to its sensitivity to the acidic pH whereas mCherry is resistant to acidic pH and its signal persists over time. The ratio between GFP and mCherry signal has been commonly used to represent the degree of acidification in a number of studies. We quantified the GFP/mCherry signal ratio in the phagosomal lumen over time and normalized it against the “0 min” time point when a nascent phagosome had just formed. We refer to this normalized value as an “acidification index”. As shown in the new Figure 13, in wild-type embryos, the acidification index of the phagosomal lumen is reduced continuously after phagosome formation. In contrast, in *cup-5* mutants, the reduction of the acidification index is minimal, demonstrating that without the functional lysosomes, phagosome acidification is blocked or severely defective. In *atg-7* mutants, a moderate acidification defect is observed. This moderate defect likely contributes to the phagosome degradation defect observed in *atg-7* mutants, which is severely defective in the formation of autophagosomes. Autophagosomes thus contribute to the acidification of the phagosomal lumen. We have described these results in “Results”, in a section entitled “Impairing autophagosomes biogenesis results in moderate defects in phagosome acidification and the digestion of apoptotic-cell DNA”.

If so, are there differences in enzyme activity between WT and mutants lacking autophagosomes?

During the revision period, we have developed one particular assay for the hydrolytic enzyme activity – the endonuclease activity inside the phagosomal lumen. This activity, which is primarily carried out by a lysosomal DNaseII named NUC-1, catalyzes the degradation of the chromatin DNA of apoptotic cells inside phagosomes. We have established a HIS-72::mCherry fusion protein as a reporter for chromatin DNA degradation. HIS-72, a histone B H3.3, is a subunit of the nucleosome, the building unit of the chromatin. HIS-72::mCherry labels the nucleus of an apoptotic cell inside a phagosome. We found that the degradation of chromatin DNA of the engulfed cell corpse mediated by NUC-1 caused the shrinkage of the cell corpse nucleus over time in the wild-type background (Figure 14). In contrast, in *cup-5* mutant embryos, which are severely defective in the lysosomal biogenesis and functions, minimum DNA degradation activity was detected, consistent with the notion that the DNA degradation activity in phagosomes is primarily provided by lysosomes (Figure 14). In comparison, in *atg-7* mutant embryos, a relatively moderate defect in the DNA degradation efficiency was observed (Figure 14). These new results are reported in “Results”, in a section entitled “Impairing autophagosomes biogenesis results in moderate defects in phagosome acidification and the digestion of apoptotic-cell DNA”, and in a new Figure 14.

It is unclear whether the modest defects in phagosomal acidification and DNA degradation displayed by the *atg-7* null mutants, in which autophagosomes biogenesis is primarily blocked, are sufficient to cause the intermediate delay in phagosome degradation. Less than 41% of the autophagosomes that fuse to phagosomes are autolysosomes that have the features of both autophagosomes and lysosomes. It is possible that autolysosomes bring some lysosomal activities to the phagosomal lumen and membrane. However, given that ~60% of the autophagosomes that are fused to phagosomes are not autolysosomes, we propose that autophagosomes additionally provide unique, lysosomal-independent substances and/or activities to phagosomes for the degradation of phagosomal cargos. Further investigation is required to identify such substances and/or activities.

3) Genetic pathway: What are the effects of double mutants blocking both lysosome-phagosome fusion and autophagosome-phagosome fusion?

Our previous findings [2] and the discovery made in this manuscript indicate that RAB-7 activity is required for both the lysosome-phagosome and autophagosomephagosome fusions. In rab-7(ok511)(m^-^z^-^) null mutant embryos, both the lysosomephagosome and autophagosome-phagosome fusion are severely defective, even blocked ([2] and Figure 8 of this manuscript). To answer the above question, we quantified the phagosome degradation defect in *rab-7* mutant embryos. We measured the phagosome lifespan in *rab-7* mutant embryos (Figure 6) and compared the result to that measured in the *atg-7*, *lgg-1*, and *lgg-2* mutant embryos. We found that the phagosome lifespan is significantly longer in *rab-7* mutants than in all of the other mutants (Figure 6, Results, “RAB-7 and the HOPS complex are essential for the fusion between autophagosomes and phagosomes”). The severe versus intermediate phagosome degradation defects displayed by *rab-7* and autophagy mutants, respectively, strongly suggest that autophagosomes and lysosomes both contribute to phagosome degradation, and that autophagosomes provide a phagosome degradation activity that is in an additive manner to lysosomes.

In the event that there are major technical limitations to performing these studies, please explain these.Reviewer #1 (Recommendations for the authors):1. Rather than using red and green for fluorescence images, I suggest using magenta and green, especially in those figures were the two colors our overlaid.

Thank you for the suggestion. We have converted all the red images to magenta.

2. The authors use a visual assay and genetics to provide evidence that the LC3-positive vesicles that fuse with phagosomes are double-membrane vesicles and autophagosomes. Is there any evidence for double-membrane vesicles from EM images that the authors may have or are aware of in the literature?

1. Previously, using electron microscopy and immunogold labeling, Manil-Segalen et al., have shown that the GFP::LGG-1 and GFP::LGG-2 labeled puncta observed in *C. elegans* embryos are double-membrane autophagosomes [1]. This result established LGG-1 and LGG-2 as specific markers for autophagosomes.

2. Gonadal sheath cells are engulfing cells for apoptotic germ cells. Using transmission EM (TEM), we have identified, inside gonadal sheath cells, double-membrane vesicles that are attached to the phagosomes containing germ cell corpses. The morphology of these vesicles resembles autophagosomes. This observation provides a strong support to our model that autophagosomes are recruited to phagosomes and subsequently fuse to phagosomes. Please see our answer under “Essential Revisions – Response to request 1)**”** for a detailed answer to the above question.

3. Figure 6. Could the authors provide a quantification of LGG-1, LGG-2 and double positive vesicles? And could RAB-7 specifically be on LGG-1- and LGG-2 positive vesicles (Figure 7)?

Yes. We have quantified the distribution pattern of the three classes of the LGG-labeled phagosomes in the engulfing cells for cell corpses C1, C2, and C3, and found that the average distribution pattern of LGG-1^+^ LGG-2^-^, LGG-1^-^ LGG-2^+^, and LGG1^+^ LGG-2^+^ puncta is 35.8%, 40.7%, and 23.5%, respectively, indicating the existence of a comparable amount of each class of puncta in engulfing cells. We summarized the result of the quantification in a graph in the revised Figure 7C.

In wild-type embryos that co-express mCherry::LGG and GFP::RAB-7 reporters, we also quantified the percentage of LGG^+^ puncta that are also RAB-7^+^. We found that in the cytoplasm of the ventral hypodermal cells, on average 66.2% of LGG-1^+^ and 62.5% of LGG-2^+^ puncta are RAB-7^+^, respectively (revised Figure 8E). Remarkably, the LGG-1^+^ or LGG-2^+^ puncta found on the surfaces of phagosomes are 100% RAB-7^+^ (revied Figure 8F), suggesting that the presence of RAB-7 on autophagosomes is requested for the incorporation of autophagosomes to phagosomes. This hypothesis was proven true by our further finding presented in (Figure 8 G-J). We have reported these new results in “Results”, in a section entitled, “The small GTPase RAB-7 is enriched on the surfaces of autophagosomes”.

4. ced-1 is required first for phagosome formation and subsequently for phagosome degradation. How can the authors look at autophagosome-phagosome fusion in ced-1 null mutants? Is the phagosome formation defect only partially penetrant?

Yes, the defect in phagosome formation is only partially penetrant in the *ced1(e1735)* null mutant embryos. Two major and one minor engulfment pathways have been discovered to act in parallel to promote the engulfment of apoptotic cells [3,4]. CED-1 leads one major pathway. The other major pathway is led by the CED-10 Rac GTPase and the CED-5/CED-12 complex, a bipartite nucleotide exchange factor for CED-10 [3]. The minor pathway is controlled by the small GTPase RAB-35 [4]. In *ced1(e1735)* null mutant embryos, we have characterized the different defects of apoptotic cell clearance and observed defects in all of the following events: recognition, engulfment, and phagosome degradation [2] (Figure S4 of [3]). In the *ced-1(e135)* background, even 100 min after the appearance of the cell death morphology, in 19% of the embryos the C3 cell corpse remains unengulfed; in 81% of the embryos, C3s are engulfed inside phagosomes, albeit at a slower pace (Figure S4 of [3]). Therefore, phagosomes containing C3 are available for time-lapse characterization. The engulfment defect in *ced-1* mutants is partially penetrant due to the presence of the engulfment pathways led by CED-10 and RAB-35, as in *ced-1; ced-5* and *ced-1; rab-35* double mutants, the recognition and engulfment defects are much worse than in each of the single mutants [4](Figure S7 of [2]). In *ced-1; rab-35; ced-5* triple mutants, the recognition and engulfment defects are further enhanced over that of the double mutants [4](Figure S7 of [2]). The engulfment activities provided by the CED-10 and RAB-35 pathways thus enabled phagosome formation in a portion of the *ced-1* mutant embryos and allowed us to identify and further characterize the phagosome degradation defects caused by mutations in *ced-1*. We added this explanation to “Results**”,** in a section entitled “The CED-1 pathway drives the recruitment of autophagosomes to phagosomes”.

5. Do the authors know whether blocking lysosome-phagosome fusion AND autophagosome-phagosome fusion results in a more severe engulfment defect compared to blocking only one of these fusion events? Basically, is there an additive or synergistic effect?

To answer this question, we measured the cell corpse clearance defect, in particular, the phagosome degradation defect caused by the deletion mutation of *rab-7*. Previously, the *rab-7(ok511)* null mutants were reported to display severe defects in the incorporation of lysosomes to phagosomes [2,5]. In this manuscript, we report the blockage of autophagosome-phagosome fusion caused by the *rab-7(ok511)* mutation. Therefore, in *rab-7* mutants, the fusion events of both lysosomes and autophagosomes to phagosomes are severely defective or blocked. Previously we reported that the *rab-7* null mutation blocked phagosome degradation [2]. Here we measured the phagosome degradation defect in *rab-7(ok511)(m^-^z^-^)* embryos using the same assay used to measure the degradation efficiency in the *atg-7*, *lgg-1,* and *lgg-2* mutants (Figure 6). We found that the *rab-7* mutation nearly completely blocks phagosome degradation, yet the *atg-7* and *lgg* mutations only delay the degradation to different extents. These results indicate that the autophagosomes contribute to phagosome degradation in a manner additive to lysosomes. We reported the new results in the revised Figure 6, and described it in “Results”, in a section entitled “RAB-7 and the HOPS complex are essential for the fusion between autophagosomes and phagosomes”.

Reviewer #2 (Recommendations for the authors):My main concern is really intepretation and the model. Do autophagosomes fuse with phagosomes and contribute to corpse digestion as suggested?EM showing double-membrane nature of these autophagosomes seems like it is missing and might be needed. The arguments are all based on marker topology, which is reasonable, but a phagosome is a dynamic thing. For instance, as phagosomes digest contents like corpses, they get smaller. Does that membrane get internalized into the phagosome? If so, how does that affect the interpretation of the markers going inside the phagosome at later time points?

During the revision period, we have performed transmission EM analysis and observed the close association of double membrane vesicles with phagosomes that contain germ cell corpses. These double-membrane vesicles resemble autophagosomes in morphology. Please see “Essential Revisions – Response to request 1)” for a detailed explanation of the rationale of this experiment, and the new Figure 4, and the section entitled “The autophagosomes-phagosome interaction is a general phenomenon observed in embryos and the adult gonad” in “Results” for the description of experimental results. Consistent with the EM observation, in the gonadal sheath cells, puncta labeled with mCherry::LGG-1 are observed to enrich on phagosomal surfaces, similar to the enrichment pattern on phagosomal surfaces in embryos. Together, these results suggest that like in the gonad, in embryos, the LGG-labeled puncta attached to phagosomal surfaces are very likely double-membrane organelles that resemble autophagosomes.

What if this is a mechanism that allows phagosomes to grow their lipid membranes, rather than fusion to digest what's in the autophagosome? Atg2/9 drive the transfer of lipids from the ER to autophagosomes, that's how autophagosomes grow. Maybe autophagosomes in this situation are intermediates that serve as important lipid sources for phagosomes? Is there an autophagosome target (inside) or component not in phagosomes that one could track to show actual deposition and subsequent degradation? How else can these possibilities be discriminated?

This hypothesis is conceivable. The incorporation of intracellular vesicles to phagosomes would definitely cause the expansion of the phagosomal membrane. One particular function of the increased phagosome membrane amount might be to support the extension of the transient phagosomal tubules that aids the recruitment of lysosomal particles to phagosomes. We added it to the revised “Discussion”.

The argument that they don't fuse is reasonable, but what if autophagosomes themselves are engulfed completely into the phagosome? They show a lysosomal marker doesn't do that as an argument, which is fine, but that really only says lysosomes are acting as they normally do in this context and that their assays would detect that. What if autophagosomes are different? In the next section on ATG7 they assert this (internal fluorescence) as a marker for insertion of the internal part of a double-membrane vesicle into a phagosome, but it may or may not be.

We propose that the double-membrane autophagosomes fuse with phagosomes primarily because we have observed not only the attachment of LGG-labeled autophagosomes to the phagosomal surfaces but also the subsequent accumulation of three autophagosomal membrane markers: LGG-1, LGG-2, and ATG-9 inside the phagosomal lumen. The entry of these markers to phagosomal lumen require two key properties: (1) the vesicles that fuse to the phagosomal membrane are of double membrane, and (2) the fusion between the outer membrane of the vesicle and the phagosomal membrane must happen, allowing the entry of the inner circle into the phagosomal lumen. The inner membrane is also labeled with one or more of the above markers. On the other hand, the fusion of the single membrane lysosomes to the phagosomal membrane results in the incorporation of the CTNS^-1^ marker into the phagosomal membrane but not the phagosomal lumen (Figure 1 —figure supplement 1). Figure 1K illustrates how the fusions of single-membrane vesicles or double-membrane vesicles to phagosomes result in the different localization patterns of the vesicle membrane marker. If the outer autophagosomal membrane does not fuse with the phagosomal membrane, it is hard to explain how the LGG markers are observed inside the phagosomal lumen. The essential roles of RAB-7 and VPS^-1^8 in the entry of the LGG^+^ puncta into the phagosomal lumen (see the next paragraph) further support the fusion between autophagosomes and phagosomes.

Regarding the possibility that autophagosomes might be engulfed into the phagosome, formally it is possible; however, in the literature, there is no report of phagosomes further engulfing other vesicles in the cytoplasm. In the endocytic pathway, vesicles undergo dynamic conversions primarily through fusion and fission and the fusion and fission events retain the topology of the vesicles. In the aspect of membrane trafficking, phagosomes behave like lysosomes except that phagosomes are much bigger in size. For example, the small GTPase RAB-7 and the HOPs complex are enriched on the surfaces of both phagosomes and lysosomes and act as membrane tethering factors that facilitate fusion. It is known that Rab7 is essential for mediating the fusion between autophagosomes and lysosomes. In this study, we found that mutations in *rab-7* and *vps^-1^8* block or severely impair the entry of the LGG markers into the phagosomal lumen, respectively, although they do not affect the attachment of LGG^+^ puncta on the phagosomal surfaces (Figure 8). These results provide strong support to the hypothesis that autophagosomes actually fuse to phagosomes.

What if LAPs in worms don't need ulk1/atg13/atg14, and they are single-membrane LAPs that are made differently? The data suggest these vesicles are double-membrane, but there are alternate interpretations (e.g. single vesicles that go in, and the stable reporters are showing that). How else are these different, or the same as LAPs in mammals?

Mammalian LAPs are defined as single-membrane vesicles with LC3 on their surfaces. The biogenesis of mammalian LAP vesicles does not need the Ulk1, Atg13, or Atg14 proteins, but needs other proteins required for the biogenesis of autophagosomes. In *C. elegans*, in the atg-13 and epg-8 mutant embryos, like in the atg-7 mutant embryos, very few LGG^+^ puncta were observed (Figure 2 —figure supplements 1-2). This observation argues against the existence of a substantial pool of atg-13-independent LAP vesicles in C. elegans embryos, at least not in the hypodermal cells, or in the pharyngeal or intestinal precursor cells, in which P_ced-1_ is active. This observation thus reveals a significant difference between C. elegans embryonic cells and cultured mammalian cells, in the latter of which LAP vesicles were observed. In addition, although Fazeli et al., [7] claimed that during C. elegans embryogenesis, the 2^nd^ polar body is cleared by LAP, these authors did not attempt to observe LAP vesicles [7]. Therefore, whether the singlemembrane LAP vesicles exist in C. elegans embryos has not been demonstrated. It is possible that the biogenesis of the LAP vesicles might be induced by environmental stresses. However, stress-induction of LAP vesicles has not been reported in C. elegans.

The idea here is that the autophagosomes are needed for digestion of corpses, but what if corpses and stuff in autophagosomes are co-degraded in phagosomes? In this model, the autophagosomes would be additional material digested in what started as a phagosome. If that is blocked with mutations affecting autophagy, it could be that the delays in digestion of corpses observed are due to a checkpoint that is relevant to autophagosome fusion/digestion, rather than there being a direct role for autophagosomes in cell corpse degradation.

The model proposed above is formally possible, although currently we do not know whether such a checkpoint exists. In engulfing cells, when autophagosomes are fused to phagosomes, the cargos of autophagosomes are indeed co-degraded inside phagosomes. On the other hand, in most other cells, the cargos in autophagosomes are primarily degraded by lysosomes after the fusion of autophagosomes and lysosomes. Autophagy is a phenomenon that occurs broadly, not limited to cells that engulf apoptotic cells. Given that lysosomes are the presumed major force for degrading autophagosomal cargos, it is unclear whether phagosomes would make a significant contribution to the degradation of autophagosomal cargos. Therefore, such a checkpoint, if exists, does not seem to significantly facilitate the degradation of cargos of autophagosomes in general.

I think the authors have done a great job convincing me that autophagosomes are important, and that all the genes they describe are needed, but I am not fully convinced of the model they have landed on. It seems like there are many other possibilities. I think those need to be discussed.

We agree with Reviewer 2 that there are multiple possible mechanisms behind the novel observations reported in this manuscript. This report is not sufficient to answer the fundamental question of how autophagosomes facilitate phagosome degradation.

We believe that to identify the answer to this question, a substantial body of future investigation is needed. In the “Discussion” of the revised manuscript, we discussed multiple mechanisms that autophagosomes might employ to facilitate phagosome degradation, including the mechanisms raised by Reviewer 2. In summary, the points we propose are:

1. The LGG^+^ puncta is a combination of two subpopulations: autophagosomes that are already fused to lysosomes and become autolysosomes (LGG^+^ NUC-1^+^) and autophagosomes that have not fused to lysosomes (LGG^+^ NUC-1^-^) (Figure 12). Autolysosomes might contribute multiple activities to phagosomes, one of which is an activity that is similar to that contributed by lysosomes that are fused to phagosomes. On the other hand, multiple lines of evidence suggest that autophagosomes also contribute activities that are different from that of lysosomes.

2. What could be those unique mechanisms provided by the autophagosomes that facilitate phagosome degradation?

(a) The fusion, as suggested by Reviewer 2, incorporates the outer membrane of the autophagosome to the phagosomal membrane and thus should cause the expansion of the phagosomal membrane. Increasing the overall amount of phagosomal membrane might aid the degradation of the apoptotic cell by facilitating the recruitment of lysosomes to the phagosomal surfaces. We previously reported that phagosomes extend transient lipid tubules to capture lysosomal particles in the cytoplasm and bring these particles back to the phagosomes for fusion [2,8]. It is possible that the increased phagosomal membrane material supports the extension of these lipid tubules. -- (b) The second possible mechanism proposes that the outer membrane of the autophagosomes provides important signaling molecules to the phagosomal membrane.

These molecules could be proteins and/or lipids. One candidate signaling molecule is PtdIns(3)P. On the phagosomal surfaces, PtdIns(3)P recruits PtdIns(3)P-binding proteins including the sorting nexins SNX-1, SNX-6, and LST-4/SNX-9 and the HOPs complex, which subsequently drive multiple membrane remodeling events that promote phagosome maturation [8,9]. In *C. elegans*, PtdIns(3)P is presented on phagosomal membrane over time in a two-peak pattern [10]. Autophagosomes are coated with PtdIns(3)P. Judging by the timing of autophagosome-phagosome fusion, autophagosomal membranes are likely to provide PtdIns(3)P for the second peak.

In our future study, we will test the above models. The analyses will reveal molecular mechanisms that support this novel function of autophagosomes and the crosstalk between autophagosomes and phagosomes. We include the above points in the revised “Discussion”.

Reviewer #3 (Recommendations for the authors):The authors should try to assess the pH and hydrolytic activity inside the phagosomes that matured in the presence or absence of autophagosomes to more convincingly demonstrate that lysosomal fusion occurred normally and that, importantly, a new, heretofore unappreciated factor is required for proper degradation of the contents.

Thanks to Reviewer 3 for the above suggestions. Below I will first describe the new experimental results we have obtained during the revision period and then discuss our thoughts regarding the mechanisms of how autophagosomes contribute to phagosome degradation.

We developed a P*_his-72_ his-72::gfp::mCherry* reporter as a pH reporter for the phagosomal lumen. The HIS-72::GFP::mCherry fusion protein inside the nucleus of an apoptotic cell is internalized into the phagosomal lumen when engulfment occurs. Due to GFP, but not mCherry, being sensitive to acidic pH, in the phagosomal lumen, the GFP signal will gradually reduce over time while the mCherry signal remains stable. We thus use the signal intensity ratio of GFP/mCherry as an index of acidification over time.

We found that, compared to wild-type, in *atg-7* mutants, acidification is delayed. However, compared to the *cup-5* mutants, in which the lysosomal function is severely defective, the acidification defect in *atg-7* mutants appears to be rather modest. *cup-5* encodes a lysosomal TRP channel protein and a homolog of human Mucolipin IV.

We also developed an assay to monitor the hydrolytic activity of NUC-1, a lysosomal endonuclease called DNaseII. We found that NUC-1 acts in the phagosomal lumen of the engulfing cells to degrade chromatin DNA of the engulfed apoptotic cells (unpublished results). During the above study, we established HIS-72::mCherry as a reporter for monitoring the degradation of the chromatin DNA. HIS-72, a histone B H3.3, together with other histones and the DNA wrapping around them, form the nucleosome, the fundamental subunit of the chromatin.

Inside the phagosomal lumen, the HIS-72::mCherry signal appears as a condensed red disc, representing the nucleus of the engulfed apoptotic cell. We found that in the wildtype strain, the degradation of chromatin DNA of the apoptotic cell inside the phagosomal lumen resulted in the subsequent reduction of the size of the mCherry^+^ disc. On the contrary, in the *nuc-1* mutant background, the size of the mCherry^+^ disc inside the phagosomal lumen remains the same over a long period of time (>60 min), indicating the lack of degradation of chromatin DNA (unpublished result). Thus the reduction of the size of the HIS-72::mCherry disc is a good representation of the DNA degradation activity of NUC-1. Besides *nuc-1* mutants, in *cup-5(n3264)* mutant embryos, within 60 min post phagosome formation, the average diameter of the mCherry^+^ disc remains at 89% of the 0 min-value (Figure 14E). These above results have validated HIS72::mCherry as a chromatin DNA digestion marker. In *atg-7(bp411)* mutant embryos, that value is reduced to 66% of the 0 min-value; in addition, the diameter of the mCherry^+^ disc in 25% of the samples is > 80% of the 0 min-value. These results indicate that in *atg-7* mutants, there is a defect in the digestion of chromatin DNA, yet this defect is rather moderate.

Overall, the two new assays allowed us to identify the modest defects in phagosome acidification and DNA digestion caused by the *atg-7(bp411)* mutation. These defects likely contribute to the inefficient phagosome degradation. However, it is not clear whether these modest defects are sufficient to cause the overall delay of phagosome degradation observed from the autophagy mutants (Figure 6). Less than 41% of the LGG^+^ puncta population observed to fuse with phagosomes is autolysosomes (LGG^+^ NUC-1^+^ puncta). The autolysosome population might contribute lysosomal activity to phagosomes after fusion. Meanwhile, the rest of the LGG^+^ population, which are autophagosomes that are not fused to lysosomes, are likely to contribute non-lysosomal activities to phagosomes after fusion. Actually, autolysosomes are also capable of contributing non-lysosomal activities to phagosomes. The non-lysosomal activities might regulate a currently unknown aspect(s) of phagosome degradation. Alternatively, they might regulate the acidification and/or DNA hydrolysis in a manner independent of lysosomes. We revised the Discussion section, adding our hypothesis of the possible contribution of both the lysosomal and non-lysosomal activities by the autophagosomes and autolysosomes populations. We also proposed two alternative candidate substances from autophagosomes that might act to facilitate phagosome degradation. (Please also see our response to Reviewer #2’s “Recommendations for the authors, point 6”.)

They should also establish the nature of the fusing structures: are they truly double-membraned autophagosomes or have they undergone fusion with lysosomes (i.e. become autolysosomes)?

Using transmission electron microscopy (TEM), we have detected multiple double-membrane vesicles on the surfaces of phagosomes that contain germ cell corpses (Figure 4). These vesicles resemble the morphology of autophagosomes (Figure 4).

This evidence supports the conclusion that autophagosomes fuse to phagosomes. Please see our “Essential Revisions – Response to request 1)” for the description of the TEM experiment. In addition, as already mentioned, in strains co-expressing the LGG and NUC-1 markers, 59.3% and 63.5% of LGG-1^+^ and LGG-2^+^ vesicles found on phagosomal surfaces and subsequently fuse to phagosomes are NUC-1^-^, respectively (Figure 12), which indicate that the majority of the LGG^+^ puncta that fuse to phagosomes are autophagosomes that are not fused to autolysosomes. Furthermore, in *rab-7* mutants, in which the fusion between autophagosomes and lysosomes is blocked, LGG^+^ particles are heavily enriched on the surfaces of phagosomes (Figure 8), further demonstrating that autophagosomes are able to attach to phagosomes without prior fusion to lysosomes.

References

1. Manil-Segalen M, Lefebvre C, Jenzer C, Trichet M, Boulogne C, et al. (2014) The C.

elegans LC3 acts downstream of GABARAP to degrade autophagosomes by

interacting with the HOPS subunit VPS39. Dev Cell 28: 43-55.

2. Yu X, Lu N, Zhou Z (2008) Phagocytic receptor CED-1 initiates a signaling pathway

for degrading engulfed apoptotic cells. PLoS Biol 6: e61.

3. Reddien PW, Horvitz HR (2004) The engulfment process of programmed cell

death in *Caenorhabditis elegans*. Annu Rev Cell Dev Biol 20: 193-221.

4. Haley R, Wang Y, Zhou Z (2018) The small GTPase RAB-35 defines a third

pathway that is required for the recognition and degradation of apoptotic

cells. PLoS Genet 14: e1007558.

5. Guo P, Hu T, Zhang J, Jiang S, Wang X (2010) Sequential action of Caenorhabditis

elegans Rab GTPases regulates phagolysosome formation during apoptotic

cell degradation. Proc Natl Acad Sci U S A 107: 18016-18021.

6. Tian Y, Li Z, Hu W, Ren H, Tian E, et al. (2010) *C. elegans* screen identifies

autophagy genes specific to multicellular organisms. Cell 141: 1042-1055.

7. Fazeli G, Stetter M, Lisack JN, Wehman AM (2018) *C. elegans* Blastomeres Clear

the Corpse of the Second Polar Body by LC3-Associated Phagocytosis. Cell

Rep 23: 2070-2082.

8. Lu N, Shen Q, Mahoney TR, Liu X, Zhou Z (2011) Three sorting nexins drive the

degradation of apoptotic cells in response to PtdIns(3)P signaling. Mol Biol

Cell 22: 354-374.

9. Lu N, Zhou Z (2012) Membrane trafficking and phagosome maturation during the

clearance of apoptotic cells. Int Rev Cell Mol Biol 293: 269-309.

10. Lu N, Shen Q, Mahoney TR, Neukomm LJ, Wang Y, et al. (2012) Two π 3-kinases

and one π 3-phosphatase together establish the cyclic waves of phagosomal

PtdIns(3)P critical for the degradation of apoptotic cells. PLoS Biol 10:

e1001245.

11. Li Z, Venegas V, Nagaoka Y, Morino E, Raghavan P, et al. (2015) Necrotic Cells

Actively Attract Phagocytes through the Collaborative Action of Two Distinct

PS-Exposure Mechanisms. PLoS Genet 11: e1005285.

12. Gao J, Langemeyer L, Kummel D, Reggiori F, Ungermann C (2018) Molecular

mechanism to target the endosomal Mon1-Ccz1 GEF complex to the preautophagosomal

structure. *eLife* 7.

13. Hegedűs K, Takats S, Boda A, Jipa A, Nagy P, et al. (2016) The Ccz1-Mon1-Rab7

module and Rab5 control distinct steps of autophagy. Mol Biol Cell 27: 3132-

3142.